# A Federated Generalized Expectation-Maximization Algorithm for Mixture Models with an Unknown Number of Components

**Michael Ibrahim, Nagi Gebraeel & Weijun Xie**
H. Milton Stewart School of Industrial and Systems Engineering
Georgia Institute of Technology
Atlanta, GA 30332, USA
`{mibrahim41,ngebraeel3,wxie}@gatech.edu`

## Abstract

We study the problem of federated clustering when the total number of clusters $K$ across clients is unknown, and the clients have heterogeneous but potentially overlapping cluster sets in their local data. To that end, we develop `FedGEM`: a federated generalized expectation-maximization algorithm for the training of mixture models with an unknown number of components. Our proposed algorithm relies on each of the clients performing EM steps locally, and constructing an uncertainty set around the maximizer associated with each local component. The central server utilizes the uncertainty sets to learn potential cluster overlaps between clients, and infer the global number of clusters via closed-form computations. We perform a thorough theoretical study of our algorithm, presenting probabilistic convergence guarantees under common assumptions. Subsequently, we study the specific setting of isotropic GMMs, providing tractable, low-complexity computations to be performed by each client during each iteration of the algorithm, as well as rigorously verifying assumptions required for algorithm convergence. We perform various numerical experiments, where we empirically demonstrate that our proposed method achieves comparable performance to centralized EM, and that it outperforms various existing federated clustering methods.

## 1 Introduction

Original equipment manufacturers (OEMs) of capital-intensive industrial systems, such as power generators and medical imaging systems, often enter into lucrative long-term service contracts (LTSCs) with their clients, guaranteeing adherence to stringent reliability standards. Failure to meet such guarantees can incur multi-million dollar penalties (Schimmoller, 2001; Thompson et al., 2003). To manage these risks, OEMs must be able to accurately detect and diagnose faults in a timely manner (Lei et al., 2020; Dutta et al., 2023; Shuming et al., 2024). However, OEMs face several critical challenges. First, OEMs **do not have prior knowledge of all the possible fault classes**. Second, OEMs cannot rely solely on labels provided by their clients due to the absence of a global labeling standard and differing maintenance practices, which result in inconsistent labels. Third, clients cannot readily share their raw data with the OEM due to the size and dimensionality of the data, and privacy concerns. Thus, centralized model training is infeasible.

Federated Learning (FL) (McMahan et al., 2017; Konečný et al., 2016) offers a promising solution. However, the majority of existing FL efforts (Li et al., 2020; Wang et al., 2020; Karimireddy et al., 2020; Arivazhagan et al., 2019; Lee et al., 2023) assume that all clients share identical cluster sets and are primarily focused on supervised learning. Unfortunately, these assumptions are not compatible with our problem setting as they violate one or more of the critical challenges mentioned above. Recent works on unsupervised FL (Dennis et al., 2021; Stallmann & Wilbik, 2022; Garst & Reinders, 2024; Bárcena et al., 2024; Yfantis et al., 2025) relax the assumption of identical cluster sets across clients. However, they still assume that the server knows the total number of unique clusters in advance—again, an assumption that does not hold in our problem setting. While this assumption

is relaxed by Zhang et al. (2025), their algorithm has two critical limitations: (i) it requires clients to share arrays of the same size (i.e. *dimensionality and cardinality*) as the raw data, and (ii) client data can be easily reconstructed at the central server via simple computations on the information shared by the clients, causing a violation of privacy.

This paper focuses on developing an **unsupervised federated learning methodology** for distributed clustering with an unknown number of clusters across privacy-constrained clients with high-dimensional data. Our methodology enables a central server to (i) infer the total number of distinct clusters (components) that emerge across all clients without requiring access to raw data or prior knowledge of the cluster count, and (ii) determine the cluster memberships of each client.

**Contributions.** We introduce `FedGEM`: the first federated generalized expectation-maximization (GEM) algorithm that can be used for the training of mixture models **without prior knowledge of the global number of components**. Our algorithm allows clients with overlapping clusters to collaborate on the training of cluster centers, whereas cluster weights are set locally at each client. This allows for model personalization, where local cluster weights can adapt the global model to client-specific distributions. We summarize our main contributions next.

1. We develop the first federated GEM (`FedGEM`) algorithm for the training of mixture models **without prior knowledge of the total number of components**. Our algorithm relies on uncertainty sets obtained by each client for each local component by solving an optimization problem. Intersections between the uncertainty sets enable the central server to detect cluster overlaps between clients via **closed-form computations**, allowing for collaborative model training.

2. We rigorously study the convergence properties of our algorithm and prove that iterates converge to a neighborhood of the ground truth model parameters with a certain probability under common assumptions. This allows our algorithm to correctly estimate the true total number of unique clusters.

3. We examine various theoretical aspects of our proposed algorithm in the context of multi-component isotropic Gaussian Mixture Models (GMMs). To that end:

    (a) We derive a low-complexity, tractable, and bi-convex reformulation of the optimization problem that is solved by the client to obtain the local uncertainty sets.

    (b) We prove the first-order stability (FOS) condition for multi-component isotropic GMMs, which allows us to study the contraction region and prove convergence of our proposed algorithm.

4. We perform a thorough empirical evaluation on popular and synthetic datasets, showing that our algorithm outperforms state-of-the-art ones while scaling well with problem size, at times even outperforming methods with prior knowledge of the cluster count. We also highlight our algorithm's strong performance in various problem settings, including ones that violate modeling assumptions.

## 2  RELATED WORKS

**Federated Learning.** The canonical FL algorithm, `FedAvg` (McMahan et al., 2017), is primarily designed for *supervised* deep learning. It aggregates model gradients across clients to train a single global model. However, it can perform poorly under non-IID client data, often converging to suboptimal solutions. Numerous methods address this issue, including `FedProx` (Li et al., 2020), `FedNova` (Wang et al., 2020), SCAFFOLD (Karimireddy et al., 2020), `FedPer` (Arivazhagan et al., 2019), and `FedL2P` (Lee et al., 2023). However, these efforts overwhelmingly focus on supervised settings and assume that all clients have *identical cluster sets* in their training data.

Several works have attempted to relax the common cluster set assumption. For example, `FedEM` (Marfoq et al., 2021) trains a global mixture model with localized component weights to support personalization. A different version of `FedEM` is introduced by Dieuleveut et al. (2021), focusing on reducing client heterogeneity. Additionally, `FedGMM` (Wu et al., 2023) tackles covariate shift using Gaussian mixtures. However, these methods assume **prior knowledge of the global number of components**, making them unsuitable for real-world problems with unknown cluster counts.

**Federated Clustering.** Recent efforts have explored *unsupervised* federated clustering, allowing clients to have heterogeneous cluster sets. Examples of such efforts include k-FED (Dennis et al., 2021), FFCM (Stallmann & Wilbik, 2022), and `FedKmeans` (Garst & Reinders, 2024), among others (Bárcena et al., 2024; Yfantis et al., 2025). However, these works still require **prior knowledge of the global number of clusters**, limiting their applicability in many real-world problems.

To the best of our knowledge, only AFCL (Zhang et al., 2025) attempts federated clustering without requiring prior knowledge of the global cluster number. However, this work involves clients sharing arrays of the same size as the local data. It also suffers from significant privacy vulnerabilities that allow data reconstruction at the server via simple scalar multiplication and subtraction operations.

**Centralized Clustering with an Unknown Cluster Number.** A canonical example of such models is the Dirichlet Process Gaussian Mixture Model (DP-GMM) (Antoniak, 1974), which extends GMMs by placing a nonparametric Dirichlet Process prior over the mixture components. Other approaches include the density-based DBSCAN (Ester et al., 1996), the scalable variational inference approach for DP mixture models (Hughes & Sudderth, 2013), and the neural network-based DeepDPM (Ronen et al., 2022). However, all of these methods assume centralized access to the full training dataset.

## 3 PROBLEM SETTING

We consider a federated clustering problem with $G$ clients and an **unknown** number $K$ of total clusters (we use "cluster" and "component" interchangeably). Each client $g$ has access to $N_g$ local data samples $\{\widehat{\boldsymbol{x}}_{n_g}\}_{n_g=1}^{N_g}$ generated from a local mixture model $\mathcal{M}_g(\boldsymbol{x}) = \sum_{k_g=1}^{K_g} \pi_{k_g} p_{k_g}(\boldsymbol{x}|\boldsymbol{\theta}_{k_g}^*)$, where $K_g$ is the local number of clusters, and $p_{k_g}(\boldsymbol{x}|\boldsymbol{\theta}_{k_g}^*)$ are the independent component distributions parameterized by ground truth parameters $\boldsymbol{\theta}_{k_g}^*$ and weighted by fixed $\pi_{k_g}$ for all $k_g \in [K_g]$. We denote the vectorized concatenation of all ground truth parameters at client $g$ by $\boldsymbol{\theta}_g^*$. We assume that $K_g$ is known for all clients $g \in [G]$, whereas the global $K$ is **unknown**. We also assume that clients may have some overlapping clusters, but no client has all the clusters locally, i.e., $2 \leqslant K_g < K, \ \forall g \in [G]$.

We denote the minimum and maximum distances between any two unique ground truth cluster parameters by $R_{\min}$ and $R_{\max}$, respectively. That is $R_{\min} = \min_{i,j\in[K],i\neq j} ||\boldsymbol{\theta}_i^* - \boldsymbol{\theta}_j^*||_2$ and $R_{\max} = \max_{i,j\in[K],i\neq j} ||\boldsymbol{\theta}_i^* - \boldsymbol{\theta}_j^*||_2$. These quantities are used to study the convergence behavior of our algorithm and do not need to be known in advance to use our algorithm. We make the following crucial assumption to support algorithmic convergence analysis (in Section 4).

**Assumption 1** (Ground Truth Parameters)**.** Each global cluster $k \in [K]$ is parameterized by a fixed ground truth $\boldsymbol{\theta}_k^*$ that is consistent across all clients where the cluster is present. However, the weight assigned to the cluster may vary locally across clients.

*Remark* 1. Assumption 1 motivates our algorithm design, where clients with overlapping clusters can collaborate on learning the shared cluster parameters while retaining personalized cluster weights. This respects the non-IID nature of the federated data while enabling collaborative training.

At client $g$, we denote the local population expected complete-data log-likelihood by

$$Q_g(\boldsymbol{\theta}_g|\boldsymbol{\theta}_g') := \mathbb{E}_{\boldsymbol{x}\sim\mathcal{M}(\boldsymbol{x})}\left[\sum_{k_g=1}^{K_g} \gamma_{k_g}(\boldsymbol{x},\boldsymbol{\theta}_g')\log(\pi_{k_g}p_{k_g}(\boldsymbol{x}|\boldsymbol{\theta}_{k_g}))\right],$$

where $\gamma_{k_g}(\boldsymbol{x},\boldsymbol{\theta}_g')$ is the posterior responsibility function of the $k_g^{th}$ component, computed using current parameters $\boldsymbol{\theta}_g'$. Similarly, we denote the local finite-sample expected complete-data log-likelihood function by

$$\widehat{Q}_g(\boldsymbol{\theta}_g|\boldsymbol{\theta}_g') := \frac{1}{N_g}\sum_{n_g=1}^{N_g}\sum_{k_g=1}^{K_g} \gamma_{k_g}(\widehat{\boldsymbol{x}}_{n_g},\boldsymbol{\theta}_g')\log(\pi_{k_g}p_{k_g}(\widehat{\boldsymbol{x}}_{n_g}|\boldsymbol{\theta}_{k_g})).$$

## 4 OVERVIEW OF FEDGEM ALGORITHM

Our proposed `FedGEM` algorithm consists of two stages: (i) an iterative collaborative training stage, and (ii) a single-step final aggregation stage. (**Pseudo-code in Algorithm 1 in Appendix A.1**).

The collaborative training stage can be summarized as follows. **Client:** (i) performs (potentially multiple) EM steps locally, (ii) solves an optimization problem to obtain the radius of an uncertainty

set for each component centered at its corresponding maximizer of $\widehat{Q}_g(\boldsymbol{\theta}_g|\boldsymbol{\theta}'_g)$, and (iii) broadcasts the maximizer and uncertainty set radius pair for each component to the server. **Server:** performs aggregation using overlaps between uncertainty sets and re-broadcasts updates to clients.

In the final aggregation step, the server merges cluster estimates from different clients if they are within a specific radius of each other. This enables the server to estimate the total number of unique global clusters and determine the cluster membership of each client. Before discussing our algorithm, we make the following assumption, which can be verified for common models such as GMM.

**Assumption 2** (Strong Concavity). *Each of the $K_g$ terms in the population $Q_g(\boldsymbol{\theta}_g|\boldsymbol{\theta}'_g)$ or finite-sample $\widehat{Q}_g(\boldsymbol{\theta}_g|\boldsymbol{\theta}'_g)$ at client $g$ are strongly concave in $\boldsymbol{\theta}_g$ for all $g \in [G]$.*

### 4.1 CLIENT COMPUTATIONS

Each client $g$ performs two vital tasks during each iteration $t$ of our algorithm: (i) it performs (potentially multiple) EM steps given current model parameters $\boldsymbol{\theta}_g^{(t-1)}$, and (ii) it solves for the radius $\varepsilon_{k_g}^{(t)}$ of the uncertainty set $\mathcal{U}_{k_g}^{(t)}$ associated with the maximizer of each local component $k_g$. Firstly, we examine the EM steps, which are displayed next.

$$\textbf{E-step:} \quad \gamma_{k_g}(\widehat{\boldsymbol{x}}_{n_g}, \boldsymbol{\theta}_g^{(t-1)}) \leftarrow \frac{\pi_{k_g} p_{k_g}(\widehat{\boldsymbol{x}}_{n_g}|\boldsymbol{\theta}_{k_g}^{(t-1)})}{\sum_{j_g=1}^{K_g} \pi_{j_g} p_{k_g}(\widehat{\boldsymbol{x}}_{n_g}|\boldsymbol{\theta}_{j_g}^{(t-1)})} \quad \forall k_g \in [K_g], \ \forall n_g \in [N_g] \quad (1)$$

$$\textbf{M-step:} \quad \widehat{M}_{k_g}(\boldsymbol{\theta}_g^{(t-1)}) \leftarrow \arg\max_{\boldsymbol{\theta}_{k_g} \in \mathbb{R}^d} \sum_{n_g=1}^{N_g} \gamma_{k_g}(\widehat{\boldsymbol{x}}_{n_g}, \boldsymbol{\theta}_g^{(t-1)}) \log(\pi_{k_g} p_{k_g}(\widehat{\boldsymbol{x}}_{n_g}|\boldsymbol{\theta}_{k_g})) \quad \forall k_g \in [K_g] \quad (2)$$

Next, client $g$ solves for an uncertainty set $\mathcal{U}_{k_g}^{(t)}$ to capture potential perturbations in $\widehat{M}_{k_g}(\boldsymbol{\theta}_g^{(t-1)})$ associated with each component $k_g \in [K_g]$. This uncertainty set is defined as a Euclidean ball $\mathbb{B}_2(\widehat{M}_{k_g}(\boldsymbol{\theta}_g^{(t-1)}); \sqrt{\varepsilon_{k_g}^{(t)}})$ centered at $\widehat{M}_{k_g}(\boldsymbol{\theta}_g^{(t-1)})$ and whose radius is $\sqrt{\varepsilon_{k_g}^{(t)}}$. We construct the uncertainty set such that any iterate $\widehat{m}_{k_g}(\boldsymbol{\theta}_g^{(t-1)}) \in \mathbb{B}_2(\widehat{M}_{k_g}(\boldsymbol{\theta}_g^{(t-1)}); \sqrt{\varepsilon_{k_g}^{(t)}})$ does not decrease the finite-sample expected complete-data log-likelihood function from the previous iteration. That is

$$\sum_{n_g=1}^{N_g} \gamma_{k_g}(\widehat{\boldsymbol{x}}_{n_g}, \boldsymbol{\theta}_g^{(t-1)}) \log(\pi_{k_g} p_{k_g}(\widehat{\boldsymbol{x}}_{n_g}|\widehat{m}_{k_g}(\boldsymbol{\theta}_g^{(t-1)}))) \geq$$

$$\sum_{n_g=1}^{N_g} \gamma_{k_g}(\widehat{\boldsymbol{x}}_{n_g}, \boldsymbol{\theta}_g^{(t-1)}) \log(\pi_{k_g} p_{k_g}(\widehat{\boldsymbol{x}}_{n_g}|\boldsymbol{\theta}_{k_g}^{(t-1)})).$$

This renders our proposed algorithm an instance of a GEM, allowing it to exhibit similar convergence behavior locally at client $g$ to the EM algorithm as we show later. Client $g$ may obtain the radius $\sqrt{\varepsilon_{k_g}^{(t)}}$ of the component's $k_g$ uncertainty set by solving the following optimization problem, which admits a unique solution as we argue in Proposition 1.

$$J_{k_g}(\boldsymbol{\theta}_g^{(t-1)}) :=$$

$$\max_{\varepsilon_{k_g}} \quad \varepsilon_{k_g}$$

$$\text{s.t.} \quad \sum_{n_g=1}^{N_g} \gamma_{k_g}(\widehat{\boldsymbol{x}}_{n_g}, \boldsymbol{\theta}_g^{(t-1)}) \log(\pi_{k_g} p_{k_g}(\widehat{\boldsymbol{x}}_{n_g}|\widehat{m}_{k_g}(\boldsymbol{\theta}_g^{(t-1)}))) \geq$$

$$\sum_{n_g=1}^{N_g} \gamma_{k_g}(\widehat{\boldsymbol{x}}_{n_g}, \boldsymbol{\theta}_g^{(t-1)}) \log(\pi_{k_g} p_{k_g}(\widehat{\boldsymbol{x}}_{n_g}|\boldsymbol{\theta}_{k_g}^{(t-1)})) \ \forall \widehat{m}_{k_g}(\boldsymbol{\theta}_g^{(t-1)}) \in \mathbb{B}_2(\widehat{M}_{k_g}(\boldsymbol{\theta}_g^{(t-1)}); \sqrt{\varepsilon_{k_g}})$$

$$(3)$$

**Proposition 1** (Local Uncertainty Set Radius Problem). *Suppose Assumption 2 holds. Then, there must exist a unique solution $\varepsilon_{k_g} \geq 0$ to the optimization problem $J_{k_g}(\boldsymbol{\theta}_g^{(t-1)})$ for all components $k_g \in [K_g]$ and all clients $g \in [G]$. (Proof in Appendix C.1).*

After completing local computations, each client $g$ transmits a tuple $(\widehat{M}_{k_g}(\boldsymbol{\theta}_g^{(t-1)}), \varepsilon_{k_g}^{(t)})$ of the obtained local maximizer and uncertainty set radius for component $k_g \in [K_g]$ to the central server. This, however, only applies in the *collaborative training stage*. During the *final aggregation step*, each client transmits a tuple $(\widehat{M}_{k_g}(\boldsymbol{\theta}_g^{(t-1)}), \varepsilon_{k_g}^{\text{final}})$ to the central server, where $\varepsilon_{k_g}^{\text{final}}$ is the final aggregation radius for component $k_g$, and is treated as a user-defined hyperparameter.

## 4.2 SERVER COMPUTATIONS

In both *collaborative training* and *final aggregation* stages, the server uses the uncertainty sets $\mathcal{U}_{k_g}^{(t)} \ \forall k_g \in [K_g], \ \forall g \in [G]$ to identify cluster overlaps between clients. This allows the server to group clients' components into *super-clusters* via pairwise comparisons and a series of closed-form computations. Specifically, the server begins by initializing an estimate $\widehat{K}^{(t)} = 0$. It then checks if the uncertainty sets of components $k_g$ at client $g$ and $k_{g'}$ at client $g'$ overlap. That is, it checks if: $||\widehat{M}_{k_g}(\boldsymbol{\theta}_g^{(t-1)}) - \widehat{M}_{k_{g'}}(\boldsymbol{\theta}_{g'}^{(t-1)})||_2 \leqslant \sqrt{\varepsilon_{k_g}^{(t)}} + \sqrt{\varepsilon_{k_{g'}}^{(t)}}$. If this holds, then the server groups the two components $k_g$ and $k_{g'}$ into a single super-cluster. Consequently, if one or both of the components already belong to a super-cluster, the server performs super-cluster merge and updates $\widehat{K}^{(t)}$. If there is no overlap, the components are assigned to different super-clusters, and $\widehat{K}^{(t)}$ is updated accordingly. This repeats for all $k_g \in [K_g]$ and $k_{g'} \in [K_{g'}]$ at all clients $g, g' \in [G]$.

During the *collaborative training stage*, the server relies on uncertainty set intersections to compute an updated parameter vector $\boldsymbol{\theta}_{k_g}^{(t)}$ for component $k_g$ at client $g$. This updated vector *remains within its respective uncertainty set*, thereby facilitating convergence. This is achieved by initializing a set $\mathcal{T}_{k_g}^{(t)}$ of vectors containing only the estimate $\widehat{M}_{k_g}(\boldsymbol{\theta}_g^{(t-1)})$ for each component $k_g$ at client $g$. Subsequently, if any intersections are found between $\mathcal{U}_{k_g}^{(t)}$ and any other $\mathcal{U}_{k_{g'}}^{(t)}$ for any $g' \in [G] \backslash g$, then an optimal vector $\boldsymbol{\nu}*$ is added to both the sets $\mathcal{T}_{k_g}^{(t)}$ and $\mathcal{T}_{k_{g'}}^{(t)}$. This vector $\boldsymbol{\nu}*$ can be written as:

$$\boldsymbol{\nu}* = \widehat{M}_{k_g}(\boldsymbol{\theta}_g^{(t-1)}) + \texttt{clip}\left(0.5, 1 - \frac{\sqrt{\varepsilon_{k_{g'}}^{(t)}}}{w}, \frac{\sqrt{\varepsilon_{k_g}^{(t)}}}{w}\right)\left(\widehat{M}_{k_g}(\boldsymbol{\theta}_g^{(t-1)}) - \widehat{M}_{k_{g'}}(\boldsymbol{\theta}_{g'}^{(t-1)})\right),$$

where $w = ||\widehat{M}_{k_g}(\boldsymbol{\theta}_g^{(t-1)}) - \widehat{M}_{k_{g'}}(\boldsymbol{\theta}_{g'}^{(t-1)})||_2$, and the $\texttt{clip}(x, a, b)$ function limits the input $x$ to the range $[a, b]$. After all comparisons are complete, the server obtains the updated parameters $\boldsymbol{\theta}_{k_g}^{(t)}$ for component $k_g$ at client $g$ by aggregating all the vectors in set $\mathcal{T}_{k_g}^{(t)}$.

In contrast, in the *final aggregation step*, the server aggregates all the estimates $\widehat{M}_{k_g}(\boldsymbol{\theta}_g^{(t-1)})$ of components $k_g$ that belong in the same super-cluster. This ensures that clients eventually reach consensus on the parameters of shared clusters. We present the server computations pseudo-code in Appendix A.2. We also present a method for potentially improving the efficiency of the server computations and an analysis of communication costs incurred by our algorithm in Appendix B.5.

## 4.3 CONVERGENCE ANALYSIS

We provide a convergence analysis for our algorithm in the finite-sample setting. This is built upon a population convergence analysis, which we provide in Appendix B.1. The idea in our convergence proofs is to show that an algorithm for component $k_g$ at client $g$ whose iterates are $\hat{m}_{k_g}(\boldsymbol{\theta}_g^{(t-1)}) \in \mathbb{B}_2(\widehat{M}_{k_g}(\boldsymbol{\theta}_g^{(t-1)}); \sqrt{\varepsilon_{k_g}^{(t)}})$ converges at a desirable rate to some neighborhood of the true parameters $\boldsymbol{\theta}_{k_g}^*$. This ensures that estimates of the same component from different clients can eventually be aggregated due to their proximity at convergence. Our convergence analysis relies on the FOS property introduced by Balakrishnan et al. (2014), which is defined next. Subsequently, we provide key technical assumptions, followed by our convergence results.

**Definition 1** (First-Order Stability)**.** The expected complete-data log-likelihood function $Q(\cdot|\boldsymbol{\theta})$ is said to obey first-order stability with parameter $\beta$ if for any $\boldsymbol{\theta}_k \in \mathbb{B}_2(\boldsymbol{\theta}_k^*; a) \ \forall k \in [K]$ we have that

$$||\nabla Q(M(\boldsymbol{\theta})|\boldsymbol{\theta}) - \nabla Q(M(\boldsymbol{\theta})|\boldsymbol{\theta}^*)||_2 \leqslant \beta||\boldsymbol{\theta} - \boldsymbol{\theta}^*||_2, \tag{4}$$

where $\beta \in \mathbb{R}$ is a constant, and $\boldsymbol{\theta}$ is the vectorized concatenation of all $\boldsymbol{\theta}_k$ for all $k \in [K]$.

**Assumption 3** (First-Order Stability). The expected complete-data log-likelihood $Q_g(\boldsymbol{\theta}_g|\boldsymbol{\theta}'_g)$ at client $g$ obeys the FOS condition with parameter $\beta_g$, such that $0 \leqslant \beta_g < \lambda_g$ for all $g \in [G]$, where $\lambda_g$ is the strong concavity parameter of $Q_g(\boldsymbol{\theta}_g|\boldsymbol{\theta}'_g)$.

**Assumption 4** (Continuity). The local population and finite-sample $Q_g(\boldsymbol{\theta}_g|\boldsymbol{\theta}'_g)$ and $\widehat{Q}_g(\boldsymbol{\theta}_g|\boldsymbol{\theta}'_g)$, respectively, at client $g$ are continuous in both of their arguments.

**Assumption 5** (Likelihood Boundedness). The local population and finite-sample true log-likelihood functions $\mathcal{L}_g^*(\boldsymbol{\theta}_g)$ and $\widehat{\mathcal{L}}_g^*(\boldsymbol{\theta}_g)$, respectively, at client $g$ are bounded from above.

**Assumption 6** (Finite-Sample and Population M-Step Proximity). Let $\mathbb{A} = \prod_{k_g=1}^{K_g} \mathbb{B}_2(\boldsymbol{\theta}_{k_g}^*, a_g)$, and $\epsilon_g^{\text{unif}}(N_g, \delta_g) \leqslant (1 - \frac{\beta_g}{\lambda_g}) a_g$ be some constant. Then, with probability (w.p.) at least $(1 - \delta_g)$,

$$\sup_{\boldsymbol{\theta}'_g \in \mathbb{A}} ||\widehat{M}_{k_g}(\boldsymbol{\theta}'_g) - M_{k_g}(\boldsymbol{\theta}'_g)||_2 \leqslant \epsilon_g^{\text{unif}}(N_g, \delta_g),$$

where $M_{k_g}(\boldsymbol{\theta}'_g)$ is the M-step map associated with the population $Q_g(\boldsymbol{\theta}_g|\boldsymbol{\theta}'_g)$.

*Remark* 2. Assumptions 3 - 6 are standard assumptions that are commonly utilized in works focused on EM algorithms such as (Balakrishnan et al., 2014; Yan et al., 2017; Marfoq et al., 2021), and are verifiable for isotropic GMMs as we show later.

As shown by Balakrishnan et al. (2014), if Assumption 6 holds and the population M-step iterates converge as described in Appendix B.1, then the finite-sample M-step iterates converge to a neighborhood of the true component parameters $\boldsymbol{\theta}_{k_g}^*$ w.p. at least $(1 - \delta_g)$. We express this mathematically next, followed by Theorem 1 asserting convergence to a single point rather than oscillating.

$$||\widehat{M}_{k_g}(\boldsymbol{\theta}_g^{(t-1)}) - \boldsymbol{\theta}_{k_g}^*||_2 \leqslant \frac{\beta_g}{\lambda_g}||\boldsymbol{\theta}_{k_g}^{(t-1)} - \boldsymbol{\theta}_{k_g}^*||_2 + \frac{1}{1 - \frac{\beta_g}{\lambda_g}}\epsilon_g^{\text{unif}}(N_g, \delta_g) \quad \forall \boldsymbol{\theta}_{k_g}^{(t-1)} \in \mathbb{B}_2(\boldsymbol{\theta}_{k_g}^*; a_g), \quad (5)$$

where $a_g$ is the radius of the contractive region associated with the population EM iterates.

**Theorem 1** (Single-Point EM Convergence). *Suppose Assumptions 1 through 6 hold, and that $\boldsymbol{\theta}_{k_g}^{(t-1)} \in \mathbb{B}_2(\boldsymbol{\theta}_{k_g}^*, a_g)$. Then the finite sample EM iterates $\widehat{M}_{k_g}(\boldsymbol{\theta}_{k_g}^{(t-1)})$ converge to a single point within radius $\frac{1}{1 - \frac{\beta_g}{\lambda_g}}\epsilon_g^{\text{unif}}(N_g, \delta_g)$ from the ground truth parameters $\boldsymbol{\theta}_{k_g}^*$. (Proof in Appendix C.2).*

Now, consider a local finite-sample GEM algorithm whose update during each iteration $t$ is any $\widehat{m}_{k_g}(\boldsymbol{\theta}_g^{(t-1)}) \in \mathbb{B}_2(\widehat{M}_{k_g}(\boldsymbol{\theta}_g^{(t-1)}); \sqrt{\varepsilon_{k_g}^{(t)}})$, where the radius $\sqrt{\varepsilon_{k_g}^{(t)}}$ is obtained by solving the problem in (3). We show in Theorem 2 next that this algorithm exhibits very similar convergence behavior to that shown in (5). Subsequently, we show in Theorem 3 that our proposed $\mathtt{FedGEM}$ algorithm infers the *true* global number of clusters $K$ with a certain probability.

**Theorem 2** (Local Convergence of Finite-Sample GEM). *Suppose Assumptions 1 through 6 hold. Consider a GEM algorithm whose iterate $\widehat{m}_{k_g}(\theta_g^{(t-1)})$ at iteration $t$ is such that $\widehat{m}_{k_g}(\boldsymbol{\theta}_g^{(t-1)}) \in \mathbb{B}_2(\widehat{M}_{k_g}(\boldsymbol{\theta}_g^{(t-1)}); \sqrt{\varepsilon_{k_g}^{(t)}})$, where the radius $\sqrt{\varepsilon_{k_g}^{(t)}}$ is obtained by solving the problem in (3). Then, this algorithm converges to a neighborhood of the ground truth parameters $\boldsymbol{\theta}_{k_g}^*$ as follows:*

$$||\widehat{m}_{k_g}(\boldsymbol{\theta}_g^{(t-1)}) - \boldsymbol{\theta}_{k_g}^*||_2 \leqslant \frac{\beta_g}{\lambda_g}||\boldsymbol{\theta}_{k_g}^{(t-1)} - \boldsymbol{\theta}_{k_g}^*||_2 + \frac{1}{1 - \frac{\beta_g}{\lambda_g}}\epsilon_g^{\text{unif}}(N_g, \delta_g) + \widehat{\epsilon}(t) \quad \forall \boldsymbol{\theta}_{k_g}^{(t-1)} \in \mathbb{B}_2(\boldsymbol{\theta}_{k_g}^*; a_g),$$

*w.p. at least $1 - \delta_g$, with $\widehat{\epsilon}(t) = ||\widehat{M}_{k_g}(\boldsymbol{\theta}_g^{(t-1)}) - \boldsymbol{\theta}_{k_g}^{(t-1)}||_2 \to 0$ as $t \to \infty$. (Proof in Appendix C.3).*

**Theorem 3** (Number of Clusters Inference). *Suppose that all the assumptions associated with Theorem 2 hold, and that $||\widehat{M}_{k_g}(\boldsymbol{\theta}_g^{(t-1)}) - \boldsymbol{\theta}_{k_g}^{(t-1)}||_2$ diminishes to 0 at a sufficiently fast rate. Suppose further that the final aggregation radius $\varepsilon_{k_g}^{final}$ at client $g$ is set such that $\varepsilon_{k_g}^{final} = \epsilon_g^{unif}(N_g, \delta_g)$, and such that $\max_{g \in [G], k_g \in [K_g]} \varepsilon_{k_g}^{final} \leqslant \frac{R_{min}}{4}$. Then, the final $\widehat{K}^*$ inferred by the $\mathtt{FedGEM}$ algorithm is equivalent to the true $K$ w.p. at least $\prod_{g=1}^{G} \prod_{k_g=1}^{K_g}(1 - \delta_g)$. (Proof in Appendix C.4).*

## 5   FEDGEM FOR MULTI-COMPONENT ISOTROPIC GMMS

**Model Setup.**   Now that we have introduced our FedGEM algorithm, and studied its convergence in a general sense, we examine it in the context of an isotropic GMM. More specifically, we consider the setting where each client $g$ data is governed by a local mixture model $GMM_g(\boldsymbol{x}) = \sum_{k_g=1}^{K_g} \pi_{k_g} \phi(\boldsymbol{x}|\boldsymbol{\theta}_{k_g}^*, I_d)$, where $\phi(\boldsymbol{x}|\boldsymbol{\theta}_{k_g}^*, I_d)$ is the Gaussian density with identity covariance. We denote the minimum and maximum component weights at client $g$ by $\pi_{\min_g}$ and $\pi_{\max_g}$, respectively. Moreover, we denote the ratio $\kappa_g = \frac{\pi_{\max_g}}{\pi_{\min_g}}$. The population and finite-sample M-steps associated with this model admit the following closed-form solutions.

$$\textbf{Population M-step:} \qquad M_{k_g}(\boldsymbol{\theta}_g') = \frac{\mathbb{E}_{\boldsymbol{x} \sim GMM_g(\boldsymbol{x})}\big[\gamma_{k_g}(\boldsymbol{x}, \boldsymbol{\theta}_g')\boldsymbol{x}\big]}{\mathbb{E}_{\boldsymbol{x} \sim GMM_g(\boldsymbol{x})}\big[\gamma_{k_g}(\boldsymbol{x}, \boldsymbol{\theta}_g')\big]} \qquad \forall k_g \in [K_g]. \qquad (6a)$$

$$\textbf{Finite-sample M-step:} \qquad \widehat{M}_{k_g}(\boldsymbol{\theta}_g') = \frac{\sum_{n_g=1}^{N_g} \gamma_{k_g}(\widehat{\boldsymbol{x}}_{n_g}, \boldsymbol{\theta}_g')\widehat{\boldsymbol{x}}_{n_g}}{\sum_{n_g=1}^{N_g} \gamma_{k_g}(\widehat{\boldsymbol{x}}_{n_g}, \boldsymbol{\theta}_g')} \qquad \forall k_g \in [K_g]. \qquad (6b)$$

Note that for the described model, the population and finite-sample expected complete-data log-likelihood functions $Q_g(\boldsymbol{\theta}_g|\boldsymbol{\theta}_g')$ and $\widehat{Q}_g(\boldsymbol{\theta}_g|\boldsymbol{\theta}_g')$, respectively, are strongly concave in $\boldsymbol{\theta}_g$ and continuous in both of their arguments. Moreover, if $\boldsymbol{\theta}_g' = \boldsymbol{\theta}_g^*$, the strong concavity parameter of $Q_g(\boldsymbol{\theta}_g|\boldsymbol{\theta}_g')$ is $\pi_{\min_g}$. Finally, the true population and finite-sample log-likelihood functions associated with this model are bounded from above due to the identity covariances and fixed weights for all components.

As shown in (6b), the finite-sample M-step associated with our model admits a closed form. Therefore, it remains to derive a tractable reformulation of the uncertainty set radius problem in (3). Next, we introduce Theorem 4, where we derive a bi-convex, 2-dimensional reformulation of the problem in (3). Additionally, we introduce a solution Algorithm 4 in Appendix B.2 to solve the problem, accompanied by Proposition 2 in Appendix B.2 asserting that the algorithm enjoys a low worst-case time complexity. Finally, we provide a **preliminary differential privacy discussion** for the finite-sample maximizers shared by the clients in Appendix B.4.

**Theorem 4** (Radius Problem Reformulation). *The semi-infinite optimization problem $J_{k_g}(\boldsymbol{\theta}_g')$ in (3) admits the following tractable, bi-convex, 2-dimensional reformulation for the isotropic GMM described in this section. (Proof in Appendix C.5).*

$$J_{k_g}(\boldsymbol{\theta}_g') = \qquad\qquad\qquad\qquad\qquad\qquad\qquad\qquad\qquad\qquad\qquad\qquad (7)$$

$$\max_{\varepsilon_{k_g}, \alpha_{k_g} \in \mathbb{R}} \quad \varepsilon_{k_g}$$

$$\text{s. t.} \qquad \varepsilon_{k_g}\alpha_{k_g}^2 + \left[\sum_{n_g=1}^{N_g} \gamma_{k_g}(\widehat{\boldsymbol{x}}_{n_g}, \boldsymbol{\theta}_g')\left(||\widehat{\boldsymbol{x}}_{n_g} - \widehat{M}_{k_g}(\boldsymbol{\theta}_g')||_2^2 - ||\widehat{\boldsymbol{x}}_{n_g} - \boldsymbol{\theta}_{k_g}'||_2^2 - \varepsilon_{k_g}\right)\right]\alpha_{k_g} +$$

$$\left(\sum_{n_g=1}^{N_g} \gamma_{k_g}(\widehat{\boldsymbol{x}}_{n_g}, \boldsymbol{\theta}_g')\right) \sum_{n_g=1}^{N_g} \gamma_{k_g}(\widehat{\boldsymbol{x}}_{n_g}, \boldsymbol{\theta}_g')||\widehat{\boldsymbol{x}}_{n_g} - \boldsymbol{\theta}_{k_g}'||_2^2 \leqslant 0$$

$$\alpha_{k_g} \geqslant \sum_{n_g=1}^{N_g} \gamma_{k_g}(\widehat{\boldsymbol{x}}_{n_g}, \boldsymbol{\theta}_g').$$

**Convergence Analysis.**   To guarantee the convergence of our FedGEM algorithm for the multi-component isotropic GMM, we verify three key properties. Namely, we present Theorems 6, 7, and 8 in Appendix B.3 to establish the FOS property of $Q_g(\boldsymbol{\theta}_g|\boldsymbol{\theta}_g')$, study the conditions under which $M_{k_g}(\boldsymbol{\theta}_g)$ is contractive, and establish a probabilistic upper bound on the distance between the population $M_{k_g}(\boldsymbol{\theta}_g)$ and the finite-sample $\widehat{M}_{k_g}(\boldsymbol{\theta}_g)$, respectively. Consequently, the convergence of our algorithm for isotropic GMMs follows from these results. However, these results require the clusters to be *well-separated*.

# 6  NUMERICAL EXPERIMENTS

We present a comprehensive set of numerical experiments that benchmark the performance of our proposed `FedGEM` algorithm against leading state-of-the-art federated clustering methods. All numbers reported in this section are averaged over 50 repetitions. Confidence intervals and error bars represent one standard deviation. Randomness in repetitions arises from initialization, cluster assignments, and data shuffling/generation. We assign equal $\pi_{k_g}$ for all $k_g \in [K_g]$ at client $g$. Moreover, we weigh each client $g$ by its sample count $N_g$, and we set the final aggregation radius equivalently for all clients. In all experiments, clusters are assigned randomly to the clients, whereby each client $g$ has $K_g$ clusters such that $2 \leqslant K_g < K$. More experimental details and results are provided in Appendix D. Additionally, a **Scalability Study** is provided in Appendix E, demonstrating that our algorithm scales exceptionally well with problem size compared to relevant benchmarks. All code associated with this paper is available at `https://github.com/mibrahim41/FedGEM`.

**Our Method.** Our method is the isotropic GMM from Section 5 trained via our `FedGEM` algorithm.

**Evaluation Metrics.** We utilize the Silhouette Score (SS) (Rousseeuw, 1987) for hyperparameter tuning as it does not require label knowledge. However, we mainly rely on the Adjusted Rand Index (ARI) (Hubert & Arabie, 1985) to evaluate model performance as it is robust to cluster shape and size unlike other metrics. We report experimental results in SS in Appendix D.4.

**Hyperparameters.** We tune the final aggregation radius for `FedGEM` via cross validation. However, we do not directly tune the radius itself. Instead, we use the heuristic $\varepsilon_{k_g}^{\text{final}} = \frac{\upsilon_g \widehat{R}_{\min_g}}{\pi_g \sqrt{N_g}}$, where $\widehat{R}_{\min_g}$ is the minimum distance between any two estimated cluster centroids at client $g$, and $\upsilon_g$ is the hyperparameter we tune (set equivalently across all clients for simplicity). This heuristic allows the final aggregation radius to scale with the feature space and the number of samples available at each client. We provide a thorough discussion on hyperparameter tuning is provided in Appendix D.3. We set hyperparameters for the benchmark methods as described in their associated works.

## 6.1  BENCHMARKING

This study aims to compare the performance of our proposed method to that of various existing methods using an array of popular benchmark datasets.

**Datasets.** We use MNIST, Fashion MNIST (FMNIST), Extended MNIST (EMNIST), CIFAR-10, and 4 other datasets from the UCI repository. For all datasets, we use $70\%$ of the samples for training and the rest for testing, except for EMNIST where we use $50\%$ of the samples for training. We use $G = 100$ for MNIST, FMNIST, and CIFAR-10, $G = 25$ for EMNIST, and $G = 5$ for UCI datasets.

**Baselines.** We compare our method to 2 centralized and 5 federated clustering methods from the literature. The centralized ones are a GMM and a DP-GMM (Antoniak, 1974). The federated methods are k-FED (Dennis et al., 2021), FFCM-avg1 and FFCM-avg2 (Stallmann & Wilbik, 2022), `FedKmeans` (Garst & Reinders, 2024), and AFCL (Zhang et al., 2025). Note that DP-GMM and AFCL are the only benchmarks that do not assume prior knowledge of $K$.

**Results.** The ARI attained by all models and the estimated number of clusters estimated by models with unknown $K$ are shown in Tables 1 and 2, respectively. We observe that our method consistently outperforms AFCL for all datasets, which is the only other federated clustering model with unknown $K$. Additionally, our model also consistently outperforms DP-GMM, which can be attributed to its more accurate number of cluster estimation. Another key observation is that our method even outperforms some clustering algorithms with known $K$ in various datasets. This result underscores the significant practical impact of our proposed method, which does not require prior knowledge of $K$. We highlight that similar trends are observed when performance is evaluated via the SS as shown in Appendix D.4, emphasizing the impact of our model. Despite our model's strong performance, we also observe that it largely overestimates the number of clusters in datasets such as CIFAR-10, Frog A, and Frog B. While its estimate is the best achieved out of the models compared, this can potentially be further improved in future work by examining more complex mixture models. Finally, we note that the datasets used are verifiably **non-Gaussian** via a Henze-Zirkler multivariate normality test, and likely include cluster overlaps. This demonstrates that our model can perform well in practice even when assumptions are violated.

Table 1: ARI attained by all methods on tested datasets. (**Bold** = best, underline = second best.)

| Model | Known $K$? | MNIST | FMNIST | EMNIST | CIFAR-10 | Abalone | Frog A | Frog B | Waveform |
|---|---|---|---|---|---|---|---|---|---|
| GMM (central) | Yes | .287 ±.067 | .385 ±.023 | .235 ±.010 | .402 ±.022 | .096 ±.028 | .447 ±.097 | .448 ±.172 | .262 ±.016 |
| k-FED | Yes | .354 ±.082 | .288 ±.101 | .223 ±.031 | .358 ±.043 | .100 ±.030 | .617 ±.144 | .467 ±.148 | **.277** ±.061 |
| FFCM-avg1 | Yes | .148 ±.031 | .164 ±.030 | .025 ±.007 | .312 ±.035 | .096 ±.029 | .470 ±.094 | .442 ±.112 | .254 ±.029 |
| FFCM-avg2 | Yes | .336 ±.053 | .352 ±.038 | .114 ±.011 | **.513** ±.028 | **.102** ±.032 | **.720** ±.149 | **.645** ±.117 | .268 ±.057 |
| FedKmeans | Yes | **.640** ±.035 | **.449** ±.027 | **.285** ±.007 | .437 ±.024 | .098 ±.033 | .546 ±.143 | .492 ±.118 | .260 ±.015 |
| DP-GMM (central) | No | .115 ±.021 | .179 ±.011 | .120 ±.015 | .068 ±.003 | .075 ±.023 | .326 ±.090 | .223 ±.065 | .255 ±.015 |
| AFCL | No | .038 ±.002 | .035 ±.002 | .089 ±.005 | .034 ±.002 | .062 ±.020 | .344 ±.108 | .272 ±.079 | .157 ±.036 |
| FedGEM (**ours**) | No | **.452** ±.049 | **.287** ±.057 | **.285** ±.022 | **.286** ±.033 | **.138** ±.056 | **.552** ±.129 | **.468** ±.117 | **.335** ±.078 |

Table 2: Estimated number of clusters for algorithms with unknown $K$.

| Model | MNIST | FMNIST | EMNIST | CIFAR-10 | Abalone | Frog A | Frog B | Waveform |
|---|---|---|---|---|---|---|---|---|
| True $K$ | 10 | 10 | 47 | 10 | 7 | 10 | 8 | 3 |
| DP-GMM (central) | 110.20 ±6.09 | 86.34 ±5.61 | 247.40 ±23.46 | 364.03 ±12.57 | 15.46 ±2.20 | 29.46 ±5.49 | 24.38 ±4.24 | 6.84 ±0.65 |
| AFCL | 501.82 ±18.00 | 501.37 ±22.51 | 575.39 ±55.63 | 502.97 ±19.70 | 25.00 ±4.43 | 28.42 ±6.42 | 23.96 ±4.26 | 12.52 ±1.11 |
| FedGEM (**ours**) | **13.63** ±2.29 | **17.59** ±6.61 | **58.67** ±5.23 | **37.72** ±13.60 | **12.14** ±3.65 | **23.94** ±9.99 | **20.00** ±7.07 | **4.42** ±1.40 |

## 6.2 SENSITIVITY

This study evaluates the performance of our model as $R_{\min}$ changes. This includes *non-well-separated* settings, which violate the convergence conditions of the GMM in Section 5. We also examine the sensitivity of our algorithm to its hyperparameter in Appendix D.6.

**Dataset.** The data used for this experiment is isotropic Gaussian clusters generated via the `make_blobs` module in Python. We control $R_{\min}$, requiring that the centers of at least two clusters in each dataset be $R_{\min}$ apart. Moreover, we study three key settings: i) nominal: data is balanced across clients and clusters, ii) client imbalance: the data distribution across clients is $[40\%, 24\%, 16\%, 16\%, 4\%]$, and iii) cluster imbalance: the local data for each client is randomly distributed across the local clusters. For all settings we use $G = 5$, $N_{\text{train}} = 2500$, and $N_{\text{test}} = 5000$.

**Baseline.** We compare our model to a centralized GMM trained via EM as the latter represents a strong benchmark. This allows us to quantify the effects of our model's federation and unknown $K$.

**Result.** Figure 1 illustrates that the performance for both models improves as $R_{\min}$ increases. Notably, our proposed model achieves very close performance to the centralized GMM, and even outperforms it with $R_{\min} \in \{1, 2\}$. This can be attributed to cluster heterogeneity across clients. That is, each client only has a subset of the total clusters, so each client's clustering problem is potentially easier than the centralized problem. This can cause each client to perform better individually than the centralized model, which is trained on all clusters. Indeed, the benefits gained from cluster heterogeneity in clustering problems were first observed by Dennis et al. (2021).

It is worth noting that some of the settings explored in this study involve **overlapping clusters**, violating the *well-separated* cluster assumption required for convergence. However, our model still achieves very competitive performance. This suggests that even when some assumptions are violated, our algorithm can still converge in practice to a well-performing model. Finally, we observe that our model's estimate $\widehat{K}^*$ across all experimental settings is very close to the true number of

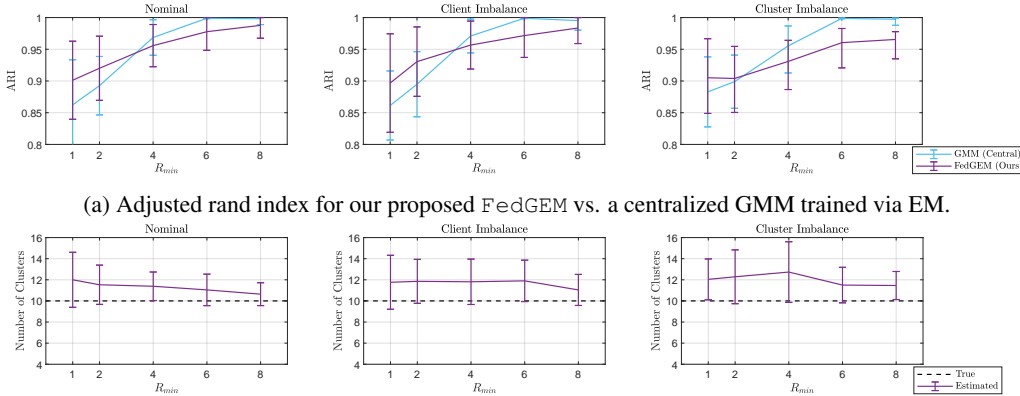

(a) Adjusted rand index for our proposed `FedGEM` vs. a centralized GMM trained via EM.

(b) Number of clusters estimated by our proposed `FedGEM` vs. the true number of clusters.

Figure 1: Results of the sensitivity study.

clusters $K$. While it tends to overestimate slightly in most settings, performance could potentially be further improved through better tuning of the final aggregation radius hyperparameter.

## 7    CONCLUSIONS

We introduce `FedGEM`: a federated GEM algorithm for training mixture models with an unknown number of components, geared towards federated clustering for clients whose local cluster sets are heterogeneous but potentially overlapping. Our algorithm requires clients to perform local EM steps, and compute an uncertainty set centered at the maximizer corresponding to each component. These uncertainty sets are then shared with the server. The server leverages uncertainty set intersections to infer overlap between clients' clusters, allowing it to perform model parameter aggregation and to estimate the total number of unique clusters. We study theoretical aspects of our algorithm, where we prove probabilistic convergence under standard assumptions. Subsequently, we study our algorithm in the context of isotropic GMMs. To that end, we derive a tractable and bi-convex reformulation of the problem used by each client to obtain the uncertainty sets, and we verify key assumptions required to prove convergence. We empirically demonstrate that our proposed algorithm outperforms existing ones through a series of numerical experiments utilizing synthetic and popular datasets. We provide a thorough discussion on limitations and future work in Appendix F.3.

### ACKNOWLEDGMENTS

This work was funded by the National Aeronautics and Space Administration (NASA), Space Technology Research Institute (STRI) Habitats Optimized for Missions of Exploration (HOME) 'SmartHab' Project (grant No. 80NSSC19K1052). Weijun Xie was supported in part by National Science Foundation grant 2246414, and by the Office of Naval Research grant N00014-24-1-2066.

We thank the reviewers and the AC for their valuable input, which allowed us to improve the presentation and content of our paper.

### ETHICS STATEMENT

All software and datasets utilized in this work are used under proper licenses, as detailed in Appendix D. We do not release any data as part of our submission, and we provide full references to all datasets used.

### REPRODUCIBILITY STATEMENT

We have taken various steps to ensure the ease of reproducibility of both the theoretical and experimental aspects of this paper. On the theoretical side, we have provided full formal and complete proofs for all theoretical results, as well as all the required assumptions and a full description of

the problem setting. More specifically, the detailed description of the problem setting is provided in Section 3, whereas the required assumptions are provided in throughout the main body of the paper. Additionally, supplementary theoretical results along with their formal proofs are provided in Appendix B. The proofs of all theoretical results presented in the main body of the paper are provided in Appendix C. Finally, detailed explanations and interpretations of all assumptions and theoretical results are provided in Appendix F. On the experimental side, we have provided a summarized description of our experimental settings and results in Section 6, whereas we provided full detail on all aspects of the experiments as well as supplementary results in Appendices D and E. This includes all software and hardware details, all dataset license information and preprocessing details, as well as hyperparameter details. Finally, we have made all the code used to run our experiments along with detailed instructions available at `https://github.com/mibrahim41/FedGEM`.

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

# A Federated Generalized Expectation-Maximization Algorithm for Mixture Models with an Unknown Number of Components (Appendices)

## Contents

# A SUPPLEMENTARY PSEUDO-CODE

In this section we provide detailed pseudo-code for our `FedGEM` algorithm, as well as detailed pseudo-code for the server computations both in the collaborative training and final aggregation phases.

## A.1 DETAILED FEDGEM PSEUDO-CODE

---

**Algorithm 1** `FedGEM`

---

**Input**: Number of communication rounds $T$, Number of local steps $S_g$, Final aggregation radius $\varepsilon_{k_g}^{\text{final}}$ and fixed weights $\pi_{k_g} \ \forall k_g \in [K_g], \forall g \in [G]$

**Output**: Final $\boldsymbol{\theta}_{k_g}^{\text{final}}$ for all components $k_g \in [K_g]$ and clients $g \in [G]$, inferred $\widehat{K}^*$

1:   *INITIALIZATION*
2: **for** clients $g = 1, \ldots, G$ in parallel **do**
3:     Initialize $\boldsymbol{\theta}_{k_g}^{(0)}$ for all $k_g \in [K_g]$ via `k-means++`.
4: **end for**
5: *COLLABORATIVE TRAINING*
6: **for** round $t = 0, \ldots, T$ **do**
7:     *Clients*
8:     **for** clients $g = 1, \ldots, G$ in parallel **do**
9:         $\boldsymbol{\theta}_{k_g}^{(t-1,0)} \leftarrow \boldsymbol{\theta}_{k_g}^{(t-1)}$ for all $k_g \in [K_g]$
10:        **for** step $s_g = 1, \ldots, S_g$ **do**
11:           Compute $\gamma_{k_g}(\widehat{\boldsymbol{x}}_{n_g}, \boldsymbol{\theta}_{k_g}^{(t-1,s_g-1)})$ via E-step in (1) for all $k_g \in [K_g]$ and samples $n_g \in [N_g]$.
12:           Compute $\widehat{M}_{k_g}(\boldsymbol{\theta}_{k_g}^{(t-1,s_g-1)})$ via M-step in (2) for all $k_g \in [K_g]$.
13:           Update $\boldsymbol{\theta}_{k_g}^{(t,s_g)} \leftarrow \widehat{M}_{k_g}(\boldsymbol{\theta}_{k_g}^{(t-1,s_g-1)})$ for all $k_g \in [K_g]$.
14:        **end for**
15:        Solve for $\varepsilon_{k_g}^{(t)} \leftarrow \arg\max_{\varepsilon_{k_g} \in \mathbb{R}} J_{k_g}(\boldsymbol{\theta}_{k_g}^{(t,S_g)})$ via problem in (3) for all $k_g \in [K_g]$.
16:        **if** $t < T$ **then**
17:          Transmit tuple $(\boldsymbol{\theta}_{k_g}^{(t,S_g)}, \varepsilon_{k_g}^{(t)})$ for all $k_g \in [K_g]$ to central server.
18:        **else**
19:          Transmit tuple $(\boldsymbol{\theta}_{k_g}^{(t,S_g)}, \varepsilon_{k_g}^{\text{final}})$ for all $k_g \in [K_g]$ to central server.
20:        **end if**
21:     **end for**
22:     *Server*
23:     **if** t < T **then**
24:        Update $(\boldsymbol{\theta}_{k_g}^{(t)}, \widehat{K}^{(t)}) \leftarrow$ `server_update`$(\boldsymbol{\theta}_{k_g}^{(t,S_g)}, \varepsilon_{k_g}^{(t)})$ for all $k_g \in [K_g]$ and $g \in [G]$ via Algorithm 2 in Appendix A.2.
25:        Transmit $\boldsymbol{\theta}_{k_g}^{(t+1)}$ to clients for all $k_g \in [K_g]$ and and $g \in [G]$.
26:     **else**
27:        *FINAL AGGREGATION*
28:        Compute $(\boldsymbol{\theta}_{k_g}^{\text{final}}, \widehat{K}^*) \leftarrow$ `server_final_aggregation`$(\boldsymbol{\theta}_{k_g}^{(t,S_g)}, \varepsilon_{k_g}^{(t)})$ for all $k_g \in [K_g]$ and $g \in [G]$ via Algorithm 3 in Appendix A.2.
29:        Transmit $\boldsymbol{\theta}_{k_g}^{\text{final}}$ to clients for all $k_g \in [K_g]$ and and $g \in [G]$.
30:     **end if**
31: **end for**

---

A.2 DETAILED SERVER COMPUTATIONS PSEUDO-CODE

---

**Algorithm 2** `server_update`$(\boldsymbol{\theta}_{k_g}, \varepsilon_{k_g})$

---

**Input**: $\boldsymbol{\theta}_{k_g}$ and $\varepsilon_{k_g}$ for all clients $g \in [G]$ and components $k_g \in [K_g]$ at the $g^{th}$ client
**Output**: Updated $\boldsymbol{\theta}'_{k_g}$ for all clients $g \in [G]$ and components $k_g \in [K_g]$ at client $g$, $\widehat{K}^*$

1: Initialize $\widehat{K} = 0$.
2: Initialize set $\mathcal{T}_{k_g}$ containing only $\boldsymbol{\theta}_{k_g}$ for each $g \in [G]$ and $k_g \in [K_g]$.
3: Initialize $\text{comp}(g, k_g).\text{assigned} \leftarrow \text{False}$ for all $g \in [G], k_g \in [K_g]$.
4: Initialize $\text{comp}(g, k_g).\text{supercluster} \leftarrow \text{Null}$ for all $g \in [G], k_g \in [K_g]$.
5: **for** client $g_1 = 1, \ldots, G$ **do**
6:     **for** component $k_{g_1} = 1, \ldots, K_{g_1}$ **do**
7:         **for** client $g_2 = g_1, \ldots, G$ **do**
8:             **for** component $k_{g_2} = 1, \ldots, K_{g_2}$ **do**
9:             **if** $||\widehat{M}_{k_g}(\boldsymbol{\theta}'_g) - \widehat{M}_{k_{g'}}(\boldsymbol{\theta}'_{g'})||_2 \leqslant \sqrt{\varepsilon_{k_g}} + \sqrt{\varepsilon_{k_{g'}}}$ **then**
10:                 $\boldsymbol{\nu}^* \leftarrow \widehat{M}_{k_g}(\boldsymbol{\theta}'_g) + \text{clip}\left(0.5, 1 - \frac{\sqrt{\varepsilon_{k_{g'}}}}{w}, \frac{\sqrt{\varepsilon_{k_g}}}{w}\right)\left(\widehat{M}_{k_g}(\boldsymbol{\theta}'_g) - \widehat{M}_{k_{g'}}(\boldsymbol{\theta}'_{g'})\right)$.
11:                 $\mathcal{T}_{k_{g_1}} \leftarrow \mathcal{T}_{k_{g_1}} \cup \boldsymbol{\nu}^*$.
12:                 $\mathcal{T}_{k_{g_2}} \leftarrow \mathcal{T}_{k_{g_2}} \cup \boldsymbol{\nu}^*$.
13:                 **if** $\text{comp}(g_1, k_{g_1}).\text{assigned} = \text{False} \And \text{comp}(g_2, k_{g_2}).\text{assigned} = \text{True}$ **then**
14:                     $\text{comp}(g_1, k_{g_1}).\text{supercluster} \leftarrow \text{comp}(g_2, k_{g_2}).\text{supercluster}$.
15:                     $\text{comp}(g_1, k_{g_1}).\text{assigned} \leftarrow \text{True}$.
16:                 **else if** $\text{comp}(g_1, k_{g_1}).\text{assigned} = \text{True} \And \text{comp}(g_2, k_{g_2}).\text{assigned} = \text{False}$ **then**
17:                     $\text{comp}(g_2, k_{g_2}).\text{supercluster} \leftarrow \text{comp}(g_1, k_{g_1}).\text{supercluster}$.
18:                     $\text{comp}(g_2, k_{g_2}).\text{assigned} \leftarrow \text{True}$.
19:                 **else if** $\text{comp}(g_1, k_{g_1}).\text{assigned} = \text{True} \And \text{comp}(g_2, k_{g_2}).\text{assigned} = \text{True}$ **then**
20:                     **if** $\text{comp}(g_1, k_{g_1}).\text{supercluster} \mathrel{!=} \text{comp}(g_2, k_{g_2}).\text{supercluster}$ **then**
21:                       $\text{comp}(g', k_{g'}).\text{supercluster} \leftarrow \text{comp}(g_1, k_{g_1}).\text{supercluster} \; \forall k_{g'}$ such that $\text{comp}(g', k_{g'}).\text{supercluster} = \text{comp}(g_2, k_{g_2}).\text{supercluster}$.
22:                     $\widehat{K} \leftarrow \widehat{K} - 1$
23:                     Reorganize supercluster numbers for all components.
24:                   **end if**
25:                 **else if** $\text{comp}(g_1, k_{g_1}).\text{assigned} = \text{False} \And \text{comp}(g_2, k_{g_2}).\text{assigned} = \text{False}$ **then**
26:                   $\widehat{K} \leftarrow \widehat{K} + 1$.
27:                   $\text{comp}(g_1, k_{g_1}).\text{supercluster} \leftarrow \widehat{K}$.
28:                   $\text{comp}(g_2, k_{g_2}).\text{supercluster} \leftarrow \widehat{K}$.
29:                   $\text{comp}(g_1, k_{g_1}).\text{assigned} \leftarrow \text{True}$.
30:                   $\text{comp}(g_2, k_{g_2}).\text{assigned} \leftarrow \text{True}$.
31:                 **end if**
32:             **end if**
33:         **end for**
34:         **end for**
35:     **if** $\text{comp}(g_1, k_{g_1}).\text{assigned} = \text{False}$ **then**
36:         $\widehat{K} \leftarrow \widehat{K} + 1$.
37:         $\text{comp}(g_1, k_{g_1}).\text{supercluster} \leftarrow \widehat{K}$.
38:         $\text{comp}(g_1, k_{g_1}).\text{assigned} \leftarrow \text{True}$.
39:     **end if**
40:     **end for**
41: **end for**
42: **for** client $g = 1, \ldots, G$ **do**
43:     **for** component $k_g = 1, \ldots, K_g$ **do**
44:         $\boldsymbol{\theta}'_{k_g} \leftarrow$ aggregate of elements in $\mathcal{T}_{k_g}$.
45:     **end for**
46: **end for**
47: $\widehat{K}^* \leftarrow \widehat{K}$

---

---

**Algorithm 3** server_final_aggregation($\boldsymbol{\theta}_{k_g}, \varepsilon_{k_g}^{\text{final}}$)

---

**Input**: $\boldsymbol{\theta}_{k_g}$ and $\varepsilon_{k_g}^{\text{final}}$ for all clients $g \in [G]$ and components $k_g \in [K_g]$ at the $g^{th}$ client

**Output**: Final $\boldsymbol{\theta}_{k_g}^{\text{final}}$ for all clients $g \in [G]$ and components $k_g \in [K_g]$ at client $g$, $\widehat{K}^*$

1: Initialize $\widehat{K} = 0$.
2: Initialize set $\mathcal{T}_{k_g}$ containing only $\boldsymbol{\theta}_{k_g}$ for each $g \in [G]$ and $k_g \in [K_g]$.
3: Initialize comp$(g, k_g)$.assigned $\leftarrow$ False for all $g \in [G], k_g \in [K_g]$.
4: Initialize comp$(g, k_g)$.supercluster $\leftarrow$ Null for all $g \in [G], k_g \in [K_g]$.
5: **for** client $g_1 = 1, \ldots, G$ **do**
6:    **for** component $k_{g_1} = 1, \ldots, K_{g_1}$ **do**
7:       **for** client $g_2 = g_1, \ldots, G$ **do**
8:          **for** component $k_{g_2} = 1, \ldots, K_{g_2}$ **do**
9:             **if** $||\widehat{M}_{k_g}(\boldsymbol{\theta}'_g) - \widehat{M}_{k_{g'}}(\boldsymbol{\theta}'_{g'})||_2 \leqslant \sqrt{\varepsilon_{k_g}} + \sqrt{\varepsilon_{k_{g'}}}$ **then**
10:                **if** comp$(g_1, k_{g_1})$.assigned = False & comp$(g_2, k_{g_2})$.assigned = True **then**
11:                   comp$(g_1, k_{g_1})$.supercluster $\leftarrow$ comp$(g_2, k_{g_2})$.supercluster.
12:                   comp$(g_1, k_{g_1})$.assigned $\leftarrow$ True.
13:                   $\mathcal{T}_{k_{g_1}} \leftarrow \mathcal{T}_{k_{g_1}} \cup \mathcal{T}_{k_{g_2}}$.
14:                   $\mathcal{T}_{k_{g_2}} \leftarrow \mathcal{T}_{k_{g_2}} \cup \boldsymbol{\theta}_{k_{g_1}}$.
15:                **else if** comp$(g_1, k_{g_1})$.assigned = True & comp$(g_2, k_{g_2})$.assigned = False
               **then**
16:                   comp$(g_2, k_{g_2})$.supercluster $\leftarrow$ comp$(g_1, k_{g_1})$.supercluster.
17:                   comp$(g_2, k_{g_2})$.assigned $\leftarrow$ True.
18:                   $\mathcal{T}_{k_{g_2}} \leftarrow \mathcal{T}_{k_{g_2}} \cup \mathcal{T}_{k_{g_1}}$.
19:                   $\mathcal{T}_{k_{g_1}} \leftarrow \mathcal{T}_{k_{g_1}} \cup \boldsymbol{\theta}_{k_{g_2}}$.
20:                **else if** comp$(g_1, k_{g_1})$.assigned = True & comp$(g_2, k_{g_2})$.assigned = True
               **then**
21:                   **if** comp$(g_1, k_{g_1})$.supercluster != comp$(g_2, k_{g_2})$.supercluster **then**
22:                      $\mathcal{T}_{\text{temp},1} \leftarrow \mathcal{T}_{k_{g_1}}$.
23:                      $\mathcal{T}_{\text{temp},2} \leftarrow \mathcal{T}_{k_{g_2}}$.
24:                      $\mathcal{T}_{k_{g'}} \leftarrow \mathcal{T}_{k_{g'}} \cup \mathcal{T}_{\text{temp},1}$    $\forall k_{g'}$ such that comp$(g', k_{g'})$.supercluster = comp$(g_2, k_{g_2})$.supercluster.
25:                      $\mathcal{T}_{k_{g'}} \leftarrow \mathcal{T}_{k_{g'}} \cup \mathcal{T}_{\text{temp},2}$    $\forall k_{g'}$ such that comp$(g', k_{g'})$.supercluster = comp$(g_1, k_{g_1})$.supercluster.
26:                      comp$(g', k_{g'})$.supercluster $\leftarrow$ comp$(g_1, k_{g_1})$.supercluster $\forall k_{g'}$ such that comp$(g', k_{g'})$.supercluster = comp$(g_2, k_{g_2})$.supercluster.
27:                      $\widehat{K} \leftarrow \widehat{K} - 1$
28:                      Reorganize supercluster numbers for all components.
29:                   **end if**
30:                **else if** comp$(g_1, k_{g_1})$.assigned = False & comp$(g_2, k_{g_2})$.assigned = False
               **then**
31:                   $\widehat{K} \leftarrow \widehat{K} + 1$.
32:                   comp$(g_1, k_{g_1})$.supercluster $\leftarrow \widehat{K}$.
33:                   comp$(g_2, k_{g_2})$.supercluster $\leftarrow \widehat{K}$.
34:                   comp$(g_1, k_{g_1})$.assigned $\leftarrow$ True.
35:                   comp$(g_2, k_{g_2})$.assigned $\leftarrow$ True.
36:                   $\mathcal{T}_{k_{g_1}} \leftarrow \mathcal{T}_{k_{g_1}} \cup \boldsymbol{\theta}_{k_{g_2}}$.
37:                   $\mathcal{T}_{k_{g_2}} \leftarrow \mathcal{T}_{k_{g_2}} \cup \boldsymbol{\theta}_{k_{g_1}}$.
38:                **end if**
39:             **end if**
40:          **end for**
41:       **end for**
42:    **if** comp$(g_1, k_{g_1})$.assigned = False **then**
43:       $\widehat{K} \leftarrow \widehat{K} + 1$.
44:       comp$(g_1, k_{g_1})$.supercluster $\leftarrow \widehat{K}$.
45:       comp$(g_1, k_{g_1})$.assigned $\leftarrow$ True.
46:    **end if**
47:    **end for**
48: **end for**
49: **for** client $g = 1, \ldots, G$ **do**
50:    **for** component $k_g = 1, \ldots, K_g$ **do**
51:       $\boldsymbol{\theta}_{k_g}^{\text{final}} \leftarrow$ aggregate of elements in $\mathcal{T}_{k_g}$.
52:    **end for**
53: **end for**
54: $\widehat{K}^* \leftarrow \widehat{K}$

---

## B    SUPPLEMENTARY THEORETICAL RESULTS AND ANALYSIS

### B.1    POPULATION CONVERGENCE ANALYSIS FOR FEDGEM ALGORITHM

In this section we study the convergence behavior of our proposed algorithm in the population setting. The population convergence of our algorithm relies on the convergence of the local EM algorithm at each client $g$ to the likelihood maximizers $\boldsymbol{\theta}_{k_g}^*$ for all $k_g \in [K_g]$. As discussed by Balakrishnan et al. (2014), this requires that the local $Q_g(\boldsymbol{\theta}_g|\boldsymbol{\theta}_g')$ to satisfy the first-order stability (FOS) condition defined in 1. Indeed, if Assumption 3 holds, then Balakrishnan et al. (2014) prove that the population EM algorithm at client $g$ converges to the ground truth parameters $\boldsymbol{\theta}_{k_g}^*$ for component $k_g$ geometrically as follows:

$$||M_{k_g}(\boldsymbol{\theta}_g^{(t-1)}) - \boldsymbol{\theta}_{k_g}^*||_2 \leqslant \frac{\beta_g}{\lambda_g}||\boldsymbol{\theta}_{k_g}^{(t-1)} - \boldsymbol{\theta}_{k_g}^*||_2 \qquad \forall \boldsymbol{\theta}_{k_g}^{(t-1)} \in \mathbb{B}_2(\boldsymbol{\theta}_{k_g}^*; a_g), \tag{8}$$

where $a_g$ is the radius of the contraction region for all components $k_g$ located at client $g$, and $M_{k_g}(\cdot)$ is the population M-step map defined next

$$M_{k_g}(\boldsymbol{\theta}_g') := \underset{\boldsymbol{\theta}_{k_g} \in \mathbb{R}^d}{\arg\max} \mathbb{E}_{\boldsymbol{x} \sim \mathcal{M}_g(\boldsymbol{x})} \gamma_{k_g}(\boldsymbol{x}, \boldsymbol{\theta}_g') \log(\pi_{k_g} p_{k_g}(\boldsymbol{x}|\boldsymbol{\theta}_{k_g})) \quad \forall k_g \in [K_g]$$

Now, consider a local GEM algorithm whose update during each iteration $t$ is any $m_{k_g}(\boldsymbol{\theta}_g^{(t-1)}) \in \mathbb{B}_2(M_{k_g}(\boldsymbol{\theta}_g^{(t-1)}); \sqrt{\varepsilon_{k_g}^{(t)}})$, where the radius $\sqrt{\varepsilon_{k_g}^{(t)}}$ is obtained by solving the problem in (3). We show in Theorem 5 next that this algorithm exhibits very similar convergence behavior to that shown in (8).

**Theorem 5** (Local Convergence of Population GEM). *Suppose Assumptions 1 through 5 hold. Consider a GEM algorithm whose iterate $m_{k_g}(\theta_g^{(t-1)})$ at iteration $t$ is such that $m_{k_g}(\theta_g^{(t-1)}) \in \mathbb{B}_2(M_{k_g}(\boldsymbol{\theta}_g^{(t-1)}); \sqrt{\varepsilon_{k_g}^{(t)}})$, where the radius $\sqrt{\varepsilon_{k_g}^{(t)}}$ is obtained by solving the population counterpart of the problem in (3). Then, this algorithm converges to the ground truth parameters $\boldsymbol{\theta}_{k_g}^*$ as follows:*

$$||m_{k_g}(\boldsymbol{\theta}_g^{(t-1)}) - \boldsymbol{\theta}_{k_g}^*||_2 \leqslant \frac{\beta_g}{\lambda_g}||\boldsymbol{\theta}_{k_g}^{(t-1)} - \boldsymbol{\theta}_{k_g}^*||_2 + \epsilon(t) \qquad \forall \boldsymbol{\theta}_{k_g}^{(t-1)} \in \mathbb{B}_2(\boldsymbol{\theta}_{k_g}^*; a_g), \tag{9}$$

*where $\epsilon(t) = ||M_{k_g}(\boldsymbol{\theta}_g^{(t-1)}) - \boldsymbol{\theta}_{k_g}^{(t-1)}||_2^2 \to 0$ as $t \to \infty$.*

*Proof.*

$$||m_{k_g}(\boldsymbol{\theta}_g^{(t-1)}) - \boldsymbol{\theta}_{k_g}^*||_2 = ||m_{k_g}(\boldsymbol{\theta}_g^{(t-1)}) - M_{k_g}(\boldsymbol{\theta}_g^{(t-1)}) + M_{k_g}(\boldsymbol{\theta}_g^{(t-1)}) - \boldsymbol{\theta}_{k_g}^*||_2 \tag{10a}$$

$$\leqslant ||M_{k_g}(\boldsymbol{\theta}_g^{(t-1)}) - \boldsymbol{\theta}_{k_g}^*||_2 + ||M_{k_g}(\boldsymbol{\theta}_g^{(t-1)}) - m_{k_g}(\boldsymbol{\theta}_g^{(t-1)})||_2 \tag{10b}$$

$$\leqslant \frac{\beta_g}{\lambda_g}||\boldsymbol{\theta}_{k_g}^{(t-1)} - \boldsymbol{\theta}_{k_g}^*||_2 + ||M_{k_g}(\boldsymbol{\theta}_g^{(t-1)}) - m_{k_g}(\boldsymbol{\theta}_g^{(t-1)})||_2 \tag{10c}$$

$$< \frac{\beta_g}{\lambda_g}||\boldsymbol{\theta}_{k_g}^{(t-1)} - \boldsymbol{\theta}_{k_g}^*||_2 + \underbrace{||M_{k_g}(\boldsymbol{\theta}_g^{(t-1)}) - \boldsymbol{\theta}_{k_g}^{(t-1)}||_2}_{\epsilon^{(t)}}, \tag{10d}$$

where (10b) follows from the triangle inequality, (10c) relies on the convergence of $M_{k_g}(\theta_g)$ with $\frac{\beta_g}{\lambda_g} < 1$, and (10d) follows from the definition of $m_{k_g}(\theta_g)$. Now, we consider the $||M_{k_g}(\boldsymbol{\theta}_g^{(t-1)}) - \boldsymbol{\theta}_{k_g}^{(t-1)}||_2$ term. Firstly, observe that $\boldsymbol{\theta}_{k_g}^{(t)}$ for all $t \in [T]$ are iterates of a GEM algorithm. Moreover, recall that we assume that the true log-likelihood of our problem is bounded from above, and the expected complete-data log-likelihood $Q_g(\boldsymbol{\theta}_g|\boldsymbol{\theta}_g')$ is continuous in both its conditioning and input arguments. Therefore, by Theorem 1 in (Wu, 1983), the iterates must converge to a a stationary value of the true log-likelihood. This suggests that the quantity $\left[Q_g(M_g(\boldsymbol{\theta}_g^{(t-1)})|\boldsymbol{\theta}_g^{(t-1)}) - Q_g(\boldsymbol{\theta}_g^{(t-1)}|\boldsymbol{\theta}_g^{(t-1)})\right] \to 0$ as $t \to \infty$. Finally, by the assumed strong concavity of the expected complete-data log-likelihood function everywhere, we have that

$$Q_g(M_g(\boldsymbol{\theta}_g^{(t-1)})|\boldsymbol{\theta}_g^{(t-1)}) - Q_g(\boldsymbol{\theta}_g^{(t-1)}|\boldsymbol{\theta}_g^{(t-1)}) \geqslant \frac{\lambda_g}{2}||M_g(\boldsymbol{\theta}_g^{(t-1)}) - \boldsymbol{\theta}_g^{(t-1)}||_2^2,$$

proving that $||M_{k_g}(\boldsymbol{\theta}_g^{(t-1)}) - \boldsymbol{\theta}_{k_g}^{(t-1)}||_2 \to 0$ as $t \to \infty$. This concludes the proof. $\square$

## B.2 Solution Algorithm for Client Radius Problem Reformulation for Isotropic GMMs

In this section, we introduce the low-complexity Algorithm 4, which can be used to solve the reformulated client radius problem in (7). Subsequently, we introduce Proposition 2, which establishes the worst-case time complexity of Algorithm 4.

---

**Algorithm 4** Radius Problem $J_{k_g}(\boldsymbol{\theta}'_g)$ (7) Solution Algorithm

---

**Input:** $\boldsymbol{\theta}'_{k_g}, \widehat{M}_{k_g}(\boldsymbol{\theta}'_g), \varepsilon^{(0)}_{k_g,\text{lb}} = 0, \varepsilon^{(0)}_{k_g,\text{ub}} = ||\widehat{M}_{k_g}(\boldsymbol{\theta}'_g) - \boldsymbol{\theta}'_{k_g}||_2^2$
**Parameters:** Number of iterations $I$
**Output:** $\varepsilon^*_{k_g}$
 1: **for** $i = 0, \dots, I$ **do**
 2: $\quad \widehat{\varepsilon}^{(i)}_{k_g} \leftarrow \frac{\varepsilon^{(i)}_{k_g,\text{lb}} + \varepsilon^{(i)}_{k_g,\text{ub}}}{2}$
 3: $\quad$ Solve for $t^{(i)}_{k_g}$ minimizer of $F_k(\boldsymbol{\theta}'_g, \widehat{\varepsilon}^{(i)}_{k_g})$ (11).
 4: $\quad$ **if** $t^{(i)}_{k_g} = 0$ **then**
 5: $\quad\quad \varepsilon^{(i+1)}_{k_g,\text{lb}} \leftarrow \widehat{\varepsilon}^{(i)}_{k_g}$
 6: $\quad\quad \varepsilon^{(i+1)}_{k_g,\text{ub}} \leftarrow \varepsilon^{(i)}_{k_g,\text{ub}}$
 7: $\quad$ **else**
 8: $\quad\quad \varepsilon^{(i+1)}_{k_g,\text{ub}} \leftarrow \widehat{\varepsilon}^{(i)}_{k_g}$
 9: $\quad\quad \varepsilon^{(i+1)}_{k_g,\text{lb}} \leftarrow \varepsilon^{(i)}_{k_g,\text{lb}}$
10: $\quad$ **end if**
11: **end for**

---

where $F_k(\boldsymbol{\theta}'_g, \widehat{\varepsilon}^{(i)}_{k_g})$ is the optimization problem shown next.

$$F_k(\boldsymbol{\theta}'_g, \widehat{\varepsilon}^{(i)}_{k_g}) =$$

$$\begin{cases} \min\limits_{t_{k_g}, \alpha_{k_g} \in \mathbb{R}} \quad t_{k_g} \\[2mm] \text{s.t.} \quad \varepsilon_{k_g}\alpha_{k_g}^2 + \left[ \sum\limits_{n_g=1}^{N_g} \gamma_{k_g}(\widehat{\boldsymbol{x}}_{n_g}, \boldsymbol{\theta}'_g)\left( ||\widehat{\boldsymbol{x}}_{n_g} - \widehat{M}_{k_g}(\boldsymbol{\theta}'_g)||_2^2 - ||\widehat{\boldsymbol{x}}_{n_g} - \boldsymbol{\theta}'_{k_g}||_2^2 - \varepsilon_{k_g}\right) \right]\alpha_{k_g} + \\[4mm] \qquad\qquad\qquad - t_{k_g} + \left( \sum\limits_{n_g=1}^{N_g} \gamma_{k_g}(\widehat{\boldsymbol{x}}_{n_g}, \boldsymbol{\theta}'_g)\right) \sum\limits_{n_g=1}^{N_g} \gamma_{k_g}(\widehat{\boldsymbol{x}}_{n_g}, \boldsymbol{\theta}'_g)||\widehat{\boldsymbol{x}}_{n_g} - \boldsymbol{\theta}'_{k_g}||_2^2 \leq 0 \\[4mm] \alpha_{k_g} \geq \sum\limits_{n_g=1}^{N_g} \gamma_{k_g}(\widehat{\boldsymbol{x}}_{n_g}, \boldsymbol{\theta}'_g), \qquad\qquad t_{k_g} \geq 0. \end{cases}$$

$$(11)$$

**Proposition 2** (Local Radius Algorithm Convergence). *The Algorithm 4 converges to an optimal solution $\varepsilon^*_{k_g}$ of the optimization problem in* (7) *at a linear rate, with a worst-case time complexity of $\mathcal{O}(\log(\epsilon_{tol}^{-1}))$ per iteration, where $\epsilon_{tol}^{-1}$ is the solution tolerance of the feasibility problem* (11).

*Proof.* Consider the optimization problem in (7). Firstly, observe that the first constraint is non-decreasing in $\varepsilon_{k_g}$ for a fixed $\alpha_{k_g}$. This is because $\varepsilon_{k_g}\left(\alpha_{k_g}^2 - \alpha_{k_g}\sum_{n_g=1}^{N_g}\gamma_{k_g}(\widehat{\boldsymbol{x}}_{n_g}, \boldsymbol{\theta}'_g)\right) \geq 0$ due to the constraint that $\alpha_{k_g} \geq \sum_{n_g=1}^{N_g}\gamma_{k_g}(\widehat{\boldsymbol{x}}_{n_g}, \boldsymbol{\theta}'_g)$. Moreover, note that by the strong concavity of $\widehat{Q}_g(\boldsymbol{\theta}_g|\boldsymbol{\theta}'_g)$, we have that $\varepsilon_{k_g}$ must obey $0 \leq \sqrt{\varepsilon_{k_g}} \leq ||\widehat{M}_{k_g}(\boldsymbol{\theta}'_g) - \boldsymbol{\theta}'_{k_g}||_2$. Furthermore, it also follows from the strong concavity of $\widehat{Q}_g(\boldsymbol{\theta}_g|\boldsymbol{\theta}'_g)$ that $\varepsilon_{k_g} = 0$ must always be a feasible solution to the problem given that the algorithm has not yet converged. This is because the strong concavity of $\widehat{Q}_{k_g}(\boldsymbol{\theta}_{k_g}|\boldsymbol{\theta}'_g)$ for the GMM discussed in Section 5 suggests that $\sum_{n_g=1}^{N_g}\gamma_{k_g}(\widehat{\boldsymbol{x}}_{n_g}, \boldsymbol{\theta}'_g)||\widehat{\boldsymbol{x}}_{n_g} - \boldsymbol{\theta}'_{k_g})||_2^2 \geq \sum_{n_g=1}^{N_g}\gamma_{k_g}(\widehat{\boldsymbol{x}}_{n_g}, \boldsymbol{\theta}'_g)||\widehat{\boldsymbol{x}}_{n_g} - \widehat{M}_{k_g}(\boldsymbol{\theta}'_g)||_2^2$ with equality attained if and only if $\boldsymbol{\theta}'_{k_g} =$

$\widehat{M}_{k_g}(\boldsymbol{\theta}'_g)$. Therefore, $\alpha_{k_g}$ can be made arbitrarily large to make sure the constraint is satisfied. Combined with the uniqueness result presented in Proposition 1, these facts suggest that we can use a bisection approach such as the one shown in Algorithm 4 to obtain the optimal radius. Moreover, this also suggests that we can use the optimization problem presented in (11) to check the feasibility of a given $\widehat{\varepsilon}_{k_g}$. More specifically, an optimal $t_{k_g} = 0$ suggests that the constraints are already satisfied, and therefore the estimated $\widehat{\varepsilon}_{k_g}$ is feasible, and vice versa.

Now, observe that the feasibility check problem in (1) is a quadratically constrained quadratic program (QCQP) with 3 constraints and 2 1-dimensional decision variables. Therefore, it can readily be solved via the barrier method with worst-case time complexity of $\mathcal{O}(\log(\epsilon_{\text{tol}}^{-1}))$, where $\epsilon_{\text{tol}}^{-1}$ is the solution tolerance of the feasibility problem (11) (Nesterov & Nemirovskii, 1994). □

### B.3 SUPPLEMENTARY THEOREMS VERIFYING ASSUMPTIONS FOR ISOTROPIC GMMS

In this section, we verify three key assumptions to guarantee the convergence of our proposed `FedGEM` for the GMM discussed in Section 5. More specifically, we begin by proving that the population $Q_g(\boldsymbol{\theta}_g|\boldsymbol{\theta}'_g)$ function associated with the GMM under study obeys the FOS condition in Theorem 6. Subsequently, we derive the radius of the region for which the population M-step map for this model is indeed contractive in Theorem 7. Finally, we derive the upper bound on the distance between the population and finite-sample M-step maps in Theorem 8.

**Theorem 6** (GMM First-Order Stability). *Suppose $R_{min} = \tilde{\Omega}(\sqrt{\min\{d, K_g\}})$. Then the function $Q_g(\boldsymbol{\theta}_g|\boldsymbol{\theta}'_g)$ associated with the GMM described in this section obeys the first-order stability condition defined in 1 for all $\boldsymbol{\theta}_{k_g} \in \mathbb{B}_2(\boldsymbol{\theta}^*_{k_g}, a_g) \ \forall k_g \in [K_g]$, where $a_g \leqslant \frac{R_{min}}{2} - \sqrt{\min\{d, K_g\}} \max\{4\sqrt{2[\log(R_{min}/4)]_+}, 8\sqrt{3}\}$. That is*

$$||\nabla Q_g(M(\boldsymbol{\theta}_g)|\boldsymbol{\theta}_g) - \nabla Q(M(\boldsymbol{\theta}_g)|\boldsymbol{\theta}^*_g)||_2 \leqslant \beta_g||\boldsymbol{\theta}_g - \boldsymbol{\theta}^*_g||_2,$$

*with*

$$\beta_g = (1 + \pi'_g)\beta'_g + \pi'_g \pi_{max_g},$$

*where $\pi'_g$ is a constant depending on $K_g$, $R_{min}$, $a_g$, $d$, $\pi_{min_g}$, and $\pi_{max_g}$ whose explicit form can be found in the proof, and*

$$\beta'_g = K_g^2(2\kappa + 4)(2R_{max} + \min\{d, K_g\})^2 \exp\left(-\left(\frac{R_{min}}{2} - a_g\right)^2 \sqrt{\min\{d, K_g\}}/8\right),$$

*Proof.* We begin this proof by studying the FOS condition for each component $k_g$ separately. Firstly, note that

$$\nabla_{\boldsymbol{\theta}_{k_g}} Q(\boldsymbol{\theta}_g|\boldsymbol{\theta}'_g) = \mathbb{E}_{\boldsymbol{x}}[\gamma_{k_g}(\boldsymbol{x}, \boldsymbol{\theta}_g)(\boldsymbol{x} - \boldsymbol{\theta}'_{k_g})].$$

Therefore, we can plug in $\boldsymbol{\theta}_{k_g} = M_{k_g}(\boldsymbol{\theta}_g)$ to obtain the following.

$$||\nabla_{M_{k_g}(\boldsymbol{\theta}_g)} Q(M_g(\boldsymbol{\theta}_g)|\boldsymbol{\theta}_g) - \nabla_{M_{k_g}(\boldsymbol{\theta}_g)} Q(M_g(\boldsymbol{\theta}_g)|\boldsymbol{\theta}^*_g)||_2 \tag{12a}$$

$$= ||\mathbb{E}_{\boldsymbol{x}}[(\gamma_{k_g}(\boldsymbol{x}, \boldsymbol{\theta}_g) - \gamma_{k_g}(\boldsymbol{x}, \boldsymbol{\theta}^*_g))(\boldsymbol{x} - M_{k_g}(\boldsymbol{\theta}_g))]||_2 \tag{12b}$$

$$= ||\mathbb{E}_{\boldsymbol{x}}[(\gamma_{k_g}(\boldsymbol{x}, \boldsymbol{\theta}_g) - \gamma_{k_g}(\boldsymbol{x}, \boldsymbol{\theta}^*_g))(\boldsymbol{x} - \boldsymbol{\theta}_{k_g} + \boldsymbol{\theta}_{k_g} - M_{k_g}(\boldsymbol{\theta}_g))]||_2 \tag{12c}$$

$$= ||\mathbb{E}_{\boldsymbol{x}}[(\gamma_{k_g}(\boldsymbol{x}, \boldsymbol{\theta}_g) - \gamma_{k_g}(\boldsymbol{x}, \boldsymbol{\theta}^*_g))(\boldsymbol{x} - \boldsymbol{\theta}_{k_g})] -$$
$$\mathbb{E}_{\boldsymbol{x}}[(\gamma_{k_g}(\boldsymbol{x}, \boldsymbol{\theta}_g) - \gamma_{k_g}(\boldsymbol{x}, \boldsymbol{\theta}^*_g))(M_{k_g}(\boldsymbol{\theta}_g) - \boldsymbol{\theta}_{k_g})]||_2 \tag{12d}$$

$$\leqslant \underbrace{||\mathbb{E}_{\boldsymbol{x}}[(\gamma_{k_g}(\boldsymbol{x}, \boldsymbol{\theta}_g) - \gamma_{k_g}(\boldsymbol{x}, \boldsymbol{\theta}^*_g))(\boldsymbol{x} - \boldsymbol{\theta}_{k_g})]||_2}_{A_1} +$$

$$\underbrace{||\mathbb{E}_{\boldsymbol{x}}[(\gamma_{k_g}(\boldsymbol{x}, \boldsymbol{\theta}_g) - \gamma_{k_g}(\boldsymbol{x}, \boldsymbol{\theta}^*_g))(M_{k_g}(\boldsymbol{\theta}_g) - \boldsymbol{\theta}_{k_g})]||_2}_{A_2}, \tag{12e}$$

where (12d) follows by the linearity of the expectation operator, and (12e) follows from the triangle inequality. Now, it is established in Theorem 4 in (Yan et al., 2017) that

$$A_1 \leqslant \frac{\beta'_g}{K_g} \sum_{k_g=1}^{K_g} ||\boldsymbol{\theta}_{k_g} - \boldsymbol{\theta}^*_{k_g}||_2, \tag{13}$$

where $\beta'_g$ is defined in our theorem statement. Therefore, it remains to obtain an upper bound for $A_2$. We begin this as follows.

$$A_2 := ||\mathbb{E}_{\boldsymbol{x}}[(\gamma_{k_g}(\boldsymbol{x}, \boldsymbol{\theta}_g) - \gamma_{k_g}(\boldsymbol{x}, \boldsymbol{\theta}_g^*))(M_{k_g}(\boldsymbol{\theta}_g) - \boldsymbol{\theta}_{k_g})]||_2 \tag{14a}$$

$$= ||\mathbb{E}_{\boldsymbol{x}}[(\gamma_{k_g}(\boldsymbol{x}, \boldsymbol{\theta}_g) - \gamma_{k_g}(\boldsymbol{x}, \boldsymbol{\theta}_g^*))](M_{k_g}(\boldsymbol{\theta}_g) - \boldsymbol{\theta}_{k_g})||_2 \tag{14b}$$

$$= \left|\mathbb{E}_{\boldsymbol{x}}[\gamma_{k_g}(\boldsymbol{x}, \boldsymbol{\theta}_g)] - \mathbb{E}_{\boldsymbol{x}}[\gamma_{k_g}(\boldsymbol{x}, \boldsymbol{\theta}_g^*)]\right| \, ||M_{k_g}(\boldsymbol{\theta}_g) - \boldsymbol{\theta}_{k_g}||_2 \tag{14c}$$

$$= \left|\mathbb{E}_{\boldsymbol{x}}[\gamma_{k_g}(\boldsymbol{x}, \boldsymbol{\theta}_g)] - \mathbb{E}_{\boldsymbol{x}}[\gamma_{k_g}(\boldsymbol{x}, \boldsymbol{\theta}_g^*)]\right| \, \left|\left|\frac{\mathbb{E}_{\boldsymbol{x}}[\gamma_{k_g}(\boldsymbol{x}, \boldsymbol{\theta}_g)\boldsymbol{x}]}{\mathbb{E}_{\boldsymbol{x}}[\gamma_{k_g}(\boldsymbol{x}, \boldsymbol{\theta}_g)]} - \boldsymbol{\theta}_{k_g}\right|\right|_2 \tag{14d}$$

$$= \underbrace{\frac{\left|\mathbb{E}_{\boldsymbol{x}}[\gamma_{k_g}(\boldsymbol{x}, \boldsymbol{\theta}_g)] - \mathbb{E}_{\boldsymbol{x}}[\gamma_{k_g}(\boldsymbol{x}, \boldsymbol{\theta}_g^*)]\right|}{\mathbb{E}_{\boldsymbol{x}}[\gamma_{k_g}(\boldsymbol{x}, \boldsymbol{\theta}_g)]}}_{A_{2a}} \underbrace{||\mathbb{E}_{\boldsymbol{x}}[\gamma_{k_g}(\boldsymbol{x}, \boldsymbol{\theta}_g)(\boldsymbol{x} - \boldsymbol{\theta}_{k_g})]||_2}_{A_{2b}} \tag{14e}$$

where (14b) is obtained by realizing that $M_{k_g}(\boldsymbol{\theta}_g)$ is comprised of expectations in $\boldsymbol{x}$, and thus is no longer random, and (14c) follows from the linearity of the expectation, and from the fact that the expectation terms become scalar quantities, and can therefore be taken outside the norm. Now, we must obtain upper bounds for the terms $A_{2a}$ and $A_{2b}$. In bounding $A_{2a}$, we proceed by obtaining an upper bound for the numerator and a lower bound for the denominator. To achieve this, firstly observe the following.

$$\mathbb{E}_{\boldsymbol{x}}[\gamma_{k_g}(\boldsymbol{x}, \boldsymbol{\theta}_g^*)] = \int_{\boldsymbol{x}} \frac{\pi_{k_g}\phi(\boldsymbol{x}|\boldsymbol{\theta}_{k_g}^*)}{\sum_{j_g=1}^{K_g} \pi_{j_g}\phi(\boldsymbol{x}|\boldsymbol{\theta}_{j_g}^*)} \left(\sum_{m_g=1}^{K_g} \pi_{m_g}\phi(\boldsymbol{x}|\boldsymbol{\theta}_{m_g}^*)\right) d\boldsymbol{x} \tag{15a}$$

$$= \int_{\boldsymbol{x}} \pi_{k_g}\phi(\boldsymbol{x}|\boldsymbol{\theta}_{k_g}^*)d\boldsymbol{x} \tag{15b}$$

$$= \pi_{k_g}. \tag{15c}$$

Next, we examine the $\mathbb{E}_{\boldsymbol{x}}[\gamma_{k_g}(\boldsymbol{x}, \boldsymbol{\theta}_g)]$ term as follows.

$$\mathbb{E}_{\boldsymbol{x}}[\gamma_{k_g}(\boldsymbol{x}, \boldsymbol{\theta}_g)] = \sum_{l_g=1}^{K_g} \pi_{l_g}\mathbb{E}_{\boldsymbol{x}}[\gamma_{k_g}(\boldsymbol{x}, \boldsymbol{\theta}_g)|\boldsymbol{x} \sim \mathcal{N}(\boldsymbol{\theta}_{l_g}^*, I_d)] \tag{16a}$$

$$= \sum_{l_g=1}^{K_g} \pi_{l_g}\int_{\boldsymbol{x}} \gamma_{k_g}(\boldsymbol{x}, \boldsymbol{\theta}_g)\phi(\boldsymbol{x}|\boldsymbol{\theta}_{l_g}^*)d\boldsymbol{x}, \tag{16b}$$

which follows from the law of total expectation. Now, we analyze two cases as follows.

- **Case 1:** $l_g = k_g$

  In this case, we have that

  $$\int_{\boldsymbol{x}} \underbrace{\gamma_{k_g}(\boldsymbol{x}, \boldsymbol{\theta}_g)}_{\leqslant 1} \phi(\boldsymbol{x}|\boldsymbol{\theta}_{l_g}^*)d\boldsymbol{x} \leqslant \int_{\boldsymbol{x}} \phi(\boldsymbol{x}|\boldsymbol{\theta}_{k_g}^*)d\boldsymbol{x} = 1.$$

  Thus, $\pi_{l_g}\int_{\boldsymbol{x}} \gamma_{k_g}(\boldsymbol{x}, \boldsymbol{\theta}_g)\phi(\boldsymbol{x}|\boldsymbol{\theta}_{l_g}^*)d\boldsymbol{x} = \pi_{k_g}$

- **Case 2:** $l_g \neq k_g$

  In this case, we have that

  $$\gamma_{k_g}(\boldsymbol{x}, \boldsymbol{\theta}_g) = \frac{\pi_{k_g}\exp(-\frac{1}{2}||\boldsymbol{x} - \boldsymbol{\theta}_{k_g}||_2^2)}{\sum_{j_g=1}^{K_g} \pi_{j_g}\exp(-\frac{1}{2}||\boldsymbol{x} - \boldsymbol{\theta}_{j_g}||_2^2)} \tag{17a}$$

  $$\leqslant \frac{\pi_{k_g}\exp(-\frac{1}{2}||\boldsymbol{x} - \boldsymbol{\theta}_{k_g}||_2^2)}{\pi_{l_g}\exp(-\frac{1}{2}||\boldsymbol{x} - \boldsymbol{\theta}_{l_g}||_2^2)} \tag{17b}$$

  $$= \frac{\pi_{k_g}}{\pi_{l_g}}\exp\left[-\frac{1}{2}(||\boldsymbol{x} - \boldsymbol{\theta}_{k_g}||_2^2 - ||\boldsymbol{x} - \boldsymbol{\theta}_{l_g}||_2^2)\right]. \tag{17c}$$

Now, define event $\mathcal{A}_{k_g, r_g} = \{\boldsymbol{x} \colon \boldsymbol{x} \sim \mathcal{N}(\boldsymbol{\theta}_{l_g}^*, I_d), ||\boldsymbol{x} - \boldsymbol{\theta}_{l_g}^*||_2 \leqslant r_g\}$, where $r_g \in \mathbb{R}$ is some constant, which we will obtain bounds for later. Since $l_g \neq k_g$ we can obtain a lower bound for the quantity $||\boldsymbol{x} - \boldsymbol{\theta}_{l_g}||_2$ for $\boldsymbol{x} \in \mathcal{A}_{k_g, r_g}$ via the triangle inequality as follows.

$$
\begin{aligned}
||\boldsymbol{x} - \boldsymbol{\theta}_{k_g}||_2 &= ||\boldsymbol{x} - \boldsymbol{\theta}_{l_g}^* + \boldsymbol{\theta}_{l_g}^* - \boldsymbol{\theta}_{k_g}||_2 \\
&\geqslant ||\boldsymbol{\theta}_{l_g}^* - \boldsymbol{\theta}_{k_g}||_2 - ||\boldsymbol{\theta}_{l_g}^* - \boldsymbol{x}||_2 \\
&\geqslant ||\boldsymbol{\theta}_{l_g}^* - \boldsymbol{\theta}_{k_g}^* + \boldsymbol{\theta}_{k_g}^* - \boldsymbol{\theta}_{k_g}||_2 - r_g \\
&\geqslant ||\boldsymbol{\theta}_{l_g}^* - \boldsymbol{\theta}_{k_g}^*||_2 - ||\boldsymbol{\theta}_{k_g} - \boldsymbol{\theta}_{k_g}^*||_2 - r_g \\
&\geqslant R_{\min} - a_g - r_g.
\end{aligned}
$$

Similarly, we can obtain an upper bound for the quantity $||\boldsymbol{x} - \boldsymbol{\theta}_{k_g}||_2$ for $\boldsymbol{x} \in \mathcal{A}_{k_g, r_g}$ as follows.

$$
\begin{aligned}
||\boldsymbol{x} - \boldsymbol{\theta}_{l_g}||_2 &= ||\boldsymbol{x} - \boldsymbol{\theta}_{l_g}^* + \boldsymbol{\theta}_{l_g}^* - \boldsymbol{\theta}_{l_g}||_2 \\
&\leqslant ||\boldsymbol{x} - \boldsymbol{\theta}_{l_g}^*||_2 + ||\boldsymbol{\theta}_{l_g}^* - \boldsymbol{\theta}_{l_g}||_2 \\
&\leqslant r_g + a_g.
\end{aligned}
$$

Therefore, we have that

$$
\frac{\pi_{k_g}}{\pi_{l_g}} \exp\left[-\frac{1}{2}(||\boldsymbol{x} - \boldsymbol{\theta}_{k_g}||_2^2 - ||\boldsymbol{x} - \boldsymbol{\theta}_{l_g}||_2^2)\right] \leqslant \frac{\pi_{k_g}}{\pi_{l_g}} \exp\left[-\frac{1}{2}(R_{\min}^2 - 2R_{\min}(a_g + r_g))\right],
$$

with the requirement that $r_g < \frac{R_{\min}}{2} - a_g$ to ensure the negativity of the term inside the exponent. This allows us to write the following for $l_g \neq k_g$.

$$
\int_{\boldsymbol{x}} \gamma_{k_g}(\boldsymbol{x}, \boldsymbol{\theta}_g) \phi(\boldsymbol{x} | \boldsymbol{\theta}_{l_g}^*) d\boldsymbol{x}
$$

$$
\leqslant \int_{\boldsymbol{x} \in \mathcal{A}_{k_g, r_g}} \frac{\pi_{k_g}}{\pi_{l_g}} \exp\left[-\frac{1}{2}(R_{\min}^2 - 2R_{\min}(a_g + r_g))\right] \phi(\boldsymbol{x} | \boldsymbol{\theta}_{l_g}^*) d\boldsymbol{x} +
$$

$$
\int_{\boldsymbol{x} \notin \mathcal{A}_{k_g, r_g}} \underbrace{\gamma_{k_g}(\boldsymbol{x}, \boldsymbol{\theta}_g)}_{\leqslant 1} \phi(\boldsymbol{x} | \boldsymbol{\theta}_{l_g}^*) d\boldsymbol{x} \quad \text{(18a)}
$$

$$
\leqslant \frac{\pi_{k_g}}{\pi_{l_g}} \exp\left[-\frac{1}{2}(R_{\min}^2 - 2R_{\min}(a_g + r_g))\right] + \int_{\boldsymbol{x} \notin \mathcal{A}_{k_g, r_g}} \phi(\boldsymbol{x} | \boldsymbol{\theta}_{l_g}^*) d\boldsymbol{x} \quad \text{(18b)}
$$

$$
= \frac{\pi_{k_g}}{\pi_{l_g}} \exp\left[-\frac{1}{2}(R_{\min}^2 - 2R_{\min}(a_g + r_g))\right] + P(||\boldsymbol{x} - \boldsymbol{\theta}_{l_g}^*||_2 \geqslant r_g | \boldsymbol{x} \sim \mathcal{N}(\boldsymbol{\theta}_{l_g}^*, I_d)) \quad \text{(18c)}
$$

$$
\leqslant \frac{\pi_{k_g}}{\pi_{l_g}} \exp\left[-\frac{1}{2}(R_{\min}^2 - 2R_{\min}(a_g + r_g))\right] + \exp\left(-\frac{r_g \sqrt{d}}{2}\right), \quad \text{(18d)}
$$

where (18d) follows form standard tail analysis shown in Lemma 8 in (Yan et al., 2017) for $r_g \geqslant 2\sqrt{d}$.

Putting together the two cases analyzed previously, we obtain the following.

$$
\mathbb{E}_{\boldsymbol{x}}[\gamma_{k_g}(\boldsymbol{x}, \boldsymbol{\theta}_g)] \leqslant \pi_{k_g} + \sum_{l_g \in [K_g], l_g \neq k_g} \pi_{l_g} \left[\frac{\pi_{k_g}}{\pi_{l_g}} \exp\left[-\frac{1}{2}(R_{\min}^2 - 2R_{\min}(a_g + r_g))\right] + \exp\left(-\frac{r_g \sqrt{d}}{2}\right)\right]
$$

$$
= \pi_{k_g} + \pi_{k_g}(K_g - 1) \exp\left[-\frac{1}{2}(R_{\min}^2 - 2R_{\min}(a_g + r_g))\right] + (1 - \pi_{k_g}) \exp\left(-\frac{r_g \sqrt{d}}{2}\right).
$$

This allows us to directly observe that

$$
\mathbb{E}_{\boldsymbol{x}}[\gamma_{k_g}(\boldsymbol{x}, \boldsymbol{\theta}_g)] - \mathbb{E}_{\boldsymbol{x}}[\gamma_{k_g}(\boldsymbol{x}, \boldsymbol{\theta}_g^*)] \leqslant \pi_{k_g}(K_g - 1) \exp\left[-\frac{1}{2}(R_{\min}^2 - 2R_{\min}(a_g + r_g))\right] + (1 - \pi_{k_g}) \exp\left(-\frac{r_g \sqrt{d}}{2}\right).
$$

Now, it remains to obtain a lower bound for $\mathbb{E}_{\boldsymbol{x}}[\gamma_{k_g}(\boldsymbol{x}, \boldsymbol{\theta}_g)]$, which we do as follows. First, define an event $\mathcal{B}_{k_g, r_g} = \{\boldsymbol{x} \colon \boldsymbol{x} \sim \mathcal{N}(\boldsymbol{\theta}_{k_g}^*, I_d), \|\boldsymbol{x} - \boldsymbol{\theta}_{k_g}^*\|_2 \leqslant r_g)\}$, where $r_g \in \mathbb{R}$ is such that $2\sqrt{d} \leqslant r_g < \frac{R_{\min}}{2} - a_g$ as we saw previously. Then we have that

$$\mathbb{E}_{\boldsymbol{x}}[\gamma_{k_g}(\boldsymbol{x}, \boldsymbol{\theta}_g)] = P(\mathcal{B}_{k_g, r_g})\mathbb{E}_{\boldsymbol{x}}[\gamma_{k_g}(\boldsymbol{x}, \boldsymbol{\theta}_g)|\mathcal{B}_{k_g, r_g}] + P(\mathcal{B}_{k_g, r_g}^c)\mathbb{E}_{\boldsymbol{x}}[\gamma_{k_g}(\boldsymbol{x}, \boldsymbol{\theta}_g)|\mathcal{B}_{k_g, r_g}^c] \quad \text{(19a)}$$

$$\geqslant P(\mathcal{B}_{k_g, r_g})\mathbb{E}_{\boldsymbol{x}}[\gamma_{k_g}(\boldsymbol{x}, \boldsymbol{\theta}_g)|\mathcal{B}_{k_g, r_g}], \quad \text{(19b)}$$

where (19a) follows from the law of total expectation, and (19b) follows from the fact that $\gamma_{k_g}(\boldsymbol{x}, \boldsymbol{\theta}_g)$ is uniformly lower bounded by 0. Now, consider $\boldsymbol{x} \in \mathcal{B}_{k_g, r_g}$. For $j_g \in [K_g]$, $j_g \neq k_g$, we can lower bound the quantity $\|\boldsymbol{x} - \boldsymbol{\theta}_{j_g}\|_2$ via the triangle inequality as follows.

$$\begin{aligned}
\|\boldsymbol{x} - \boldsymbol{\theta}_{j_g}\|_2 &= \|\boldsymbol{x} - \boldsymbol{\theta}_{k_g}^* + \boldsymbol{\theta}_{k_g}^* - \boldsymbol{\theta}_{j_g}\|_2 \\
&\geqslant \|\boldsymbol{\theta}_{j_g} - \boldsymbol{\theta}_{k_g}^*\|_2 - \|\boldsymbol{x} - \boldsymbol{\theta}_{k_g}^*\|_2 \\
&\geqslant \|\boldsymbol{\theta}_{j_g} - \boldsymbol{\theta}_{j_g}^* + \boldsymbol{\theta}_{j_g}^* - \boldsymbol{\theta}_{k_g}^*\|_2 - r_g \\
&\geqslant \|\boldsymbol{\theta}_{j_g}^* - \boldsymbol{\theta}_{k_g}^*\|_2 - \|\boldsymbol{\theta}_{j_g} - \boldsymbol{\theta}_{j_g}^*\|_2 - r_g \\
&\geqslant R_{\min} - r_g - a_g.
\end{aligned}$$

Similarly, for $j_g = k_g$ we can upper bound the quantity $\|\boldsymbol{x} - \boldsymbol{\theta}_{j_g}\|_2$ via the triangle inequality as follows.

$$\begin{aligned}
\|\boldsymbol{x} - \boldsymbol{\theta}_{j_g}\|_2 &= \|\boldsymbol{x} - \boldsymbol{\theta}_{k_g}^* + \boldsymbol{\theta}_{k_g}^* - \boldsymbol{\theta}_{k_g}\|_2 \\
&\leqslant \|\boldsymbol{x} - \boldsymbol{\theta}_{k_g}^*\|_2 + \|\boldsymbol{\theta}_{k_g}^* - \boldsymbol{\theta}_{k_g}\|_2 \\
&\leqslant r_g + a_g.
\end{aligned}$$

Now, let us write $\gamma_{k_g}(\boldsymbol{x}, \boldsymbol{\theta}_g)$ differently to simplify the analysis. To do that, let us define the following.

$$\begin{aligned}
\breve{\gamma}_k(\boldsymbol{x}, \boldsymbol{\theta}_g) &:= \frac{1}{\gamma_{k_g}(\boldsymbol{x}, \boldsymbol{\theta}_g)} \\
&= \frac{\sum_{j_g=1}^{K_g} \pi_{j_g}\phi(\boldsymbol{x}|\boldsymbol{\theta}_{j_g})}{\pi_{k_g}\phi(\boldsymbol{x}|\boldsymbol{\theta}_{k_g})} \\
&= \sum_{j_g=1}^{K_g} \frac{\pi_{j_g}}{\pi_{k_g}} \exp\left[-\frac{1}{2}(\|\boldsymbol{x} - \boldsymbol{\theta}_{j_g}\|_2^2 - \|\boldsymbol{x} - \boldsymbol{\theta}_{k_g}\|_2^2)\right]
\end{aligned}$$

Therefore, given that $\boldsymbol{x} \in \mathcal{B}_{k_g, r_g}$, we can write

$$\begin{aligned}
\gamma_{k_g}(\boldsymbol{x}, \boldsymbol{\theta}_g) &= \frac{1}{\breve{\gamma}_k(\boldsymbol{x}, \boldsymbol{\theta}_g)} \\
&= \frac{1}{\sum_{j_g=1}^{K_g} \frac{\pi_{j_g}}{\pi_{k_g}} \exp\left[-\frac{1}{2}(\|\boldsymbol{x} - \boldsymbol{\theta}_{j_g}\|_2^2 - \|\boldsymbol{x} - \boldsymbol{\theta}_{k_g}\|_2^2)\right]} \\
&\geqslant \frac{1}{1 + \sum_{j_g \in [K_g], j_g \neq k_g} \frac{\pi_{j_g}}{\pi_{k_g}} \exp\left[-\frac{1}{2}(R_{\min}^2 - 2R_{\min}(r_g + a_g))\right]} \\
&= \frac{1}{1 + \frac{1 - \pi_{k_g}}{\pi_{k_g}} \exp\left[-\frac{1}{2}(R_{\min}^2 - 2R_{\min}(r_g + a_g))\right]}.
\end{aligned}$$

This lower bound on $\gamma_{k_g}(\boldsymbol{x}, \boldsymbol{\theta}_g)$ in the case where $\boldsymbol{x} \in \mathcal{B}_{k_g, r_g}$ allows us to write the following.

$$\mathbb{E}_{\boldsymbol{x}}[\gamma_{k_g}(\boldsymbol{x}, \boldsymbol{\theta}_g)|\mathcal{B}_{k_g, r_g}] \geqslant \frac{1}{1 + \frac{1 - \pi_{k_g}}{\pi_{k_g}} \exp\left[-\frac{1}{2}(R_{\min}^2 - 2R_{\min}(r_g + a_g))\right]}$$

Then, we can bound $\mathbb{E}_{\boldsymbol{x}}[\gamma_{k_g}(\boldsymbol{x}, \boldsymbol{\theta}_g)]$ as follows.

$$\mathbb{E}_{\boldsymbol{x}}[\gamma_{k_g}(\boldsymbol{x}, \boldsymbol{\theta}_g)] \geqslant \frac{P(\mathcal{B}_{k_g, r_g})}{1 + \frac{1 - \pi_{k_g}}{\pi_{k_g}} \exp\left[-\frac{1}{2}(R_{\min}^2 - 2R_{\min}(r_g + a_g))\right]} \tag{20}$$

$$= \frac{P(\boldsymbol{x} \sim \mathcal{N}(\boldsymbol{\theta}_{k_g}^*, I_d)) P(||\boldsymbol{x} - \boldsymbol{\theta}_{k_g}^*||_2 \leqslant r_g | \boldsymbol{x} \sim \mathcal{N}(\boldsymbol{\theta}_{k_g}^*, I_d))}{1 + \frac{1 - \pi_{k_g}}{\pi_{k_g}} \exp\left[-\frac{1}{2}(R_{\min}^2 - 2R_{\min}(r_g + a_g))\right]} \tag{21}$$

$$= \frac{\pi_{k_g} P(||\boldsymbol{x} - \boldsymbol{\theta}_{k_g}^*||_2 \leqslant r_g | \boldsymbol{x} \sim \mathcal{N}(\boldsymbol{\theta}_{k_g}^*, I_d))}{1 + \frac{1 - \pi_{k_g}}{\pi_{k_g}} \exp\left[-\frac{1}{2}(R_{\min}^2 - 2R_{\min}(r_g + a_g))\right]} \tag{22}$$

$$= \frac{\pi_{k_g} (1 - P(||\boldsymbol{x} - \boldsymbol{\theta}_{k_g}^*||_2 \geqslant r_g | \boldsymbol{x} \sim \mathcal{N}(\boldsymbol{\theta}_{k_g}^*, I_d)))}{1 + \frac{1 - \pi_{k_g}}{\pi_{k_g}} \exp\left[-\frac{1}{2}(R_{\min}^2 - 2R_{\min}(r_g + a_g))\right]} \tag{23}$$

$$\geqslant \frac{\pi_{k_g} \left[1 - \exp\left(-\frac{r_g \sqrt{d}}{2}\right)\right]}{1 + \frac{1 - \pi_{k_g}}{\pi_{k_g}} \exp\left[-\frac{1}{2}(R_{\min}^2 - 2R_{\min}(r_g + a_g))\right]} \tag{24}$$

$$\tag{25}$$

As a result, our previous analysis allows us to write the following uniform bound for the term $A_{2a}$ for all $k_g \in [K_g]$.

$$\frac{\left|\mathbb{E}_{\boldsymbol{x}}[\gamma_{k_g}(\boldsymbol{x}, \boldsymbol{\theta}_g)] - \mathbb{E}_{\boldsymbol{x}}[\gamma_{k_g}(\boldsymbol{x}, \boldsymbol{\theta}_g^*)]\right|}{\mathbb{E}_{\boldsymbol{x}}[\gamma_{k_g}(\boldsymbol{x}, \boldsymbol{\theta}_g)]}$$

$$\leqslant \max \left\{ \frac{\pi_{\max_g}(K_g - 1) \exp\left[-\frac{1}{2}(R_{\min}^2 - 2R_{\min}(a_g + r_g))\right] + (1 - \pi_{\min_g}) \exp\left(-\frac{r_g \sqrt{d}}{2}\right)}{\frac{\pi_{\min_g}\left[1 - \exp\left(-\frac{r_g \sqrt{d}}{2}\right)\right]}{1 + \frac{1 - \pi_{\min_g}}{\pi_{\max_g}} \exp\left[-\frac{1}{2}(R_{\min}^2 - 2R_{\min}(r_g + a_g))\right]}}, \right.$$

$$\left. \frac{\pi_{\max_g}\left(1 - \frac{\left[1 - \exp\left(-\frac{r_g \sqrt{d}}{2}\right)\right]}{1 + \frac{1 - \pi_{\max_g}}{\pi_{\min_g}} \exp\left[-\frac{1}{2}(R_{\min}^2 - 2R_{\min}(r_g + a_g))\right]}\right)}{\frac{\pi_{\min_g}\left[1 - \exp\left(-\frac{r_g \sqrt{d}}{2}\right)\right]}{1 + \frac{1 - \pi_{\min_g}}{\pi_{\max_g}} \exp\left[-\frac{1}{2}(R_{\min}^2 - 2R_{\min}(r_g + a_g))\right]}} \right\}$$

$$:= \pi_g'$$

Thus, it remains to obtain an upper bound for $A_{2b}$. We achieve this as follows.

$$A_{2b} := ||\mathbb{E}_{\boldsymbol{x}}[\gamma_{k_g}(\boldsymbol{x}, \boldsymbol{\theta}_g)(\boldsymbol{x} - \boldsymbol{\theta}_{k_g})]||_2 \tag{26a}$$

$$= ||\mathbb{E}_{\boldsymbol{x}}[\gamma_{k_g}(\boldsymbol{x}, \boldsymbol{\theta}_g)(\boldsymbol{x} - \boldsymbol{\theta}_{k_g})] - \mathbb{E}_{\boldsymbol{x}}[\gamma_{k_g}(\boldsymbol{x}, \boldsymbol{\theta}_g^*)(\boldsymbol{x} - \boldsymbol{\theta}_{k_g})] + \mathbb{E}_{\boldsymbol{x}}[\gamma_{k_g}(\boldsymbol{x}, \boldsymbol{\theta}_g^*)(\boldsymbol{x} - \boldsymbol{\theta}_{k_g})]||_2 \tag{26b}$$

$$\leqslant ||\mathbb{E}_{\boldsymbol{x}}[(\gamma_{k_g}(\boldsymbol{x}, \boldsymbol{\theta}_g) - \gamma_{k_g}(\boldsymbol{x}, \boldsymbol{\theta}_g^*)(\boldsymbol{x} - \boldsymbol{\theta}_{k_g})]||_2 + ||\mathbb{E}_{\boldsymbol{x}}[\gamma_{k_g}(\boldsymbol{x}, \boldsymbol{\theta}_g^*)(\boldsymbol{x} - \boldsymbol{\theta}_{k_g})]||_2 \tag{26c}$$

$$\leqslant \frac{\beta_g'}{K_g} \sum_{j_g=1}^{K_g} ||\boldsymbol{\theta}_{j_g} - \boldsymbol{\theta}_{j_g}^*||_2 + ||\mathbb{E}_{\boldsymbol{x}}[\gamma_{k_g}(\boldsymbol{x}, \boldsymbol{\theta}_g^*)\boldsymbol{x}] - \mathbb{E}_{\boldsymbol{x}}[\gamma_{k_g}(\boldsymbol{x}, \boldsymbol{\theta}_g^*)\boldsymbol{\theta}_{k_g}]||_2 \tag{26d}$$

$$= \frac{\beta_g'}{K_g} \sum_{j_g=1}^{K_g} ||\boldsymbol{\theta}_{j_g} - \boldsymbol{\theta}_{j_g}^*||_2 + \left|\left| \int_{\boldsymbol{x}} \frac{\pi_{k_g}\phi(\boldsymbol{x}|\boldsymbol{\theta}_{k_g}^*)}{\sum_{j_g=1}^{K_g} \pi_{j_g}\phi(\boldsymbol{x}|\boldsymbol{\theta}_{j_g}^*)} \boldsymbol{x} \left(\sum_{m_g=1}^{K_g} \pi_{m_g}\phi(\boldsymbol{x}|\boldsymbol{\theta}_{m_g}^*)\right) d\boldsymbol{x} - \right.$$
$$\left. \int_{\boldsymbol{x}} \frac{\pi_{k_g}\phi(\boldsymbol{x}|\boldsymbol{\theta}_{k_g}^*)}{\sum_{j_g=1}^{K_g} \pi_{j_g}\phi(\boldsymbol{x}|\boldsymbol{\theta}_{j_g}^*)} \boldsymbol{\theta}_{k_g} \left(\sum_{m_g=1}^{K_g} \pi_{m_g}\phi(\boldsymbol{x}|\boldsymbol{\theta}_{m_g}^*)\right) d\boldsymbol{x} \right|\right|_2 \tag{26e}$$

$$= \frac{\beta_g'}{K_g} \sum_{j_g=1}^{K_g} ||\boldsymbol{\theta}_{j_g} - \boldsymbol{\theta}_{j_g}^*||_2 + \pi_{k_g} ||\boldsymbol{\theta}_{k_g} - \boldsymbol{\theta}_{k_g}^*||_2 \tag{26f}$$

$$\leqslant \frac{\beta_g'}{K_g} \sum_{j_g=1}^{K_g} ||\boldsymbol{\theta}_{j_g} - \boldsymbol{\theta}_{j_g}^*||_2 + \pi_{\max_g} ||\boldsymbol{\theta}_{k_g} - \boldsymbol{\theta}_{k_g}^*||_2, \tag{26g}$$

where (26c) follows from the linearity of the expectation and the triangle inequality, and (26d) follows by leveraging the upper bound derived in Theorem 4 in (Yan et al., 2017) as discussed earlier. Therefore, we can summarize the results we have obtained thus far as follows.

$$||\nabla_{M_{k_g}(\boldsymbol{\theta}_g)} Q(M_g(\boldsymbol{\theta}_g)|\boldsymbol{\theta}_g) - \nabla_{M_{k_g}(\boldsymbol{\theta}_g)} Q(M_g(\boldsymbol{\theta}_g)|\boldsymbol{\theta}_g^*)||_2$$

$$\leqslant \frac{\beta_g'}{K_g} \sum_{j_g=1}^{K_g} ||\boldsymbol{\theta}_{j_g} - \boldsymbol{\theta}_{j_g}^*||_2 + \pi_g' \left[ \frac{\beta_g'}{K_g} \sum_{j_g=1}^{K_g} ||\boldsymbol{\theta}_{j_g} - \boldsymbol{\theta}_{j_g}^*||_2 + \pi_{\max_g} ||\boldsymbol{\theta}_{k_g} - \boldsymbol{\theta}_{k_g}^*||_2 \right]$$

$$= \frac{(1 + \pi_g')\beta_g'}{K_g} \sum_{j_g=1}^{K_g} ||\boldsymbol{\theta}_{j_g} - \boldsymbol{\theta}_{j_g}^*||_2 + \pi_g' \pi_{\max_g} ||\boldsymbol{\theta}_{k_g} - \boldsymbol{\theta}_{k_g}^*||_2.$$

Now, observe that

$$||\nabla Q(M_g(\boldsymbol{\theta}_g)|\boldsymbol{\theta}_g) - \nabla Q(M_g(\boldsymbol{\theta}_g)|\boldsymbol{\theta}_g^*)||_2^2$$

$$= \sum_{k_g=1}^{K_g} ||\nabla_{M_{k_g}(\boldsymbol{\theta}_g)} Q(M_g(\boldsymbol{\theta}_g)|\boldsymbol{\theta}_g) - \nabla_{M_{k_g}(\boldsymbol{\theta}_g)} Q(M_g(\boldsymbol{\theta}_g)|\boldsymbol{\theta}_g^*)||_2^2$$

$$\leqslant \sum_{k_g=1}^{K_g} \left[ \frac{(1 + \pi_g')\beta_g'}{K_g} \sum_{j_g=1}^{K_g} ||\boldsymbol{\theta}_{j_g} - \boldsymbol{\theta}_{j_g}^*||_2 + \pi_g' \pi_{\max_g} ||\boldsymbol{\theta}_{k_g} - \boldsymbol{\theta}_{k_g}^*||_2 \right]^2$$

Expanding each term inside the square root results in

$$||\nabla Q(M_g(\boldsymbol{\theta}_g)|\boldsymbol{\theta}_g) - \nabla Q(M_g(\boldsymbol{\theta}_g)|\boldsymbol{\theta}_g^*)||_2^2 \tag{27a}$$

$$\leqslant \sum_{k_g=1}^{K_g} \left[ \frac{(1+\pi_g')^2 \beta_g'^2}{K_g^2} \left( \sum_{j_g=1}^{K_g} ||\boldsymbol{\theta}_{j_g} - \boldsymbol{\theta}_{j_g}^*||_2 \right)^2 + \right.$$
$$\left. 2\frac{(1+\pi_g')\beta_g'}{K_g} \pi_g' \pi_{\max_g} ||\boldsymbol{\theta}_{k_g} - \boldsymbol{\theta}_{k_g}^*||_2 \sum_{j_g=1}^{K_g} ||\boldsymbol{\theta}_{j_g} - \boldsymbol{\theta}_{j_g}^*||_2 + \pi'^2 \pi_{\max_g}^2 ||\boldsymbol{\theta}_{k_g} - \boldsymbol{\theta}_{k_g}^*||_2^2 \right] \tag{27b}$$

$$= \frac{(1+\pi_g')^2 \beta_g'^2}{K_g} \left( \sum_{j_g=1}^{K_g} ||\boldsymbol{\theta}_{j_g} - \boldsymbol{\theta}_{j_g}^*||_2 \right)^2 + $$
$$2\frac{(1+\pi_g')\beta_g'}{K_g} \pi_g' \pi_{\max_g} \sum_{k_g=1}^{K_g} ||\boldsymbol{\theta}_{k_g} - \boldsymbol{\theta}_{k_g}^*||_2 \sum_{j_g=1}^{K_g} ||\boldsymbol{\theta}_{j_g} - \boldsymbol{\theta}_{j_g}^*||_2 + \pi'^2 \pi_{\max_g}^2 \sum_{k_g=1}^{K_g} ||\boldsymbol{\theta}_{k_g} - \boldsymbol{\theta}_{k_g}^*||_2^2 \tag{27c}$$

$$= \frac{(1+\pi_g')^2 \beta_g'^2}{K_g} \left( \sum_{j_g=1}^{K_g} ||\boldsymbol{\theta}_{j_g} - \boldsymbol{\theta}_{j_g}^*||_2 \right)^2 + $$
$$2\frac{(1+\pi_g')\beta_g'}{K_g} \pi_g' \pi_{\max_g} \left( \sum_{j_g=1}^{K_g} ||\boldsymbol{\theta}_{j_g} - \boldsymbol{\theta}_{j_g}^*||_2 \right)^2 + \pi'^2 \pi_{\max_g}^2 ||\boldsymbol{\theta}_g - \boldsymbol{\theta}_g^*||_2^2 \tag{27d}$$

$$\leqslant \left[ (1+\pi_g')^2 \beta_g'^2 + 2(1+\pi_g')\beta_g' \pi_g' \pi_{\max_g} + \pi'^2 \pi_{\max_g}^2 \right] ||\boldsymbol{\theta}_g - \boldsymbol{\theta}_g^*||_2^2. \tag{27e}$$

To see how we obtain (27d), consider a vector $u$ of length $K_g$, all of whose entries are 1, and a vector $v$ of length $K_g$, with $k_g^{th}$ entry $||\boldsymbol{\theta}_{k_g} - \boldsymbol{\theta}_{k_g}^*||_2$. Now we can rely on the Cauchy-Schwarz inequality to write:

$$u^\top v = \sum_{j_g=1}^{K_g} ||\boldsymbol{\theta}_{j_g} - \boldsymbol{\theta}_{j_g}^*||_2$$
$$\leqslant ||u||_2 ||v||_2$$
$$= \sqrt{K_g} \sqrt{\sum_{k_g=1}^{K_g} ||\boldsymbol{\theta}_{k_g} - \boldsymbol{\theta}_{k_g}^*||_2^2}$$
$$= \sqrt{K_g} ||\boldsymbol{\theta}_g - \boldsymbol{\theta}_g^*||_2$$

Thus, this allows us to see that

$$\left( \sum_{j_g=1}^{K_g} ||\boldsymbol{\theta}_{j_g} - \boldsymbol{\theta}_{j_g}^*||_2 \right)^2 \leqslant K_g ||\boldsymbol{\theta}_g - \boldsymbol{\theta}_g^*||_2^2.$$

Therefore, we obtain our final result by taking the square root of both sides, resulting in the following:

$$||\nabla Q(M_g(\boldsymbol{\theta}_g)|\boldsymbol{\theta}_g) - \nabla Q(M_g(\boldsymbol{\theta}_g)|\boldsymbol{\theta}_g^*)||_2 \leqslant \underbrace{((1+\pi_g')\beta_g' + \pi_g'\pi_{\max_g})}_{\beta_g} ||\boldsymbol{\theta}_g - \boldsymbol{\theta}_g^*||_2$$

$$\square$$

**Theorem 7** (GMM M-Step Contractive Behavior). *Suppose all the conditions of Theorem 6 hold, the clusters are well-separated (i.e. $R_{min} \gg 1$), the data dimensionality $d \ll \frac{R_{min}^2}{16}$ is sufficiently large, and the radius $a_g$ of the contraction region at client g is such that*

$$a_g \leqslant c_{a_g} \left\{ \frac{R_{min}}{2} - \sqrt{d}\mathcal{O}\left( \sqrt{\log\left( \max\left\{ \frac{K_g^2 \kappa_g}{\frac{\pi_{min_g} - c_{\pi_g} \pi_{max_g}}{1 + c_{\pi_g}}}, R_{max}, \min\{d, K_g\} \right\}\right)}\right)\right\}$$

*for a sufficiently small constant $0 < c_{a_g} < 1$, and $0 < c_{\pi_g} < \frac{\pi_{min_g}}{\pi_{max_g}}$ defined in the proof. Then, the contraction parameter of the population M-step of the multi-component isotropic GMM is $\frac{\beta_g}{\lambda_g} < 1$.*

*Proof.* In order to guarantee that the population M-step is contractive for the GMM described, we must show that

$$\beta_g' < \frac{\pi_{\min_g} - \pi_g' \pi_{\max_g}}{1 + \pi_g'}. \tag{28}$$

We can plug $\beta_g'$ from the statement of Theorem 6 into inequality (28) and rearrange terms to obtain the following.

$$a_g \leqslant \frac{R_{\min}}{2} - \frac{2\sqrt{2}}{\sqrt[4]{\min\{d, K_g\}}} \sqrt{\log\left( \frac{K_g^2(2\kappa_g + 4)(2R_{\max} + \min\{d, K_g\})^2}{\frac{\pi_{\min_g} - \pi_g' \pi_{\max_g}}{1 + \pi_g'}}\right)}.$$

Subsequently, we can combine this upper bound on $a$ with the one presented in the statement of Theorem 6 to obtain the following.

$$a_g \leqslant \frac{R_{\min}}{2} - \max\left\{ \underbrace{\frac{2\sqrt{2}}{\sqrt[4]{\min\{d, K_g\}}} \sqrt{\log\left( \frac{K_g^2(2\kappa_g + 4)(2R_{\max} + \min\{d, K_g\})^2}{\frac{\pi_{\min_g} - \pi_g' \pi_{\max_g}}{1 + \pi_g'}}\right)}}_{A_1}, \right.$$
$$\left. \underbrace{\sqrt{\min\{d, K_g\}} \max\{4\sqrt{2[\log(R_{\min}/4)]_+}, 8\sqrt{3}\}}_{A_2} \right\}$$

Now, we derive an upper bound to the maximization term. In doing so, we begin by obtaining upper bounds for $A_1$ and $A_2$ as follows, considering $A_1$ first.

$$A_1 := \frac{2\sqrt{2}}{\sqrt[4]{\min\{d, K_g\}}} \sqrt{\log\left( \frac{K_g^2(2\kappa_g + 4)(2R_{\max} + \min\{d, K_g\})^2}{\frac{\pi_{\min_g} - \pi_g' \pi_{\max_g}}{1 + \pi_g'}}\right)}$$

$$\leqslant c \sqrt{\log\left( \frac{K_g^2(2\kappa_g + 4)(2R_{\max} + \min\{d, K_g\})^2}{\frac{\pi_{\min_g} - \pi_g' \pi_{\max_g}}{1 + \pi_g'}}\right)} \tag{29a}$$

$$\leqslant c \sqrt{\log\left( \frac{c_1 K_g^2 \kappa_g (2R_{\max} + \min\{d, K_g\})^2}{\frac{\pi_{\min_g} - \pi_g' \pi_{\max_g}}{1 + \pi_g'}}\right)} \tag{29b}$$

$$= c \sqrt{\log\left( \frac{c_1 K_g^2 \kappa_g}{\frac{\pi_{\min_g} - \pi_g' \pi_{\max_g}}{1 + \pi_g'}}\right) + 2\log\left(2R_{\max} + \min\{d, K_g\}\right)} \tag{29c}$$

$$\leqslant c \sqrt{\min\{d, K_g\}} \sqrt{\log\left( \frac{c_1 K_g^2 \kappa_g}{\frac{\pi_{\min_g} - \pi_g' \pi_{\max_g}}{1 + \pi_g'}}\right) + c_2 \log\left(c_3 R_{\max} + e + \min\{d, K_g\}\right)}, \tag{29d}$$

where (29a) follows by plugging in a constant $c$, and recalling that $\min\{d, K_g\} > 1$, (29b) is obtained by noting that $\kappa_g \geqslant 1$ and choosing $c_1 \geqslant 6$, and (29d) again uses the fact that $\min\{d, K_g\} > 1$, as well as the monotonicity of the $\log$ function, and plugging in constants $c_2, c_3 > 1$.

Now, we derive an upper bound to $A_2$ as follows.

$$A_2 := \sqrt{\min\{d, K_g\}} \max\{4\sqrt{2[\log(R_{\min}/4)]_+}, 8\sqrt{3}\}$$

$$= \sqrt{\min\{d, K_g\}} \max\{c_1\sqrt{[\log(R_{\min}/4)]_+}, c_2\} \tag{30a}$$

$$\leqslant \sqrt{\min\{d, K_g\}} \max\{c_1\sqrt{\log(R_{\max} + e)}, c_2\} \tag{30b}$$

$$\leqslant \sqrt{\min\{d, K_g\}} \max\{c_1\sqrt{\log(R_{\max} + e)}, c_2\sqrt{\log(R_{\max} + e)}\} \tag{30c}$$

$$\leqslant \sqrt{\min\{d, K_g\}}(c_1\sqrt{\log(R_{\max} + e)} + c_2\sqrt{\log(R_{\max} + e)}) \tag{30d}$$

$$= c'\sqrt{\min\{d, K_g\}}\sqrt{\log(R_{\max} + e)}, \tag{30e}$$

where we obtain (30a) by rewriting the constants as $c_1, c_2 > 1$, (30b) is obtained by noting that $R_{\max} \geqslant R_{\min} > \frac{R_{\min}}{4}$, and adding an $e$ term inside the log to ensure that it is greater than or equal to 1. This allows us to obtain (30c), which is then upper bounded in (30d) by noting that both of the terms inside the maximization are greater than 0. Finally, we obtain the final result by combining the constants $c_1, c_2$ into a new constant $c'$.

Given the bounds derived for $A_1$ and $A_2$, we can write the following

$$\max\{A_1, A_2\}$$

$$\leqslant \sqrt{\min\{d, K_g\}} \max\left\{ c\sqrt{\log\left(\frac{c_1 K_g^2 \kappa_g}{\frac{\pi_{\min g} - \pi'_g \pi_{\max g}}{1 + \pi'_g}}\right) + c_2 \log(c_3 R_{\max} + e + \min\{d, K_g\})}, \right.$$

$$\left. c'\sqrt{\log(R_{\max} + e)} \right\} \tag{31a}$$

$$\leqslant \sqrt{\min\{d, K_g\}}(c + c')\sqrt{\log\left(\frac{c_1 K_g^2 \kappa_g}{\frac{\pi_{\min g} - \pi'_g \pi_{\max g}}{1 + \pi'_g}}\right) + c_2 \log(c_3 R_{\max} + e + \min\{d, K_g\})} \tag{31b}$$

$$\leqslant \sqrt{\min\{d, K_g\}}(c + c')\sqrt{\log\left(\frac{c_1 K_g^2 \kappa_g}{\frac{\pi_{\min g} - \pi'_g \pi_{\max g}}{1 + \pi'_g}}\right) + c_2 \log(c_3 R_{\max} + e) + c_2 \log(\min\{d, K_g\})} \tag{31c}$$

$$\leqslant \sqrt{\min\{d, K_g\}}(c + c')\sqrt{c_2 \log\left(3\max\left\{\frac{c_1 K_g^2 \kappa_g}{\frac{\pi_{\min g} - \pi'_g \pi_{\max g}}{1 + \pi'_g}}, c_3 R_{\max} + e, \min\{d, K_g\}\right\}\right)} \tag{31d}$$

$$\leqslant \sqrt{\min\{d, K_g\}}\mathcal{O}\left(\sqrt{\log\left(\max\left\{\frac{K_g^2 \kappa_g}{\frac{\pi_{\min g} - \pi'_g \pi_{\max g}}{1 + \pi'_g}}, R_{\max}, \min\{d, K_g\}\right\}\right)}\right), \tag{31e}$$

where (31b) follows by absorbing $A_2$ into $A_1$, (31c) is obtained by noting that each of the three terms within $\log$ functions are greater than 1, and therefore the log of their sum is upper bounded by the sum of their logs. Finally, (31e) is obtained by eliminating all constants. Therefore, we obtain

the following condition on $a_g$.

$$a_g \leqslant \frac{R_{\min}}{2} - \sqrt{\min\{d, K_g\}} \mathcal{O}\left(\sqrt{\log\left(\max\left\{\frac{K_g^2 \kappa_g}{\frac{\pi_{\min_g} - \pi_g' \pi_{\max_g}}{1 + \pi_g'}}, R_{\max}, \min\{d, K_g\}\right\}\right)}\right).$$

Now, recall that $\pi_g'$ (defined in the proof of Theorem 6) depends on $a_g$. However, we note that under the conditions of this theorem (i.e. $R_{\min} \gg 1$, sufficiently large $d \ll \frac{R_{\min}^2}{16}$), $\pi_g'$ approaches 0. This is true for any $a_g < c_{a_g}(\frac{R_{\min}}{2} - 2\sqrt{d})$ for some sufficiently small constant $0 < c_{a_g} < 1$ as long as $r_g$ (from the proof of Theorem 6) is set to its lower bound of $2\sqrt{d}$. Therefore, we rewrite our upper bound to ensure the condition on $a_g$ is satisfied as follows

$$a_g \leqslant c_{a_g}\left\{\frac{R_{\min}}{2} - 2\sqrt{d}\mathcal{O}\left(\sqrt{\log\left(\max\left\{\frac{K_g^2 \kappa_g}{\frac{\pi_{\min_g} - c_{\pi_g} \pi_{\max_g}}{1 + c_{\pi_g}}}, R_{\max}, \min\{d, K_g\}\right\}\right)}\right)\right\},$$

where $0 < c_{\pi_g} < \frac{\pi_{\min_g}}{\pi_{\max_g}}$ is a constant equivalent to the value of $\pi_g'$ evaluated at $\hat{a}_g = c_{a_g}(\frac{R_{\min}}{2} - 2\sqrt{d})$. We note that there must exist a $c_{a_g}$ such that the condition on $c_{\pi_g}$ is satisfied. To see this, note that with $c_{a_g} = 0$, $c_{\pi_g} < \frac{\pi_{\min_g}}{\pi_{\max_g}}$ holds trivially. This is because, as mentioned previously, $\pi_g'$ is close to 0 under the conditions of this theorem due to the exponential terms. Additionally, we observe that $\pi_g'$ is monotonic and continuous in $a_g$, and therefore is also monotonic and continuous in $c_{a_g}$. Thus, there must exist some $c_{a_g} > 0$ such that $c_{\pi_g} < \frac{\pi_{\min_g}}{\pi_{\max_g}}$. Finally, $c_{\pi_g} > 0$ holds trivially by definition of $\pi_g'$. This concludes the proof. $\qquad\square$

**Theorem 8** (GMM Finite-Sample and Population M-Step Distance). *Suppose all the conditions and definitions of Theorems 6 and 7 hold. Let* $\hat{\omega}(N_g) = \tilde{\mathcal{O}}\left(\max\left\{N_g^{-\frac{1}{2}} K_g^3 (1 + R_{max})^3 \sqrt{d} \max\{1, \log(\kappa_g)\}, (1 + R_{max})\frac{d}{\sqrt{N_g}}\right\}\right).$ *Moreover, let* $M_{k_g}(\boldsymbol{\theta}_g)$ *and* $\widehat{M}_{k_g}(\boldsymbol{\theta}_g)$ *denote the population and finite-sample M-step maps associated with the GMM described in this section, respectively. Then, we have that*

$$\sup_{\boldsymbol{\theta}_g \in \mathbb{A}_g} \left\| M_{k_g}(\boldsymbol{\theta}_g) - \widehat{M}_{k_g}(\boldsymbol{\theta}_g) \right\|_2 \leqslant \frac{1}{\hat{\tau}_{N_g}}\left(\hat{\omega}(N_g) + 2a_g\sqrt{\frac{1}{2N_g}\log\left(\frac{2}{\eta_g}\right)}\right),$$

*with probability at least* $(1 - \exp(-cd\log N_g))(1 - \eta_g)$, *where $c$ is some positive constant,* $\mathbb{A}_g = \prod_{k_g=1}^{K_g} \mathbb{B}_2(\boldsymbol{\theta}_{k_g}^*; a_g)$, *and*

$$\hat{\tau}_{N_g} \geqslant \frac{\pi_{min_g}\left[1 - \exp\left(-\frac{r_g\sqrt{d}}{2}\right)\right]}{1 + \frac{1 - \pi_{min_g}}{\pi_{max_g}}\exp\left[-\frac{1}{2}(R_{min}^2 - 2R_{min}(r_g + a_g))\right]} - \sqrt{\frac{1}{2N_g}\log\left(\frac{2}{\eta_g}\right)}.$$

*Proof.* Recall that the finite-sample expected complete-data log-likelihood function associated with out GMM model can be expressed as follows.

$$\widehat{Q}_g(\boldsymbol{\theta}_g|\boldsymbol{\theta}_g')) = \frac{1}{N_g}\sum_{n_g=1}^{N_g}\sum_{k_g=1}^{K_g}\gamma_{k_g}(\widehat{\boldsymbol{x}}_{n_g}, \boldsymbol{\theta}_g')(\log\pi_{k_g} + \log\phi(\widehat{\boldsymbol{x}}_{n_g}|\boldsymbol{\theta}_{k_g})),$$

Moreover, its gradient and Hessian can be written as follows.

$$\nabla_{\boldsymbol{\theta}_{k_g}}\widehat{Q}_g(\boldsymbol{\theta}_g|\boldsymbol{\theta}_g')) = \frac{1}{N_g}\sum_{n_g=1}^{N_g}\gamma_{k_g}(\widehat{\boldsymbol{x}}_{n_g}, \boldsymbol{\theta}_g')(\widehat{\boldsymbol{x}}_{n_g} - \boldsymbol{\theta}_{k_g}). \tag{32a}$$

$$\nabla_{\boldsymbol{\theta}_{k_g}}^2\widehat{Q}_g(\boldsymbol{\theta}_g|\boldsymbol{\theta}_g')) = -\frac{1}{N_g}\sum_{n_g=1}^{N_g}\gamma_{k_g}(\widehat{\boldsymbol{x}}_{n_g}, \boldsymbol{\theta}_g')I_d. \tag{32b}$$

Now, observe that the Hessian is a diagonal matrix whose eigenvalues are all equal to the empirical expectation of the responsibility function $\gamma_{k_g}(\widehat{\boldsymbol{x}}_{n_g}, \boldsymbol{\theta}'_g)$. Therefore, the function is strongly concave, with strong concavity parameter $\widehat{\tau}_{N_g}$, which we define explicitly later.

Thus, by the strong concavity of the $\widehat{Q}_g(\boldsymbol{\theta}_g | \boldsymbol{\theta}'_g))$ function we can write the following.

$$
\frac{\widehat{\tau}_{N_g}}{2} \left\| M_{k_g}(\boldsymbol{\theta}_g) - \widehat{M}_{k_g}(\boldsymbol{\theta}_g) \right\|_2^2
$$
$$
\leqslant \widehat{Q}_g(\widehat{M}_{k_g}(\boldsymbol{\theta}_g) | \boldsymbol{\theta}_g) - \widehat{Q}_g(M_{k_g}(\boldsymbol{\theta}_g) | \boldsymbol{\theta}_g) + \nabla_{\boldsymbol{\theta}_{k_g}} \widehat{Q}_g(\widehat{M}_{k_g}(\boldsymbol{\theta}_g) | \boldsymbol{\theta}_g)^\top (M_{k_g}(\boldsymbol{\theta}_g) - \widehat{M}_{k_g}(\boldsymbol{\theta}_g))
\tag{33a}
$$
$$
= \widehat{Q}_g(\widehat{M}_{k_g}(\boldsymbol{\theta}_g) | \boldsymbol{\theta}_g) - \widehat{Q}_g(M_{k_g}(\boldsymbol{\theta}_g) | \boldsymbol{\theta}_g).
\tag{33b}
$$

Similarly, we can use the strong concavity to write the following.

$$
\frac{\widehat{\tau}_{N_g}}{2} \left\| M_{k_g}(\boldsymbol{\theta}_g) - \widehat{M}_{k_g}(\boldsymbol{\theta}_g) \right\|_2^2
$$
$$
\leqslant \widehat{Q}_g(M_{k_g}(\boldsymbol{\theta}_g) | \boldsymbol{\theta}_g) - \widehat{Q}_g(\widehat{M}_{k_g}(\boldsymbol{\theta}_g) | \boldsymbol{\theta}_g) + \nabla_{\boldsymbol{\theta}_{k_g}} \widehat{Q}_g(M_{k_g}(\boldsymbol{\theta}_g) | \boldsymbol{\theta}_g)^\top (\widehat{M}_{k_g}(\boldsymbol{\theta}_g) - M_{k_g}(\boldsymbol{\theta}_g))
\tag{34}
$$

Summing up inequalities (33b) and (34), we can obtain the following.

$$
\widehat{\tau}_{N_g} \left\| M_{k_g}(\boldsymbol{\theta}_g) - \widehat{M}_{k_g}(\boldsymbol{\theta}_g) \right\|_2^2 \leqslant \nabla_{\boldsymbol{\theta}_{k_g}} \widehat{Q}_g(M_{k_g}(\boldsymbol{\theta}_g) | \boldsymbol{\theta}_g)^\top (\widehat{M}_{k_g}(\boldsymbol{\theta}_g) - M_{k_g}(\boldsymbol{\theta}_g))
\tag{35a}
$$
$$
\leqslant \left\| \nabla_{\boldsymbol{\theta}_{k_g}} \widehat{Q}_g(M_{k_g}(\boldsymbol{\theta}_g) | \boldsymbol{\theta}_g) \right\|_2 \left\| M_{k_g}(\boldsymbol{\theta}_g) - \widehat{M}_{k_g}(\boldsymbol{\theta}_g) \right\|_2
\tag{35b}
$$
$$
= \left\| \nabla_{\boldsymbol{\theta}_{k_g}} \widehat{Q}_g(M_{k_g}(\boldsymbol{\theta}_g) | \boldsymbol{\theta}_g) - \nabla_{\boldsymbol{\theta}_{k_g}} Q_g(M_{k_g}(\boldsymbol{\theta}_g) | \boldsymbol{\theta}_g) \right\|_2
$$
$$
\cdot \left\| M_{k_g}(\boldsymbol{\theta}_g) - \widehat{M}_{k_g}(\boldsymbol{\theta}_g) \right\|_2,
\tag{35c}
$$

where (35b) is obtained via the Cauchy-Schwarz inequality, and (35c) follows from the fact that $\nabla_{\boldsymbol{\theta}_{k_g}} Q_g(M_{k_g}(\boldsymbol{\theta}_g) | \boldsymbol{\theta}_g) = 0$ as it is evaluated at its maximizer.

Now note that if $M^{(k)}(\boldsymbol{\theta}'_g) = M_N^{(k)}(\boldsymbol{\theta}'_g)$, then the finite sample EM algorithm converges trivially by the convergence of the population EM algorithm. Thus, we focus on the cases where $M^{(k)}(\boldsymbol{\theta}'_g) \neq M_N^{(k)}(\boldsymbol{\theta}'_g)$. In this case, we can write the inequality in (35c) as follows.

$$
\widehat{\tau}_{N_g} \left\| M_{k_g}(\boldsymbol{\theta}_g) - \widehat{M}_{k_g}(\boldsymbol{\theta}_g) \right\|_2
$$
$$
\leqslant \left\| \nabla_{\boldsymbol{\theta}_{k_g}} \widehat{Q}_g(M_{k_g}(\boldsymbol{\theta}_g) | \boldsymbol{\theta}_g) - \nabla_{\boldsymbol{\theta}_{k_g}} Q_g(M_{k_g}(\boldsymbol{\theta}_g) | \boldsymbol{\theta}_g) \right\|_2
\tag{36a}
$$
$$
= \left\| \nabla_{\boldsymbol{\theta}_{k_g}} \widehat{Q}_g(\boldsymbol{\theta}_g | \boldsymbol{\theta}_g) + \nabla_{\boldsymbol{\theta}_{k_g}}^2 \widehat{Q}_g(\boldsymbol{\theta}_g | \boldsymbol{\theta}_g)(M_{k_g}(\boldsymbol{\theta}) - \boldsymbol{\theta}_{k_g}) \right.
$$
$$
\left. - \nabla_{\boldsymbol{\theta}_{k_g}} Q_g(\boldsymbol{\theta}_g | \boldsymbol{\theta}_g) - \nabla_{\boldsymbol{\theta}_{k_g}}^2 Q_g(\boldsymbol{\theta}_g | \boldsymbol{\theta}_g)(M_{k_g}(\boldsymbol{\theta}) - \boldsymbol{\theta}_{k_g}) \right\|_2
\tag{36b}
$$
$$
\leqslant \left\| \nabla_{\boldsymbol{\theta}_{k_g}} \widehat{Q}_g(\boldsymbol{\theta}_g | \boldsymbol{\theta}_g) - \nabla_{\boldsymbol{\theta}_{k_g}} Q_g(\boldsymbol{\theta}_g | \boldsymbol{\theta}_g) \right\|_2
$$
$$
+ \left\| \nabla_{\boldsymbol{\theta}_{k_g}}^2 Q_g(\boldsymbol{\theta}_g | \boldsymbol{\theta}_g)(M_{k_g}(\boldsymbol{\theta}) - \boldsymbol{\theta}_{k_g}) - \nabla_{\boldsymbol{\theta}_{k_g}}^2 Q_N(\theta | \theta)(M_{k_g}(\boldsymbol{\theta}) - \boldsymbol{\theta}_{k_g}) \right\|_2
\tag{36c}
$$
$$
= \left\| \nabla_{\boldsymbol{\theta}_{k_g}} \widehat{Q}_g(\boldsymbol{\theta}_g | \boldsymbol{\theta}_g) - \nabla_{\boldsymbol{\theta}_{k_g}} Q_g(\boldsymbol{\theta}_g | \boldsymbol{\theta}_g) \right\|_2
$$
$$
+ \left| \frac{1}{N_g} \sum_{n_g=1}^{N_g} \gamma_{k_g}(\widehat{\boldsymbol{x}}_{n_g}, \boldsymbol{\theta}_g)) - \mathbb{E}_{\boldsymbol{x}}[\gamma_{k_g}(\boldsymbol{x}, \boldsymbol{\theta}_g))] \right| \left\| M_{k_g}(\boldsymbol{\theta}_g) - \boldsymbol{\theta}_{k_g} \right\|_2
\tag{36d}
$$
$$
\leqslant \left\| \nabla_{\boldsymbol{\theta}_{k_g}} \widehat{Q}_g(\boldsymbol{\theta}_g | \boldsymbol{\theta}_g) - \nabla_{\boldsymbol{\theta}_{k_g}} Q_g(\boldsymbol{\theta}_g | \boldsymbol{\theta}_g) \right\|_2 + 2 \left| \frac{1}{N_g} \sum_{n_g=1}^{N_g} \gamma_{k_g}(\widehat{\boldsymbol{x}}_{n_g}, \boldsymbol{\theta}_g)) - \mathbb{E}_{\boldsymbol{x}}[\gamma_{k_g}(\boldsymbol{x}, \boldsymbol{\theta}_g))] \right| a_g,
\tag{36e}
$$

where (36b) follows by writing the Taylor series expansion of the function $f \colon \mathbb{R}^d \to \mathbb{R}^d$, $f(\boldsymbol{\theta}_g) = \nabla_{\boldsymbol{\theta}_{k_g}} Q_g(\boldsymbol{\theta}_g | \boldsymbol{\theta}_g)$, (36c) follows from the triangle inequality, and (36d) uses the fact that the Hessian of both the population and finite sample functions is diagonal with identical entries. Finally, (36e) leverages the fact that $M_{k_g}(\boldsymbol{\theta}), \boldsymbol{\theta}_{k_g} \in \mathbb{B}_2(\boldsymbol{\theta}_{k_g}^*, a_g)$. Now, suppose we have a training set $\widehat{\boldsymbol{X}}$ comprised of $N_g$ IID samples. Then we can write the following.

$$
P\left( \left| \sum_{n_g=1}^{N_g} \gamma_{k_g}(\widehat{\boldsymbol{x}}_{n_g}, \boldsymbol{\theta}_g)) - \mathbb{E}_{\boldsymbol{x}}\left[ \sum_{n_g=1}^{N_g} \gamma_{k_g}(\widehat{\boldsymbol{x}}_{n_g}, \boldsymbol{\theta}_g)) \right] \right| \le N_g \zeta_g \right)
$$

$$
= P\left( \left| \sum_{n_g=1}^{N_g} \gamma_{k_g}(\widehat{\boldsymbol{x}}_{n_g}, \boldsymbol{\theta}_g)) - \sum_{n_g=1}^{N_g} \mathbb{E}_{\boldsymbol{x}}\left[ \gamma_{k_g}(\boldsymbol{x}, \boldsymbol{\theta}_g)) \right] \right| \le N_g \zeta_g \right)
$$

$$
= P\left( \left| \sum_{n_g=1}^{N_g} \gamma_{k_g}(\widehat{\boldsymbol{x}}_{n_g}, \boldsymbol{\theta}_g)) - N_g \mathbb{E}_{\boldsymbol{x}}\left[ \gamma_{k_g}(\boldsymbol{x}, \boldsymbol{\theta}_g)) \right] \right| \le N_g \zeta_g \right)
$$

$$
= P\left( N_g \left| \frac{1}{N_g} \sum_{n_g=1}^{N_g} \gamma_{k_g}(\widehat{\boldsymbol{x}}_{n_g}, \boldsymbol{\theta}_g)) - \mathbb{E}_{\boldsymbol{x}}\left[ \gamma_{k_g}(\boldsymbol{x}, \boldsymbol{\theta}_g)) \right] \right| \le N_g \zeta_g \right)
$$

$$
= P\left( \left| \frac{1}{N_g} \sum_{n_g=1}^{N_g} \gamma_{k_g}(\widehat{\boldsymbol{x}}_{n_g}, \boldsymbol{\theta}_g)) - \mathbb{E}_{\boldsymbol{x}}\left[ \gamma_{k_g}(\boldsymbol{x}, \boldsymbol{\theta}_g)) \right] \right| \le \zeta_g \right),
$$

for some small $\zeta_g \in \mathbb{R}^+$. Moreover, by Hoeffding's inequality we have that

$$
P\left( \left| \sum_{n_g=1}^{N_g} \gamma_{k_g}(\widehat{\boldsymbol{x}}_{n_g}, \boldsymbol{\theta}_g)) - \mathbb{E}_{\boldsymbol{x}}\left[ \sum_{n_g=1}^{N_g} \gamma_{k_g}(\widehat{\boldsymbol{x}}_{n_g}, \boldsymbol{\theta}_g)) \right] \right| \le N_g \zeta_g \right) \ge 1 - 2\exp\left( -2N_g \zeta_g^2 \right).
$$

Therefore, this allows us to write the following.

$$
P\left( \left| \frac{1}{N_g} \sum_{n_g=1}^{N_g} \gamma_{k_g}(\widehat{\boldsymbol{x}}_{n_g}, \boldsymbol{\theta}_g)) - \mathbb{E}_{\boldsymbol{x}}\left[ \gamma_{k_g}(\boldsymbol{x}, \boldsymbol{\theta}_g)) \right] \right| \le \zeta_g \right) \ge 1 - 2\exp\left( -2N_g \zeta_g^2 \right).
$$

Now suppose we set $\eta_g = 2\exp\left( -2N_g \zeta_g^2 \right)$, then we can obtain that with probability at least $1 - \eta_g$

$$
\left| \frac{1}{N_g} \sum_{n_g=1}^{N_g} \gamma_{k_g}(\widehat{\boldsymbol{x}}_{n_g}, \boldsymbol{\theta}_g)) - \mathbb{E}_{\boldsymbol{x}}\left[ \gamma_{k_g}(\boldsymbol{x}, \boldsymbol{\theta}_g)) \right] \right| \le \sqrt{\frac{1}{2N_g} \log\left( \frac{2}{\eta_g} \right)} \tag{37}
$$

Now, let us define $\mathbb{A}_g$ as the contraction region $\prod_{k_g=1}^{K_g} \mathbb{B}_2(\boldsymbol{\theta}_{k_g}^*, a_g)$. Armed with the previous probabilistic result, we can say that with probability at least $(1 - \exp(-cd \log N_g))(1 - \eta_g)$.

$$
\sup_{\theta \in \mathbb{A}} \left\| M_{k_g}(\boldsymbol{\theta}_g) - \widehat{M}_{k_g}(\boldsymbol{\theta}_g) \right\|_2 \le \frac{1}{\widehat{\tau}_{N_g}} \left( \widehat{\omega}^{unif}(N_g) + 2a_g \sqrt{\frac{1}{2N_g} \log\left( \frac{2}{\eta_g} \right)} \right), \tag{38}
$$

where $\widehat{\omega}^{unif}(N_g)$ is defined in the theorem statement, and (38) follows directly from Theorem 5 in (Yan et al., 2017) which states that for our problem setting, $\sup_{\theta \in \mathbb{A}} \left\| \nabla_{\boldsymbol{\theta}_{k_g}} \widehat{Q}_g(\boldsymbol{\theta}_g | \boldsymbol{\theta}_g) - \nabla_{\boldsymbol{\theta}_{k_g}} Q_g(\boldsymbol{\theta}_g | \boldsymbol{\theta}_g) \right\|_2 \le \widehat{\omega}^{unif}(N_g)$ with probability at least $1 - \exp(-cd \log N_g)$, where $c$ is some positive constant. Finally, note that under inequality (37), we can say that with probability $1 - \eta_g$.

$$
\frac{1}{N_g} \sum_{n_g=1}^{N_g} \gamma_{k_g}(\widehat{\boldsymbol{x}}_{n_g}, \boldsymbol{\theta}_g)) \ge \mathbb{E}_{\boldsymbol{x}}\left[ \gamma_{k_g}(\boldsymbol{x}, \boldsymbol{\theta}_g)) \right] - \sqrt{\frac{1}{2N_g} \log\left( \frac{2}{\eta_g} \right)}
$$

$$
\Rightarrow \widehat{\tau}_{N_g} \ge \frac{\pi_{\min_g}\left[ 1 - \exp\left( -\frac{r_g \sqrt{d}}{2} \right) \right]}{1 + \frac{1 - \pi_{\min_g}}{\pi_{\max_g}} \exp\left[ -\frac{1}{2}(R_{\min}^2 - 2R_{\min}(r_g + a_g)) \right]} - \sqrt{\frac{1}{2N_g} \log\left( \frac{2}{\eta_g} \right)},
$$

which utilizes the lower bound we obtain on $\mathbb{E}_{\boldsymbol{x}}\left[\gamma_{k_g}(\boldsymbol{x},\boldsymbol{\theta}_g))\right]$ in the proof of Theorem 6. This concludes the proof.

$\square$

### B.4 Preliminary Differential Privacy Discussion for Isotropic GMMs

We study a potential method for the clients to enhance the privacy of their data. More specifically, we consider the use of differential privacy (DP). In doing so, client $g$ adds Gaussian noise to the estimated centroid $\boldsymbol{\theta}_{k_g}^{(t)}$ of local cluster $k_g$ at global iteration $t$. Next, we define various fundamental concepts in DP, first introduced in Dwork & Roth (2014).

**Definition 2** (Differential Privacy). A randomized algorithm $\mathcal{C}$ is said to be $(\rho,\mu)$-differentially private if for all $\mathcal{S} \subseteq \text{Range}(\mathcal{C})$ and any two neighboring datasets $\boldsymbol{X}$ and $\boldsymbol{X}'$ of the same size but differing only in one sample we have that $P\left(\mathcal{C}(\boldsymbol{X}) \in \mathcal{S}\right) \leqslant \exp(\rho)P\left(\mathcal{C}(\boldsymbol{X}') \in \mathcal{S}\right) + \mu$.

**Definition 3** ($\ell_2$-Sensitivity). The $\ell_2$-sensitivity $\Delta_2(f)$ of a function $f : \mathcal{X} \to \mathbb{R}^D$ is defined as $\Delta_2(f) := \max_{\boldsymbol{X},\boldsymbol{X}'} ||f(\boldsymbol{X}) - f(\boldsymbol{X}')||_2$, where $\boldsymbol{X}$ and $\boldsymbol{X}'$ are datasets of the same size, differing only in one sample.

**Definition 4** (DP via Gaussian Noise). Given a function $f : \mathcal{X} \to \mathbb{R}^D$ with $\ell_2$-sensitivity $\Delta_2(f)$, then $\mathcal{C}(\boldsymbol{X}) = f(\boldsymbol{X}) + \mathcal{N}(0,\sigma^2 I_D)$ is said to be $(\rho,\mu)-$differentially private if $\sigma \geqslant \frac{\Delta_2(f)\sqrt{2\log(\frac{1.25}{\mu})}}{\rho}$.

Now, consider the map $\widehat{M}_g(\boldsymbol{\theta}'_g) : \mathbb{R}^{dK_g} \to \mathbb{R}^{dK_g}$, which maps the current vectorized centroid estimates $\boldsymbol{\theta}'_g$ at client $g$ to the vectorized maximizer associated with all clusters. In order to guarantee differential privacy, each client $g$ independently perturbs its local $\widehat{M}_g(\boldsymbol{\theta}'_g)$ before sharing it with the server. In other words, client $g$ shares a perturbed $\widetilde{M}_g(\boldsymbol{\theta}'_g) = \widehat{M}_g(\boldsymbol{\theta}'_g) + \mathcal{N}(0,\sigma^2 I_{dK_g})$. Next, we present an assumption on the support of the data, followed by Theorem 9 where we establish the required standard deviation of the Gaussian noise to ensure that this map is differentially private.

**Assumption 7** (Bounded Support). Any sample $\widehat{\boldsymbol{x}}_{n_g}$ at client $g \in [G]$ is such that $||\widehat{\boldsymbol{x}}_{n_g}||_2 \leqslant B_{\boldsymbol{x}} \in \mathbb{R}$.

*Remark* 3. Note that Assumption 7 is not restrictive in practice. This is because data is often collected via sensors or other methods with known ranges, and can therefore be readily normalized.

**Theorem 9** (Client-to-Server Communication DP). *The perturbed estimate $\widetilde{M}_g(\boldsymbol{\theta}'_g) = \widehat{M}_g(\boldsymbol{\theta}'_g) + \mathcal{N}(0,\sigma^2 I_{dK_g})$ sent by client $g$ to the server at global iteration $t$ of the algorithm is guaranteed to be $(\rho,\mu)$-differentially private if the standard deviation of the noise satisfies*

$$\sigma \geqslant \frac{\sqrt{K_g}\left[\frac{3B_{\boldsymbol{x}}}{B_{\gamma_g}} + \frac{2B_{\boldsymbol{x}}}{B_{\gamma_g}^2}\right]\sqrt{2\log(\frac{1.25}{\mu})}}{\rho},$$

*where $B_{\boldsymbol{x}}$ is defined in Assumption 7, and $B_{\gamma_g}$ is a constant depending on client-level parameters such as $a_g$, $\pi_{min_g}$, and $\pi_{max_g}$, and is explained in the proof.*

*Proof.* In order to establish the differential privacy of the map $\widehat{M}_g(\boldsymbol{\theta}'_g)$ via Gaussian noise, it suffices to derive its $\ell_2$-sensitivity $\Delta_2(\widehat{M}_g(\boldsymbol{\theta}'_g))$. We begin by considering the maximizer $\widehat{M}_{k_g}(\boldsymbol{\theta}'_g)$ associated with cluster $k_g$. Now, consider the two datasets $\mathcal{X}_g$ and $\mathcal{X}'_g$, both of which have $N_g$ samples. Without loss of generality, suppose the datasets differ only in their last sample. That is, the first $N_g - 1$ samples are identical across the two datasets, whereas the $N_g^{th}$ samples in $\mathcal{X}_g$ and $\mathcal{X}'_g$ are $\widehat{\boldsymbol{x}}^*$ and $\widehat{\boldsymbol{x}}'$, respectively, with $\widehat{\boldsymbol{x}}^* \neq \widehat{\boldsymbol{x}}'$. To simplify notation and to highlight the datasets as the input of interest in the mapping function, let us define the following.

$$\boldsymbol{s}_{k_g} := \sum_{n_g=1}^{N_g-1} \gamma_{k_g}(\widehat{\boldsymbol{x}}_{n_g},\boldsymbol{\theta}'_g)\widehat{\boldsymbol{x}}_{n_g}, \qquad w_{k_g} := \sum_{n_g=1}^{N_g-1} \gamma_{k_g}(\widehat{\boldsymbol{x}}_{n_g},\boldsymbol{\theta}'_g),$$

$$\gamma^* := \gamma_{k_g}(\widehat{\boldsymbol{x}}^*,\boldsymbol{\theta}'_g), \qquad \gamma' := \gamma_{k_g}(\widehat{\boldsymbol{x}}',\boldsymbol{\theta}'_g), \tag{39}$$

$$f_g(\boldsymbol{\mathcal{X}}) := \widehat{M}_g(\boldsymbol{\theta}'_g), \qquad\qquad f_{k_g}(\boldsymbol{\mathcal{X}}) := \widehat{M}_{k_g}(\boldsymbol{\theta}'_g).$$

Moreover, assume without loss of generality that $\gamma' \geqslant \gamma^*$. Now, we analyze the $\ell_2$-sensitivity of $f_{k_g}(\boldsymbol{\mathcal{X}})$ as follows.

$$\|f_{k_g}(\boldsymbol{\mathcal{X}}) - f_{k_g}(\boldsymbol{\mathcal{X}}')\|_2$$

$$= \left\| \frac{\boldsymbol{s}_{k_g} + \gamma^*\widehat{\boldsymbol{x}}^*}{w_{k_g} + \gamma^*} - \frac{\boldsymbol{s}_{k_g} + \gamma'\widehat{\boldsymbol{x}}'}{w_{k_g} + \gamma'} \right\|_2 \tag{40}$$

$$= \left\| \frac{(w_{k_g} + \gamma')(\boldsymbol{s}_{k_g} + \gamma^*\widehat{\boldsymbol{x}}^*) - (w_{k_g} + \gamma^*)(\boldsymbol{s}_{k_g} + \gamma'\widehat{\boldsymbol{x}}')}{(w_{k_g} + \gamma^*)(w_{k_g} + \gamma')} \right\|_2 \tag{41}$$

$$= \frac{1}{(w_{k_g} + \gamma^*)(w_{k_g} + \gamma')} \left\|(w_{k_g} + \gamma')(\boldsymbol{s}_{k_g} + \gamma^*\widehat{\boldsymbol{x}}^*) - (w_{k_g} + \gamma^*)(\boldsymbol{s}_{k_g} + \gamma'\widehat{\boldsymbol{x}}')\right\|_2 \tag{42}$$

$$\leqslant \frac{1}{(w_{k_g} + \gamma^*)^2} \left\|(\gamma' - \gamma^*)\boldsymbol{s}_{k_g} + w_{k_g}\gamma^*\widehat{\boldsymbol{x}}^* - w_{k_g}\gamma'\widehat{\boldsymbol{x}}' + \gamma'\gamma^*(\widehat{\boldsymbol{x}}^* - \widehat{\boldsymbol{x}}')\right\|_2 \tag{43}$$

$$\leqslant \frac{1}{(w_{k_g} + \gamma^*)^2} \left[\|\boldsymbol{s}_{k_g}\|_2 + w_{k_g}\|\widehat{\boldsymbol{x}}^*\|_2 + w_{k_g}\|\widehat{\boldsymbol{x}}'\|_2 + \|\widehat{\boldsymbol{x}}^* - \widehat{\boldsymbol{x}}'\|_2\right] \tag{44}$$

$$\leqslant \frac{1}{(w_{k_g} + \gamma^*)^2} \left[\sum_{n_g=1}^{N_g-1} \gamma_{k_g}(\widehat{\boldsymbol{x}}_{n_g}, \boldsymbol{\theta}'_g)\|\widehat{\boldsymbol{x}}_{n_g}\|_2 + w_{k_g}\|\widehat{\boldsymbol{x}}^*\|_2 + w_{k_g}\|\widehat{\boldsymbol{x}}'\|_2 + \|\widehat{\boldsymbol{x}}^* - \widehat{\boldsymbol{x}}'\|_2\right] \tag{45}$$

$$\leqslant \frac{(3(w_{k_g} + \gamma^*) + 2)B_{\boldsymbol{x}}}{(w_{k_g} + \gamma^*)^2} \tag{46}$$

$$\leqslant \frac{3B_{\boldsymbol{x}}}{B_{\gamma_g}} + \frac{2B_{\boldsymbol{x}}}{B_{\gamma_g}^2}, \tag{47}$$

where 40 follows from the definition of $\widehat{M}_{k_g}(\boldsymbol{\theta}'_g)$, 43 follows from the assumption that $\gamma' \geqslant \gamma^*$, 44 follows from the fact that $0 \leqslant \gamma^* \leqslant 1$ and $0 \leqslant \gamma' \leqslant 1$, 45 follows from the fact that the norm of a sum is upper bounded by the sum of the norms of the individual terms, and finally, 46 follows from Assumption 7, and the fact that $\gamma^*$ is positive. Note that we denote the lower bound of $(w_{k_g} + \gamma^*)$ by $B_{\gamma_g}$, which depends only on parameters specific to client $g$ but not to cluster $k_g$. This lower bound can be taken as $\widehat{\tau}_{N_g}$, which is derived in the proof of Theorem 8. Now that we have the $\ell_2$-sensitivity of the maximizer of a single cluster, we obtain the overall $\ell_2$-sensitivity of the map over all clusters as follows.

$$\|f_g(\boldsymbol{\mathcal{X}}) - f_g(\boldsymbol{\mathcal{X}}')\|_2 = \sqrt{\sum_{k_g}^{K_g} \|f_{k_g}(\boldsymbol{\mathcal{X}}) - f_{k_g}(\boldsymbol{\mathcal{X}}')\|_2^2}$$

$$\leqslant \sqrt{K_g \left[\frac{3B_{\boldsymbol{x}}}{B_{\gamma_g}} + \frac{2B_{\boldsymbol{x}}}{B_{\gamma_g}^2}\right]^2}$$

$$= \sqrt{K_g}\left[\frac{3B_{\boldsymbol{x}}}{B_{\gamma_g}} + \frac{2B_{\boldsymbol{x}}}{B_{\gamma_g}^2}\right].$$

This concludes the proof. $\qquad\qquad\square$

*Remark* 4. Note that we can readily obtain an upper bound on the distance between the population $M_{k_g}(\boldsymbol{\theta_g}^{(t)})$ and the perturbed, finite-sample $\widetilde{M}_{k_g}(\boldsymbol{\theta_g}^{(t)})$. Therefore, by the same argument in Theorem 2 we can argue that iterates $\widetilde{M}_{k_g}(\boldsymbol{\theta_g}^{(t)})$ converge to the neighborhood of the ground truth parameters $\boldsymbol{\theta}^*_{k_g}$. However, the iterates do not necessarily converge to a single point within the neighborhood due to the randomness of the added noise. Further work should explore this convergence behavior more carefully, as well as the implications of DP on the privacy and convergence of the uncertainty set radius computations.

### B.5 Computational Efficiency and Communication Costs

#### B.5.1 Improving the Efficiency of Server Computations

While the pairwise server computations described in the work are intuitive, they can be inefficient for large-scale problems. To see this, consider a worst case where each client $g$ has a local number of clusters $K_g = \mathcal{O}(K)$. In this setting, the server would need to perform roughly $\mathcal{O}(G^2 K^2)$ operations, which can be very expensive for very large $G$ or $K$. To improve efficiency, the server may leverage a $d$-dimensional binary search tree (commonly known as KD tree) (Bentley, 1975). More specifically, the server would store the estimated cluster centroids $\widehat{M}_{k_g}(\boldsymbol{\theta}_g^{(t-1)})$ for all clusters $k_g \in [K_g]$ shared by clients $g \in [G]$ in the tree. Subsequently, the server would iterate over each centroid $\widehat{M}_{k_g}(\boldsymbol{\theta}_g^{(t-1)})$, obtain its $M$ nearest neighbors (by slight abuse of notation), then check for uncertainty set overlaps and perform aggregation as described in Section 4. The construction of the tree incurs a cost of $\mathcal{O}(GK \log GK)$ (Friedman et al., 1977), whereas a single nearest neighbor search incurs an expected cost close to $\mathcal{O}(\log GK)$ (Friedman et al., 1977) in practice. Therefore, assuming that the number of overlaps between uncertainty sets is significantly smaller than the total number of available uncertainty sets (that is $M \ll GK$), then we have that the total cost of constructing and using the binary search tree would be close to $\mathcal{O}(2GK \log GK) = \mathcal{O}(GK \log GK)$ in practice. This approach can improve the efficiency of server computations without impacting any other aspects of the algorithm.

#### B.5.2 A Note on Communication Costs

During each communication round of our algorithm, each client $g$ sends $K_g$ arrays of size $d$ and $K_g$ scalars to the central server, and receives $K_g$ arrays of size $d$. This results in a per-round total communication cost of approximately $2dG\tilde{K}_g + G\tilde{K}_g \leqslant 3dG\tilde{K}_g$, where $\tilde{K}_g$ is the mean number of clusters per client. We compare this to the communication cost of AFCL (Zhang et al., 2025). Due to its asynchronous nature, we assume that only $10\%$ of the clients participate in each communication round (a favorable condition for AFCL). In AFCL, each active client sends $N_g$ arrays of size $d$ to the central server, and receives $\widehat{K}$ arrays of size $d$, where $\widehat{K}$ is the estimated number of clusters. Under the assumption of roughly balanced client sample sizes, it is clear that the total per-round communication cost is approximately $0.1dG\tilde{N}_g + 0.1dG\widehat{K} > 0.1dG\tilde{N}_g$, where $\tilde{N}_g$ is the mean number of samples per client. Thus, our algorithm enjoys a lower per-round communication cost, since $\tilde{N}_g \gg 30\tilde{K}_g$ in most practical applications.

Furthermore, we theoretically prove in Theorems 2 and 5 that our algorithm achieves a linear convergence rate for all clusters at all clients. In contrast, there is no theoretical convergence rate for AFCL. However, empirical findings in (Zhang et al., 2025) suggest a near-linear convergence rate at best. This suggests that our algorithm enjoys a lower total communication cost under the setting studied.

## C Proofs

### C.1 Proof of Proposition 1

*Proof.* Recall that Assumption 2 requires each term in the finite sample $\widehat{Q}_g(\boldsymbol{\theta}_g|\boldsymbol{\theta}_g')$ at client $g$ to be strongly concave. Now, let us define $\widehat{Q}_{k_g}(\boldsymbol{\theta}_{k_g}|\boldsymbol{\theta}_g')$ as follows:

$$\widehat{Q}_{k_g}(\boldsymbol{\theta}_{k_g}|\boldsymbol{\theta}_g^{(t-1)}) := \sum_{n_g=1}^{N_g} \gamma_{k_g}(\widehat{\boldsymbol{x}}_{n_g}, \boldsymbol{\theta}_g^{(t-1)}) \log(\pi_{k_g} p_{k_g}(\widehat{\boldsymbol{x}}_{n_g}|\boldsymbol{\theta}_{k_g}).$$

Since $\widehat{M}_{k_g}(\boldsymbol{\theta}_g^{(t-1)})$ is a maximizer of $\widehat{Q}_{k_g}(\boldsymbol{\theta}_{k_g}|\boldsymbol{\theta}_g^{(t-1)})$, then by strong concavity it must be unique. Therefore, we have that $\nabla_{\boldsymbol{\theta}_{k_g}} \widehat{Q}_{k_g}(\boldsymbol{\theta}_{k_g}|\boldsymbol{\theta}_g^{(t-1)}) = 0$ if and only if $\boldsymbol{\theta}_{k_g} = \widehat{M}_{k_g}(\boldsymbol{\theta}_g^{(t-1)})$. As a result,

we must always be able to obtain a unique $\varepsilon_{k_g}^{(t)} \geqslant 0$ such that

$$\widehat{Q}_{k_g}(\boldsymbol{\theta}_{k_g}^{(t-1)}|\boldsymbol{\theta}_g^{(t-1)}) \leqslant \widehat{Q}_{k_g}(\widehat{m}(\boldsymbol{\theta}_g^{(t-1)})|\boldsymbol{\theta}_g^{(t-1)}) \leqslant \widehat{Q}_{k_g}(\widehat{M}(\boldsymbol{\theta}_g^{(t-1)})|\boldsymbol{\theta}_g^{(t-1)}),$$

$$\forall \widehat{m}(\boldsymbol{\theta}_g^{(t-1)}) \in \mathbb{B}_2(\widehat{M}(\boldsymbol{\theta}_g^{(t-1)}), \sqrt{\varepsilon_{k_g}^{(t)}}).$$

$\square$

## C.2 Proof of Theorem 1

*Proof.* Note that Theorem 2 in (Balakrishnan et al., 2014) only guarantees convergence of the finite sample M-step $\widehat{M}_{k_g}(\boldsymbol{\theta}_g)$ to the neighborhood of the true cluster parameters $\boldsymbol{\theta}_{k_g}^*$, but does not examine the behavior within the neighborhood. However, we note that the finite sample EM is still a GEM algorithm, albeit characterized via the finite sample expected complete-data log-likelihood function $\widehat{Q}_g(\boldsymbol{\theta}_g|\boldsymbol{\theta}_g')$). Recall that we assume that the function $\widehat{Q}_g(\boldsymbol{\theta}_g|\boldsymbol{\theta}_g')$ is both strongly concave and continuous in both its conditioning and input arguments. Moreover, we assume that the finite sample true log-likelihood is bounded from above. Therefore, by Theorem 1 in (Wu, 1983), the finite sample EM iterates must converge to a stationary point of the finite sample true log-likelihood. This suggests that $\left[\widehat{Q}_g(\widehat{M}_g(\boldsymbol{\theta}_g^{(t-1)})|\boldsymbol{\theta}_g^{(t-1)}) - \widehat{Q}_g(\boldsymbol{\theta}_g^{(t-1)}|\boldsymbol{\theta}_g^{(t-1)}\right] \to 0$ as $t \to \infty$. Now, by the strong concavity of $\widehat{Q}_g(\boldsymbol{\theta}_g|\boldsymbol{\theta}_g')$) we have that

$$\widehat{Q}_g(\widehat{M}_g(\boldsymbol{\theta}_g^{(t-1)})|\boldsymbol{\theta}_g^{(t-1)}) - \widehat{Q}_g(\boldsymbol{\theta}_g^{(t-1)}|\boldsymbol{\theta}_g^{(t-1)} \geqslant \frac{\widehat{\tau}_g}{2}||\widehat{M}_g(\boldsymbol{\theta}_g^{(t-1)}) - \boldsymbol{\theta}_g^{(t-1)}||_2^2,$$

where $\widehat{\tau}_g$ is the strong concavity parameter. This implies that the algorithm must converge to a single point. $\square$

## C.3 Proof of Theorem 2

*Proof.* Observe that we can write the following with probability $(1 - \delta_g)$, where $0 \leqslant \delta_g \leqslant 1$.

$$||\widehat{m}_{k_g}(\boldsymbol{\theta}_g^{(t-1)}) - \boldsymbol{\theta}_{k_g}^*||_2 = \left\|\widehat{m}_{k_g}(\boldsymbol{\theta}_g^{(t-1)}) - \widehat{M}_{k_g}(\boldsymbol{\theta}_g^{(t-1)}) + \widehat{M}_{k_g}(\boldsymbol{\theta}_g^{(t-1)}) - \boldsymbol{\theta}_{k_g}^*\right\|_2 \tag{48a}$$

$$\leqslant \left\|\widehat{m}_{k_g}(\boldsymbol{\theta}_g^{(t-1)}) - \widehat{M}_{k_g}(\boldsymbol{\theta}_g^{(t-1)})\right\|_2 + \left\|\widehat{M}_{k_g}(\boldsymbol{\theta}_g^{(t-1)}) - \boldsymbol{\theta}_{k_g}^*\right\|_2 \tag{48b}$$

$$\leqslant \left\|\boldsymbol{\theta}_{k_g}^{(t-1)} - \widehat{M}_{k_g}(\boldsymbol{\theta}_g^{(t-1)})\right\|_2 + \left\|\widehat{M}_{k_g}(\boldsymbol{\theta}_g^{(t-1)}) - \boldsymbol{\theta}_{k_g}^*\right\|_2 \tag{48c}$$

$$\leqslant \frac{\beta_g}{\lambda_g}||\boldsymbol{\theta}_{k_g}^{(t-1)} - \boldsymbol{\theta}_{k_g}^*||_2 + \frac{1}{1 - \frac{\beta_g}{\lambda_g}}\epsilon_g^{\text{unif}}(N_g, \delta_g) + \left\|\boldsymbol{\theta}_{k_g}^{(t-1)} - \widehat{M}_{k_g}(\boldsymbol{\theta}_g^{(t-1)})\right\|_2, \tag{48d}$$

where (48d) follows from the convergence of the finite-sample EM algorithm. Now, note that by the same argument used in the proof of Theorem 5, we can argue that the term $\left\|\boldsymbol{\theta}_{k_g}^{(t-1)} - \widehat{M}_{k_g}(\boldsymbol{\theta}_g^{(t-1)})\right\|_2$ goes to 0 as $t \to \infty$. This concludes the proof. $\square$

## C.4 Proof of Theorem 3

*Proof.* Firstly, note that if $||\widehat{M}_{k_g}(\boldsymbol{\theta}_g^{(t-1)}) - \boldsymbol{\theta}_{k_g}^{(t-1)}||_2$ for component $k_g \in [K_g]$ at client $g \in [G]$ diminishes to 0 at a sufficiently fast rate (such as a geometric rate for example), then the local iterates $\widehat{m}_{k_g}(\boldsymbol{\theta}_g^{(t-1)})$ of our proposed algorithm for the component converge to a sphere of radius $\frac{1}{(1 - \frac{\beta_g}{\lambda_g})}\epsilon_g^{\text{unif}}(N_g, \delta_g)$ centered at the true centroid $\boldsymbol{\theta}_{k_g}^*$ with probability of at least $(1 - \delta_g)$. Now, in the worst case, the iterates for a specific component $k \in [K]$ from all the clients containing this component will converge to some point on the surface of the local sphere for each client $g$. Therefore, if the final aggregation radius $\varepsilon_{k_g}^{\text{final}}$ for all such clients is set according to match the radius of the local neighborhood of the true parameters, then all the aggregation uncertainty sets will also contain $\boldsymbol{\theta}_{k_g}^{ast}$. Therefore, our algorithm recognizes that all these estimates belong to one component

and aggregates them. Moreover, the assumption that $\varepsilon_{k_g}^{\text{final}} \leqslant \frac{R_{\min}}{2}$ at all components $k_g \in [K_g]$ at clients $g \in [G]$ guarantees that upon convergence, the parameter estimates for different global components $k, k' \in [K]$ from all clients remain distant enough such that they are not aggregated together. This, however, relies on the iterates for all components at all clients converging to the neighborhood of their true parameters. This is why our proposed algorithm infers the correct number of global clusters with the probability provided in the Theorem statement. $\square$

## C.5 PROOF OF THEOREM 4

*Proof.* We analyze the local client $g$ problem for each component $k_g$ as follows.

$$J_{k_g}(\boldsymbol{\theta}_g')$$

$$:= \begin{cases} \max_{\varepsilon_{k_g}} \quad \varepsilon_{k_g} \\[2mm] \text{s.t.} \quad \sum_{n_g=1}^{N_g} \gamma_{k_g}(\widehat{\boldsymbol{x}}_{n_g}, \boldsymbol{\theta}_g') \log(\pi_{k_g} p_{k_g}(\widehat{\boldsymbol{x}}_{n_g} | \widehat{m}_{k_g}(\boldsymbol{\theta}_g'))) \geqslant \\[3mm] \qquad \sum_{n_g=1}^{N_g} \gamma_{k_g}(\widehat{\boldsymbol{x}}_{n_g}, \boldsymbol{\theta}_g') \log(\pi_{k_g} p_{k_g}(\widehat{\boldsymbol{x}}_{n_g} | \boldsymbol{\theta}_{k_g}')) \quad \forall \widehat{m}_{k_g}(\boldsymbol{\theta}_g') \in \mathbb{B}_2(\widehat{M}_{k_g}(\boldsymbol{\theta}_g'); \sqrt{\varepsilon_{k_g}}) \end{cases} \tag{49}$$

$$= \begin{cases} \max_{\varepsilon_{k_g}} \quad \varepsilon_{k_g} \\[2mm] \text{s.t.} \quad \underbrace{\min_{\boldsymbol{\theta}_{k_g} \in \mathbb{B}_2(\widehat{M}_{k_g}(\boldsymbol{\theta}_g'); \sqrt{\varepsilon_{k_g}})} -\sum_{n_g}^{N_g} \gamma_{k_g}(\widehat{\boldsymbol{x}}_{n_g} - \boldsymbol{\theta}_{k_g})}_{M_{k_g}(\boldsymbol{\theta}_g', \varepsilon_{k_g})} \geqslant \underbrace{-\sum_{n_g=1}^{N_g} \gamma_{k_g}(\widehat{\boldsymbol{x}}_{n_g} - \boldsymbol{\theta}_{k_g}')}_{\text{constant}}, \end{cases} \tag{50}$$

where (50) follows by ignoring terms in $\widehat{Q}_g(\boldsymbol{\theta}_g | \boldsymbol{\theta}_g')$ that do not depend on $\boldsymbol{\theta}_{k_g}$, and from the fact that $\varepsilon_{k_g} \geqslant 0 \; \forall k_g \in [K_g]$. Now, let us consider the optimization problem $M_{k_g}(\boldsymbol{\theta}_g', \varepsilon_{k_g})$ in more detail as follows.

$$M_{k_g}(\boldsymbol{\theta}_g', \varepsilon_{k_g}) = \begin{cases} \min_{\boldsymbol{\theta}_{k_g}} \quad -\sum_{n_g=1}^{N_g} \gamma_{k_g}(\widehat{\boldsymbol{x}}_{n_g}, \boldsymbol{\theta}_g') \| \boldsymbol{x}_{n_g} - \boldsymbol{\theta}_{k_g} \|_2^2 \\[3mm] \text{s.t.} \quad \boldsymbol{\theta}_{k_g} \in \mathbb{B}_2(\widehat{M}_{k_g}(\boldsymbol{\theta}_g'), \sqrt{\varepsilon_{k_g}}) \end{cases} \tag{51}$$

$$= \begin{cases} \min_{\boldsymbol{\theta}_{k_g}} \quad -\sum_{n_g=1}^{N_g} \gamma_{k_g}(\widehat{\boldsymbol{x}}_{n_g}, \boldsymbol{\theta}_g')(\boldsymbol{x}_{n_g}^\top \boldsymbol{x}_{n_g} - 2\boldsymbol{x}_{n_g}^\top \boldsymbol{\theta}_{k_g} + \boldsymbol{\theta}_{k_g}^\top \boldsymbol{\theta}_{k_g}) \\[3mm] \text{s.t.} \quad \| \widehat{M}_{k_g}(\boldsymbol{\theta}_g') - \boldsymbol{\theta}_{k_g} \|_2^2 \leqslant \varepsilon_{k_g} \end{cases} \tag{52}$$

$$= \begin{cases} \min_{\boldsymbol{\theta}_{k_g}} \quad \boldsymbol{\theta}_{k_g}^\top \underbrace{\left( -\sum_{n_g=1}^{N_g} \gamma_{k_g}(\widehat{\boldsymbol{x}}_{n_g}, \boldsymbol{\theta}_g') I \right)}_{\boldsymbol{A}_0} \boldsymbol{\theta}_{k_g}^\top + 2 \underbrace{\left( \sum_{n_g=1}^{N_g} \gamma_{k_g}(\widehat{\boldsymbol{x}}_{n_g}, \boldsymbol{\theta}_g') \boldsymbol{x}_{n_g} \right)^\top}_{\boldsymbol{b}_0} \boldsymbol{\theta}_{k_g} + \\[5mm] \qquad\qquad\qquad\qquad \boldsymbol{x}_{n_g}^\top \underbrace{\left( -\sum_{n_g=1}^{N_g} \gamma_{k_g}(\widehat{\boldsymbol{x}}_{n_g}, \boldsymbol{\theta}_g') I \right)}_{c_0} \boldsymbol{x}_{n_g}^\top \\[5mm] \text{s.t.} \quad \boldsymbol{\theta}_{k_g}^\top \underbrace{I}_{\boldsymbol{A}_1} \boldsymbol{\theta}_{k_g} + 2 \underbrace{\left( -\widehat{M}_{k_g}(\boldsymbol{\theta}_g') \right)^\top}_{\boldsymbol{b}_1} \boldsymbol{\theta}_{k_g} + \underbrace{\widehat{M}_{k_g}(\boldsymbol{\theta}_g')^\top \widehat{M}_{k_g}(\boldsymbol{\theta}_g') - \varepsilon_{k_g}}_{c_1} \leqslant 0, \end{cases} \tag{53}$$

where $\boldsymbol{A}_0, \boldsymbol{A}_1 \in \mathbb{R}^{d \times d}$, $\boldsymbol{b}_0, \boldsymbol{b}_1 \in \mathbb{R}^d$, and $c_0, c_1 \in \mathbb{R}$. Now, note that the above problem is nonconvex, since $\boldsymbol{A}_0$ is not PSD. However, note that for all $\varepsilon_{k_g} > 0$, the above problem is strictly feasible.

Therefore, the problem obeys Slater's condition and admits a strong Lagrange dual. As shown by Boyd & Vandenberghe (2004), this Lagrange dual can be formulated as the SDP shown next.

$$
M'_{k_g}(\boldsymbol{\theta}'_g, \varepsilon_{k_g}) := \begin{cases} \max\limits_{\nu_{k_g}, \alpha_{k_g}} & \nu_{k_g} \\[1ex] \text{s. t.} & \alpha_{k_g} \geqslant 0 \\[1ex] & \begin{bmatrix} \boldsymbol{A}_0 + \alpha_{k_g}\boldsymbol{A}_1 & \boldsymbol{b}_0 + \alpha_{k_g}\boldsymbol{b}_1 \\ (\boldsymbol{b}_0 + \alpha_{k_g}\boldsymbol{b}_1)^\top & c_0 + \alpha_{k_g}c_1 - \nu_{k_g} \end{bmatrix} \succeq 0, \end{cases} \tag{54}
$$

where $\alpha_{k_g}, \nu_{k_g} \in \mathbb{R}$ are dual variables. We can reformulate this problem as follows.

$M'_{k_g}(\boldsymbol{\theta}'_g, \varepsilon_{k_g})$

$$
= \begin{cases} \max\limits_{\nu_{k_g}, \alpha_{k_g}} & \nu_{k_g} \\[1ex] \text{s. t.} & \alpha_{k_g} \geqslant 0 \\[1ex] & \begin{bmatrix} \boldsymbol{A}_0 + \alpha_{k_g}\boldsymbol{A}_1 & \boldsymbol{b}_0 + \alpha_{k_g}\boldsymbol{b}_1 \\ (\boldsymbol{b}_0 + \alpha_{k_g}\boldsymbol{b}_1)^\top & c_0 + \alpha_{k_g}c_1 - \nu_{k_g} \end{bmatrix} \succeq 0, \end{cases} \tag{55}
$$

$$
= \begin{cases} \max\limits_{\nu_{k_g}, \alpha_{k_g}} & \nu_{k_g} \\[1ex] \text{s. t.} & c_0 + \alpha_{k_g}c_1 - \nu_{k_g} - \\[1ex] & \qquad (\boldsymbol{b}_0 + \alpha_{k_g}\boldsymbol{b}_1)^\top \left( \left( \alpha_{k_g} - \sum_{n_g=1}^{N_g} \gamma_{k_g}(\widehat{\boldsymbol{x}}_{n_g}, \boldsymbol{\theta}'_g) \right) I \right)^{-1} (\boldsymbol{b}_0 + \alpha_{k_g}\boldsymbol{b}_1) \geqslant 0 \\[2ex] & \alpha_{k_g} - \sum_{n_g=1}^{N_g} \gamma_{k_g}(\widehat{\boldsymbol{x}}_{n_g}, \boldsymbol{\theta}'_g) \geqslant 0 \end{cases} \tag{56}
$$

$$
= \begin{cases} \max\limits_{\nu_{k_g}, \alpha_{k_g}} & \nu_{k_g} \\[1ex] \text{s. t.} & c_0 + \alpha_{k_g}c_1 - \nu_{k_g} - \dfrac{1}{\alpha_{k_g} - \gamma_{k_g}(\widehat{\boldsymbol{x}}_{n_g}, \boldsymbol{\theta}'_g)} ||\boldsymbol{b}_0 + \alpha_{k_g}\boldsymbol{b}_1||_2^2 \geqslant 0 \\[2ex] & \alpha_{k_g} - \sum_{n_g=1}^{N_g} \gamma_{k_g}(\widehat{\boldsymbol{x}}_{n_g}, \boldsymbol{\theta}'_g) \geqslant 0 \end{cases} \tag{57}
$$

$$
= \begin{cases} \max\limits_{\nu_{k_g}, \alpha_{k_g}} & \nu_{k_g} \\[1ex] \text{s. t.} & (c_0 + \alpha_{k_g}c_1 - \nu_{k_g})(\alpha_{k_g} - \sum_{n_g=1}^{N_g} \gamma_{k_g}(\widehat{\boldsymbol{x}}_{n_g}, \boldsymbol{\theta}'_g)) \geqslant ||\boldsymbol{b}_0 + \alpha_{k_g}\boldsymbol{b}_1||_2^2 \\[2ex] & \alpha_{k_g} \geqslant \sum_{n_g=1}^{N_g} \gamma_{k_g}(\widehat{\boldsymbol{x}}_{n_g}, \boldsymbol{\theta}'_g), \end{cases} \tag{58}
$$

where (56) is obtained via the Schur complement. Now, observe that the first constraint in (58) is monotonic in $\nu_{k_g}$. Moreover, note that plugging the problem in (58) into the constraint in problem (50) can be interpreted as requiring that the maximum value of $\nu_{k_g}$ satisfying the constraint must be greater than or equal to $-\sum_{n_g=1}^{N_g} \gamma_{k_g}(\widehat{\boldsymbol{x}}_{n_g}, \boldsymbol{\theta}'_g)||\boldsymbol{x}_{n_g} - \boldsymbol{\theta}'_{k_g}||_2^2$. Thus, it suffices to require that $\nu_{k_g} = -\sum_{n_g=1}^{N_g} \gamma_{k_g}(\widehat{\boldsymbol{x}}_{n_g}, \boldsymbol{\theta}'_g)||\boldsymbol{x}_{n_g} - \boldsymbol{\theta}'_{k_g}||_2^2$ satisfies the constraint in problem (58). Therefore, we

can use this result to rewrite the problem in (50) as follows.

$$J_{k_g}(\boldsymbol{\theta}'_g)$$

$$= \begin{cases} \displaystyle\max_{\varepsilon_{k_g},\alpha_{k_g}} & \varepsilon_{k_g} \\[2mm] \text{s.t.} & \left(c_0 + \alpha_{k_g} c_1 + \displaystyle\sum_{n_g=1}^{N_g} \gamma_{k_g}(\widehat{\boldsymbol{x}}_{n_g},\boldsymbol{\theta}'_g)||\boldsymbol{x}_{n_g} - \boldsymbol{\theta}'_{k_g}||_2^2\right)\left(\alpha_{k_g} - \displaystyle\sum_{n_g=1}^{N_g} \gamma_{k_g}(\widehat{\boldsymbol{x}}_{n_g},\boldsymbol{\theta}'_g)\right) \geqslant \\[4mm] & \hspace{6cm} ||\boldsymbol{b}_0 + \alpha_{k_g}\boldsymbol{b}_1||_2^2 \\[4mm] & \alpha_{k_g} \geqslant \displaystyle\sum_{n_g=1}^{N_g} \gamma_{k_g}(\widehat{\boldsymbol{x}}_{n_g},\boldsymbol{\theta}'_g) \end{cases}$$

(59)

$$= \begin{cases} \displaystyle\max_{\varepsilon_{k_g},\alpha_{k_g}} & \varepsilon_{k_g} \\[2mm] \text{s.t.} & \varepsilon_{k_g}\alpha_{k_g}^2 + \left[\displaystyle\sum_{n_g=1}^{N_g} \gamma_{k_g}(\widehat{\boldsymbol{x}}_{n_g},\boldsymbol{\theta}'_g)\right. \\[4mm] & \left(-\widehat{M}_{k_g}(\boldsymbol{\theta}'_g)^\top \boldsymbol{x}_{n_g} + \boldsymbol{x}_{n_g}^\top \boldsymbol{x}_{n_g} + \widehat{M}_{k_g}(\boldsymbol{\theta}'_g)^\top \widehat{M}_{k_g}(\boldsymbol{\theta}'_g) - \varepsilon_{k_g} - ||\boldsymbol{x}_{n_g} - \boldsymbol{\theta}'_{k_g}||_2^2\right)\Big]\alpha_{k_g} + \\[4mm] & \left(\displaystyle\sum_{n_g=1}^{N_g} \gamma_{k_g}(\widehat{\boldsymbol{x}}_{n_g},\boldsymbol{\theta}'_g)\right)\displaystyle\sum_{n_g=1}^{N_g} \gamma_{k_g}(\widehat{\boldsymbol{x}}_{n_g},\boldsymbol{\theta}'_g)||\boldsymbol{x}_{n_g} - \boldsymbol{\theta}'_{k_g}||_2^2 \leqslant 0 \\[4mm] & \alpha_{k_g} \geqslant \displaystyle\sum_{n_g=1}^{N_g} \gamma_{k_g}(\widehat{\boldsymbol{x}}_{n_g},\boldsymbol{\theta}'_g) \end{cases}$$

(60)

$$= \begin{cases} \displaystyle\max_{\varepsilon_{k_g},\alpha_{k_g}} & \varepsilon_{k_g} \\[2mm] \text{s.t.} & \varepsilon_{k_g}\alpha_{k_g}^2 + \left[\displaystyle\sum_{n_g=1}^{N_g} \gamma_{k_g}(\widehat{\boldsymbol{x}}_{n_g},\boldsymbol{\theta}'_g)\left(||\boldsymbol{x}_{n_g} - \widehat{M}_{k_g}(\boldsymbol{\theta}'_g)||_2^2 - ||\boldsymbol{x}_{n_g} - \boldsymbol{\theta}'_{k_g}||_2^2 - \varepsilon_{k_g}\right)\right]\alpha_{k_g} + \\[4mm] & \left(\displaystyle\sum_{n_g=1}^{N_g} \gamma_{k_g}(\widehat{\boldsymbol{x}}_{n_g},\boldsymbol{\theta}'_g)\right)\displaystyle\sum_{n_g=1}^{N_g} \gamma_{k_g}(\widehat{\boldsymbol{x}}_{n_g},\boldsymbol{\theta}'_g)||\boldsymbol{x}_{n_g} - \boldsymbol{\theta}'_{k_g}||_2^2 \leqslant 0 \\[4mm] & \alpha_{k_g} \geqslant \displaystyle\sum_{n_g=1}^{N_g} \gamma_{k_g}(\widehat{\boldsymbol{x}}_{n_g},\boldsymbol{\theta}'_g) \end{cases}$$

(61)

$\square$

# D  SUPPLEMENTARY EXPERIMENTAL DETAILS AND RESULTS

In this section we provide all the details of all the experiments presented in this paper, as well as supplementary results for both the Benchmarking and Sensitivity Studies. Please note that the all the code and instructions associated with all the experiments is provided separately in the supplementary materials.

## D.1  SOFTWARE AND HARDWARE DETAILS

All the experiments presented in this work were executed on Intel Xeon Gold 6226 CPUs @ 2.7 GHz (using 10 cores) with 120 Gb of DDR4-2993 MHz DRAM. Table 3 provides more detail on all the software used in the paper.

Table 3: Details on All the Software Used in the Numerical Experiments.

| Software | Version | License |
|---|---|---|
| Gurobi | 10.0.1 | Academic license |
| MATLAB | R2021B | Academic license |
| Python | 3.10.9 | Open source license |
| Scikit-Learn | 1.2.1 | Open source license |
| Numpy | 1.23.5 | Open source license |
| Scipy | 1.10.0 | Open source license |
| UCIMLRepo | 0.0.3 | Open source license |
| TensorFlow | 2.12.0 | Open source license |

## D.2 DATASETS UTILIZED

### D.2.1 BENCHMARKING STUDY

In the benchmarking study we utilize various popular real-world datasets for evaluation. We provide more detail on the datasets in Table 4.

Table 4: Details on Datasets Utilized for UCI Experiments.

| Dataset | Abbreviation | License | $N$ | $K$ | $d$ |
|---|---|---|---|---|---|
| MNIST (LeCun et al., 2010) (embeddings: (Bickford Smith et al., 2024b)) | MNIST | CC BY 4.0 (embeddings: MIT License) | 70,000 | 10 | 10 |
| Fashion MNIST (Xiao et al., 2017) | FMNIST | CC BY 4.0 | 70,000 | 10 | 64 |
| Extended MNIST (Balanced) (Cohen et al., 2017) | EMNIST | CC BY 4.0 | 131,600 | 47 | 16 |
| CIFAR-10 (Krizhevsky et al., 2009) (embeddings: (Bickford Smith et al., 2024a) | CIFAR-10 | CC BY 4.0 (embeddings: MIT License) | 60,000 | 10 | 64 |
| Abalone (Nash et al., 1994) | Abalone | CC BY 4.0 | 4177 | 7 | 8 |
| Anuran Calls (MFCCs) (Colonna et al., 2015) | FrogA | CC BY 4.0 | 7195 | 10 | 21 |
| Anuran Calls (MFCCs) (Colonna et al., 2015) | FrogB | CC BY 4.0 | 7195 | 8 | 21 |
| Waveform Database Generator (Version 1) (Breiman & Stone, 1984) | Waveform | CC BY 4.0 | 5000 | 3 | 21 |

**Preprocessing - Image Datasets.** Rather than directly clustering the images in the MNIST, FMNIST, EMNIST, and CIFAR-10 datasets, we utilize embeddings extracted from them to reduce the computational expense of the experiments. These embeddings are extracted via variational autoencoders (VAEs). More specifically, for the MNIST dataset we utilize the vanilla VAE embeddings available at (Bickford Smith et al., 2024b), which have dimension 10. For the FMNIST and EMNIST datasets, we implement VAEs with latent dimensions 64 and 16, respectively. Subsequently, we utilize the encoded mean vectors of the samples as the data utilized for clustering. Finally, for the CIFAR-10 dataset we utilize the "Barlow" embeddings available at (Bickford Smith et al., 2024a). However, we further encode these embeddings via a VAE with latent dimension 64 as we do for FMNIST and EMNIST. All code utilized for feature extraction is provided in the supplementary materials available with the submission.

**Preprocessing - Abalone.** Since the Abalone dataset has very small clusters (some of which contain only 1 sample), we combined various clusters together. This makes sense physically, as the target label in the dataset is an integer the age of the abalone. Therefore, combining various labels into bins enforces a more categorical structure on the age. More specifically, we combined labels 0 through 5 into one cluster, kept labels 6 through 10 as separate clusters, combined labels 11 and 12 into one cluster, and combined labels 13 through 28 into one cluster.

### D.2.2 SENSITIVITY STUDY

The data utilized in this experiment is generated using the `make_blobs` module of the `scikit-learn` Python package. This module generates isotropic Gaussian clusters, making it ideal for our problem setting. The data is generated so that the centroids of the clusters have a preset minimum distance of $R_{\min}$ between them. Moreover, data generation is designed so that at least two of the generated clusters have centroids that are exactly $R_{\min}$ apart. It is worth noting that for the sensitivity study, the dataset generated during each repetition is tested 3 times for each model.

During each of those times, the model starts with a random initialization via `k-means++`. The results we report are the maximum performance obtained over the 3 initializations.

## D.3 HYPERPARAMETER DETAILS AND PERFORMANCE EVALUATION

In this section, we provide a detailed practical discussion on the tuning our algorithm's final aggregation radii. As mentioned in Section 6, we use $\varepsilon_{k_g}^{\text{final}} = \frac{v_g \widehat{R}_{\min_g}}{\pi_{k_g} \sqrt{N_g}}$ as a practical heuristic, where $v_g$ is the hyperparameter we directly tune. For simplicity, we set $v_g$ equivalently for all clients $g \in [G]$. This heuristic allows the aggregation radii to scale appropriately with the scale of the feature space and the number of samples available at each client. Additionally, it allows the final aggregation radii to adapt to each cluster at each client while requiring the tuning of only one hyperparameter.

We utilize cross-validation to tune $v_g$, using SS Rousseeuw (1987) as a performance metric. We also provide a practical guide to evaluate the estimated $\widehat{K}$ without knowledge of the true $K$. To that end, we ensure that $\widehat{K}$ does not significantly exceed $\sqrt{\sum_{g=1}^{G} K_g}$. This guide is inspired by a rough estimate that for any client $g \in [G]$, $K_g \sim \mathcal{O}(K)$, which suggests that $\sum_{g=1}^{G} K_g \sim \mathcal{O}(KG)$. Since in most practical cases we have $G > K$, then we can see that $\sqrt{\sum_{g=1}^{G} K_g} \sim \mathcal{O}(\sqrt{KG}) > \mathcal{O}(K)$.

All hyperparameters for all benchmark models are set as prescribed in their respective works. Additionally, note that the estimated number of clusters provided to the DP-GMM and AFCL models is $\sum_{g=1}^{G} K_g$, as this constitutes an upper bound for $K$. Note that this initial estimate is significantly closer to the true number of clusters for all datasets than the initial value of $\frac{\sum_{g=1}^{G} N_g}{32}$ suggested for AFCL in (Zhang et al., 2025). The reported estimated number of clusters for both algorithm is the total number of clusters to which test samples were assigned. Moreover, we run all iterative algorithms for $T = 20$ iterations, and we run our algorithm for $T = 10$ iterations. Furthermore, we utilize $S = 1$ local steps for our model in all setting, as well as $I = 10$ iterations for Algorithm 4.

It should be noted that since our model is personalized, the reported performance for our model is a weighted average of the clients' individual performance metrics.

Table 5: Hyperparameter $v_g$ tuning values for all datasets used in the Benchmarking Study.

| Dataset | Hyperparameter Value(s) |
|---------|-------------------------|
| MNIST | $2e0$ |
| FMNIST | $3e2$ |
| EMNIST | $2e0$ |
| CIFAR-10 | $2e1$ |
| Abalone | $\{5e3, 7e3, 9e3\}$ |
| Frog A | $\{1e4, 1e5, 1e6\}$ |
| Frog B | $\{1e4, 1e5, 1e6\}$ |
| Waveform | $\{1e0, 5e0, 1e1\}$ |
| Synthetic | $\{5e-1, 1e0, 5e0, 1e1, 5e1\}$ |

## D.4 SUPPLEMENTARY BENCHMARKING STUDY RESULTS

### D.4.1 SILHOUETTE SCORE

We provide additional results for our Benchmarking Study using the SS evaluation metric in Table 6. We immediately observe that our proposed method continues to attain the highest performance out of the federated methods with unknown $K$ for most datasets. Furthermore, we observe that on datasets such as Abalone and Waveform, our method outperforms even the top performing method with known $K$. Since the results using these evaluation metrics are similar to those using the ARI, we reach a strong conclusion that our proposed model has a significant practical impact. Namely, it can achieve similar performance to, or even outperform some clustering methods that assume prior

knowledge of $K$, and it often outperforms method without prior knowledge of $K$. It achieves this while being federated (i.e. not requiring any data movement), and without prior knowledge of $K$.

Table 6: SS attained by all methods on tested datasets.

| Model | Known $K$? | MNIST | FMNIST | EMNIST | CIFAR-10 | Abalone | Frog A | Frog B | Waveform |
|---|---|---|---|---|---|---|---|---|---|
| GMM (central) | Yes | .029 ±.020 | .093 ±.008 | .046 ±.006 | .191 ±.013 | .397 ±.014 | .232 ±.042 | .198 ±.089 | **.255** ±.026 |
| k-FED | Yes | .076 ±.010 | .106 ±.022 | .075 ±.010 | .177 ±.012 | **.406** ±.029 | .285 ±.066 | .278 ±.076 | .245 ±.028 |
| FFCM-avg1 | Yes | −.045 ±.030 | .009 ±.0023 | −.113 ±.026 | −.014 ±.045 | .400 ±.011 | .229 ±.051 | .259 ±.061 | .247 ±.008 |
| FFCM-avg2 | Yes | .036 ±.011 | .052 ±.018 | −.077 ±.023 | .087 ±.021 | .404 ±.016 | .272 ±.071 | **.324** ±.056 | .231 ±.042 |
| FedKmeans | Yes | **.105** ±.005 | **.127** ±.006 | **.091** ±.003 | **.203** ±.006 | .404 ±.010 | **.302** ±.054 | .289 ±.074 | .252 ±.003 |
| DP-GMM (central) | No | −.082 ±.014 | −.058 ±.014 | −.063 ±.007 | **.117** ±.005 | **.324** ±.023 | .171 ±.040 | .144 ±.025 | .115 ±.013 |
| AFCL | No | .015 ±.002 | .017 ±.003 | .028 ±.002 | .106 ±.005 | .192 ±.028 | .144 ±.048 | .156 ±.045 | .018 ±.010 |
| FedGEM (**ours**) | No | **.095** ±.012 | **.069** ±.018 | **.063** ±.009 | .094 ±.015 | .307 ±.086 | **.324** ±.082 | **.284** ±.092 | **.271** ±.011 |

### D.4.2 BENCHMARKING AGAINST MODELS WITH KNOWN $K$ VIA ESTIMATION HEURISTIC

We perform an additional study where we compare the performance of our `FedGEM` algorithm to that of one with known $K$, where $K$ is treated as a hyperparameter. We note that all results presented in this study are averages over 20 repetitions.

**Our Method.** As with all the previous experiments, our method is the isotropic GMM from Section 5 trained via our `FedGEM` algorithm.

**Evaluation Metrics.** In this experiment, we evaluate model performance based on ARI and runtime. This allows us to examine potential improvements offered by our approach in terms of clustering performance as well as the potential sacrifice in computation needed to achieve this improvement.

**Baselines.** We compare our algorithm to `FedKmeans` Garst & Reinders (2024) due to its high performance in the Benchmarking Study in Section 6, which makes it a strong baseline.

**Dataset.** We leverage dataset generated via the `make_blobs` module in Python with $K = 20$ and $R_{min} = 2$ across 4 distinct settings. Namely, we vary the maximum amount of clusters $K_{g_{max}}$ available at any one client such that $K_{g_{max}} \in \{5, 10, 15, 20\}$. Note that during each repetition, there need not be a client $g$ whose $K_g = K_{g_{max}}$. Rather, $K_{g_{max}}$ is an upper bound on $K_g$ for all $g \in [G]$, which may or may not be attained based on the randomly assigned clusters. In all settings, we use $G = 15$ clients, $d = 4$ features, $N_g = 400$ training samples per client, and $N_{test} = 2000$ test samples.

**Hyperparameters.** In this experiment, we tune the hyperparameter $v_g$ associated with our model within the values $v_g \in \{1e0, 5e0, 9e0, 1e1, 5e1\}$. Similarly, we tune $K$ for `FedKmeans`. To that end, we search over 5 equally spaced values of $K$, the smallest of which is the maximum amount of clusters $K_{g_{max}}$ available at any one client, and the largest of which is the sum $\sum_{g=1}^{G} K_g$. These values are chosen as they form lower and upper bounds on $K$ respectively. All tuning is performed through 3-fold cross-validation using the SS as a metric due to lack of labels during training time. Additionally, we note that we ran `FedKmeans` for 10 iterations, while we used $S = 10$ local iterations and $T = 1$ global round for our algorithm.

**Results.** We present the results of this study in Figure 2. Firstly, we observe that our approach offers much stronger and more stable clustering performance across most settings, with a much smaller standard deviation. This is because `FedKmeans`'s performance is very sensitive to the tuning values of $K$, whereas our algorithm's methodical inference of the cluster number allows it to enjoy improved stability across hyperparameter values and problem settings. Secondly, we observe that the performance of our approach slightly diminishes as $K_{g_{max}}$ increases. This could be

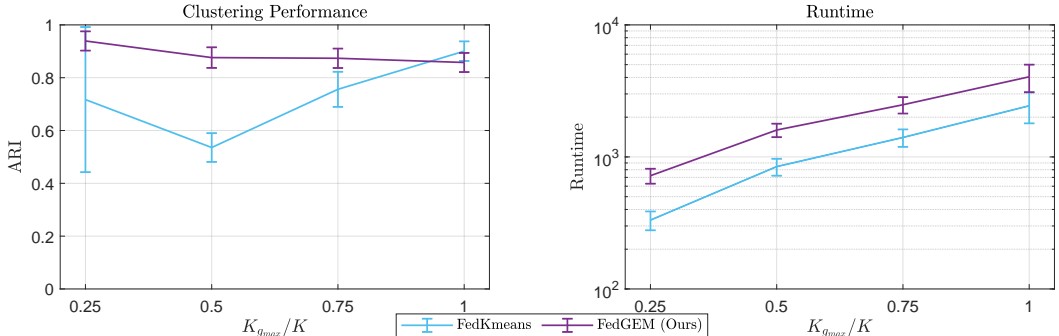

Figure 2: Results of the study comparing our proposed algorithm to one with prior knowledge of $K$ where $K$ is treated as a model hyperparameter.

attributed to two key reasons. First, we use fixed hyperparameter tuning values across all problem settings, whereas different values may lead to improved performance for higher $K_{g_{\max}}$. Secondly, the personalized nature of performance evaluation in our algorithm may result in slight performance reductions as the number of locally available clusters at each client increases. On the contrary, we observe a general increasing trend in FedKmeans's performance as $K_{g_{\max}}$ increases. This could be attributed to the smallest tuning value of $K$ approaching the true value of $K = 20$ as $K_{g_{\max}}$ increases. Indeed, we observe very strong performance at $K_{g_{\max}} = K = 20$, as the true number of clusters is included in the tuning values. However, FedKmeans's performance with $K_{g_{\max}} = 10$ highlights the potential problem with the use of such tuning heuristic for methods that assume prior knowledge of $K$. Namely, the selected tuning values may be highly incorrect, which can result in very poor performance even with the use of cross-validation. Similarly, we observe that the standard deviation of FedKmeans's performance at $K_{g_{\max}} = 5$ is very large. This is likely due to very different values of the hyperparameter $K$ being selected by the cross-validation heuristic across different repetitions, resulting in very large performance variations and instability. Finally, we observe that the improved performance and stability offered by our approach comes at a very minor additional computational expense while maintaining the same scalability as FedKmeans. We note that while these results are promising, they may differ with different datasets. This motivates future work to carefully study the conditions under which our proposed algorithm is advantageous over algorithms with known $K$, where $K$ is tuned as a hyperparameter.

### D.5 SUPPLEMENTARY SENSITIVITY STUDY RESULTS

We present the results of the Sensitivity Study utilizing the SS to compare model performance in Figure 3. Firstly, we again observe that performance of both models improves as $R_{\min}$ increases. Surprisingly, however, we see that our proposed model outperforms GMM in all setting, which does not match the ARI result. This could be explained by SS's sensitivity to the number of clusters. Indeed, a problem with a smaller number of clusters is likely to exhibit higher SS than an identical one with a larger number of clusters. Since each client only has a subset of the clusters locally, this can cause the local SS to be over-inflated. However, as seen in the ARI result, we can conclude that our proposed model offers very close performance to that of a centralized one with known $K$, which is a powerful result.

### D.6 SENSITIVITY TO HYPERPARAMETER

This study evaluates the sensitivity of our proposed algorithm to its final aggregation radius hyperparameter.

**Our Method.** As with the other numerical experiments, our method is the isotropic GMM model trained via our proposed FedGEM algorithm.

**Evaluation Metric.** We examine the sensitivity of both the ARI and the estimated number of clusters to the hyperparameter.

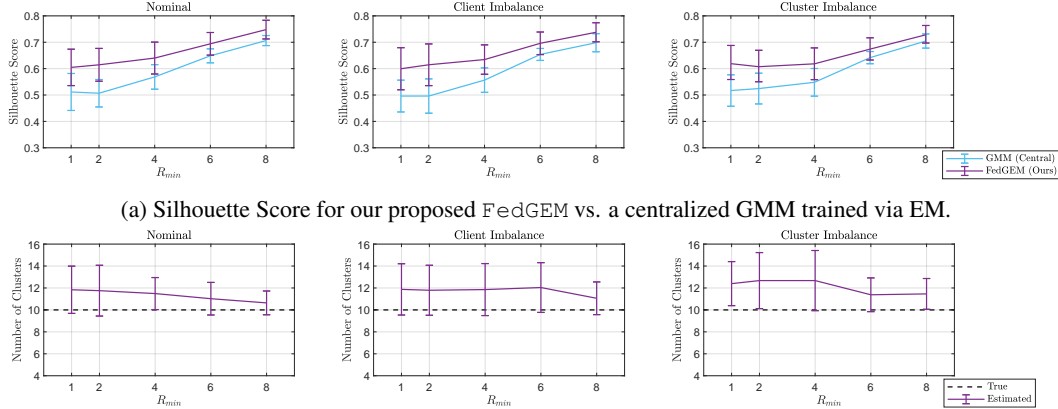

(a) Silhouette Score for our proposed `FedGEM` vs. a centralized GMM trained via EM.

(b) Number of clusters estimated by our proposed `FedGEM` vs. the true number of clusters.

Figure 3: Supplementary results of the sensitivity study.

**Hyperparameters.** Recall our final aggregation radius heuristic $\varepsilon_{k_g}^{\text{final}} = \frac{v_g \widehat{R}_{\min_g}}{\pi_{k_g} \sqrt{N_g}}$. We evaluate our model's performance for $v_g \in \{1e-1, 1e0, 1e1, 1e2\}$.

**Dataset.** The data used for this experiment is isotropic Gaussian clusters generated via the `make_blobs` module in Python. We set $R_{\min} = 4$, and we study three key settings: i) nominal: data is balanced across clients and clusters, ii) client imbalance: the data is imbalanced across clients, and iii) cluster imbalance: the portion of each cluster in the local data at each client is randomly samples followed by normalization. For all settings we use $G = 15$, $K = 10$, $N_{\text{train}} = 7500$, and $N_{\text{test}} = 2000$.

**Results.** The results of this study are displayed in Figure 4. We observe that the estimated number of clusters can be more sensitive to the choice of $v_g$ than ARI. This is intuitive, as a value of $v_g$ that is too small will result in insufficient cluster aggregation, which causes the estimated number of clusters to be overinflated. However, since clustering performance evaluation is performed locally at each client, ARI can still be somewhat stable in this setting. On the other hand, if $v_g$ is too large, this will cause estimates associated with different clusters to be aggregated together. This leads to an underestimation of the number of clusters and also significantly affects clustering performance. A key observation we make is that for an appropriately adjusted $v_g$, ARI seems to reach a peak value in the nominal case while the estimated number of clusters almost coincides with the true value. This highlights the importance of hyperparameter tuning via the protocol we present in Appendix D.3. Finally, we note that the cluster and client imbalance settings do not significantly affect our model's performance, suggesting robustness to such issues.

# E  SCALABILITY STUDY

In this section, we present two distinct studies we performed to evaluate the scalability of our proposed algorithm.

**Our Method.** As stated in the full paper, our method is the isotropic GMM model trained via our proposed `FedGEM` algorithm.

**Evaluation Metric.** We utilize runtime in seconds to evaluate the scalability of all methods.

**Baselines.** We compare our algorithm to AFCL, which is the only other federated clustering method that does not require knowledge of $K$. Additionally, we also use FFCM-avg2 and `FedKmeans` as benchmarks as they achieved strong performance in the Benchmarking Study.

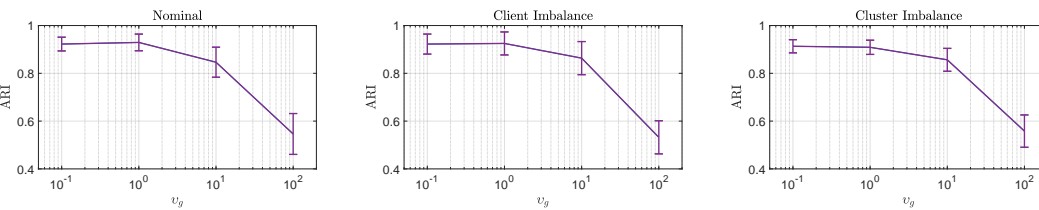

(a) Sensitivity of our algorithm's clustering performance measured via ARI to the hyperparameter value.

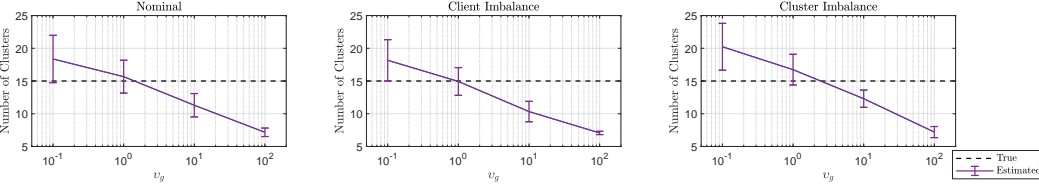

(b) Sensitivity of our algorithm's number of clusters estimation to the hyperparameter value.

Figure 4: Results on the sensitivity of our algorithm to its hyperparameter.

### E.1 SCALABILITY ON IMAGE DATASETS

We examine the runtime of some of our algorithm as well as some of the federated benchmarks on the larger-scale image datasets. This allows us to evaluate the scalability of our proposed algorithm in realistic settings.

**Hyperparameters.** All hyperparameters and experimental settings (e.g. number of clients $G$, hyperparameter settings, etc...) are exactly the same as described in detail in Appendix D.3. However, in the interest of fairness, we run all federated algorithms for $T = 10$ iterations, and we confirm that their performance after training is on par with the values reported previously.

**Datasets.** In this experiment we focus solely on the MNIST, FMNIST, EMNIST, and CIFAR-10 datasets. This is because they are on a much larger scale than the other datasets tested, therefore they provide meaningful insights into algorithm scalability.

**Results.** The results of this experiment are reported in Table 7. We observe that our algorithm achieves a much shorter runtime than AFCL (the only other federated clustering approach without prior knowledge of $K$). This emphasizes the significant practical impact of our algorithm, as it also achieved superior clustering performance and total number of cluster estimation as discussed in Section 6 and Appendix D. As we discuss in the Scalability Study on Synthetic Data in Appendix E.2, this advantage over AFCL is most likely due to improved scalability with respect to the number of clients. This suggests that our algorithm is better suited for distributed clustering problems over large networks involving large volumes of data.

Table 7: Runtime in seconds of selected federated algorithms on the image datasets evaluated.

| Model | Known $K$? | MNIST | FMNIST | EMNIST | CIFAR-10 |
|---|---|---|---|---|---|
| FFCM-avg2 | Yes | $220 \pm 18$ | $440 \pm 25$ | $6075 \pm 842$ | $188 \pm 8$ |
| FedKmeans | Yes | $30 \pm 2$ | $52 \pm 3$ | $314 \pm 42$ | $26 \pm 1$ |
| AFCL | No | $2047 \pm 246$ | $2013 \pm 204$ | $3176 \pm 722$ | $1798 \pm 165$ |
| FedGEM (**ours**) | No | $552 \pm 52$ | $645 \pm 63$ | $1628 \pm 335$ | $345 \pm 35$ |

### E.2 SCALABILITY ON SYNTHETIC DATA

This study aims to evaluate the scalability of our proposed algorithm as the size of the training dataset and the federated network grow. It also compares the scalability of our algorithm to that of multiple federated benchmarks. We note that the implementation of our algorithm used in this study relies on pairwise server computations. Therefore, scalability can likely be further improved by leveraged a KD tree as explained in Appendix B.5.

**Hyperparameters.** Since the focus of this study is more so on execution time than model performance, we did not perform hyperparameter tuning for this experiment. We fix our final aggregation radius hyperparameter $v_g = 1e0$ for all $g \in G$. We set the hyperparameters of benchmark models as prescribed in their corresponding papers. Additionally, we run all algorithms with $T = 10$ iterations. Finally, this experiment was repeated for 10 repetitions.

**Dataset.** In this experiment we utilize data generated via the `make_blobs` module in Python, which generates isotropic Gaussian clusters. We utilize $R_{\min} = 2$ across all experiments. Additionally, we study 4 distinct experimental settings, listed next.

1. **Increasing Features:** $G = 5$, $N_g = 500$, $K = 10$, $d \in \{5, 25, 45, 65\}$.
2. **Increasing Training Samples per Client:** $G = 5$, $K = 10$, $d = 15$, $N_g \in \{500, 2500, 4500, 6500\}$.
3. **Increasing Clusters:** $G = 5$, $N_g = 500$, $d = 15$, $K \in \{5, 25, 45, 65\}$.
4. **Increasing Clients:** $N = 1000$, $K = 10$, $d = 15$, $G \in \{5, 25, 45, 65\}$.

Across all experiments, we uniformly sample $K_g$ for all $g \in [G]$ such that $2 \leqslant K_g < K$.

**Results.** The results of this experiment are shown in Figure 5. Firstly, we observe that the runtime of all algorithms remains constant as the number of features increases. This suggests that all compared algorithms, including ours, scale well with the number of features. Secondly, we observe that while our algorithm exhibits a greater runtime than benchmark methods in all experimental settings, it scales at a similar rate to them as the number of samples, clusters, and clients increase. This suggests strong scalability across all settings. Moreover, we observe in the increasing number of clients setting that AFCL's runtime increases at a faster rate than our proposed algorithm. This suggests that our algorithm scales better in this setting, and can therefore be more suitable for settings with a large number of clients. This observation aligns with our results presented in the runtime analysis in Appendix E.1, where we observe that our algorithm achieves a shorter runtime than AFCL in experiments involving a high $G$ and large datasets. Combined with the fact that our algorithm exhibited better clustering performance and true number of clusters estimation across all our experiments, this highlights the significant practical impact of our proposed `FedGEM` algorithm.

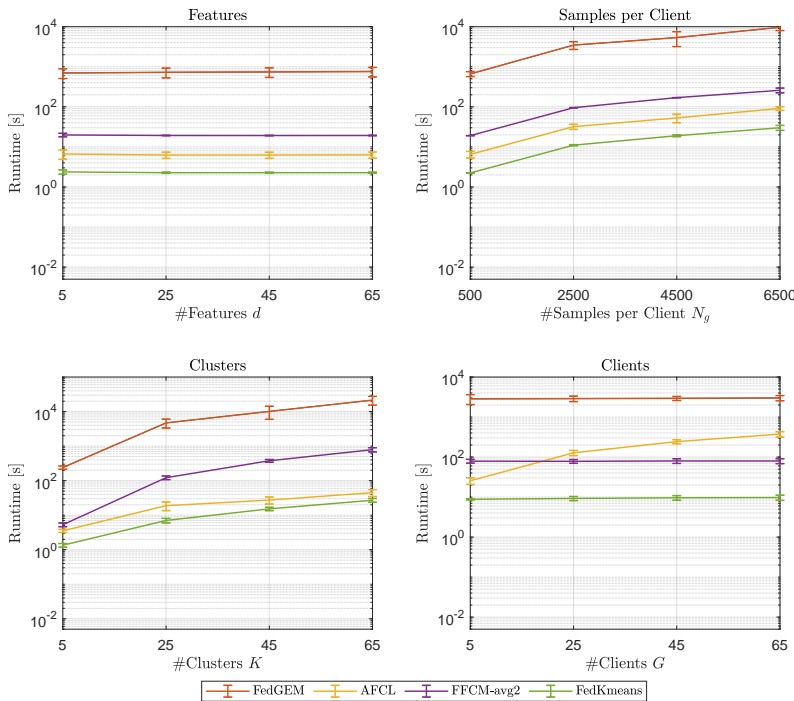

Figure 5: Results of the scalability experiment for all experimental settings and benchmark models.

## F FURTHER DISCUSSION

### F.1 JUSTIFICATION AND INTERPRETATION OF MODELING ASSUMPTIONS

- **Assumption 1: Ground Truth Parameters.** In this assumption, we enforce a modeling structure that is necessary for the convergence analysis of our algorithm. Namely, that any clusters that are shared by multiple clients, have the exact same ground truth parameters at all clients. Note that this assumption does not violate the non-IID nature of the data in FL problems. This is because cluster weights can be different across clients, and clients may have different clusters. Therefore, the data across clients is still non-IID. This assumption is common in works studying federated EM algorithms, such as (Marfoq et al., 2021).

- **Assumption 2: Strong Concavity.** This assumption requires each of the terms in the expected complete-data log-likelihood functions to be strongly concave, thereby allowing for the function to have a unique maximizer. Such assumption is very common (at least locally near the optimum) in works examining the convergence of EM algorithms such as (Balakrishnan et al., 2014). This assumption is also readily verifiable for models such as GMMs.

- **Assumption 3: First-Order Stability.** This assumption requires the expected complete-data log-likelihood to obey a Lipschitz-like smoothness constraint, introduced by Balakrishnan et al. (2014) and defined in 1. Such a technical assumption is vital for the theoretical analysis, and the derivation of convergence guarantees, but is not required for the algorithm to be used in practice.

- **Assumption 4: Continuity.** This is another technical assumption, which requires the complete-data log-likelihood function of the model used to be smooth in both its input and conditioning arguments. This is standard in many EM-related efforts, and is only required for the theoretical convergence analysis but not for algorithm use in practice.

- **Assumption 5: Likelihood Boundedness.** This assumption requires the log-likelihood of the model used to be bounded from above, although that bound need not be known. Similar to previous assumptions, this one is also purely required for the theoretical convergence analysis, but not for the use of the algorithm in practice. Note that this assumption is common in works investigating Federated EM algorithms such as (Marfoq et al., 2021), and is easily verifiable for models such as GMM under mild conditions on the covariance matrix.

- **Assumption 6: Finite-Sample and Population M-Step Proximity.** This assumption requires there to be an upper bound on the maximum difference between the population M-step and the finite-sample M-step for each cluster with a certain probability. Whereas all the previous assumptions allow us to theoretically study the convergence of our algorithm on the population level (i.e. with infinite data), this one is necessary for the finite-sample convergence analysis. Specifically, it allows us to prove that the algorithm updates made via a finite data sample indeed converge to a neighborhood of the converged population-based iterates. This assumption was utilized in works exploring the convergence of EM algorithms such as (Balakrishnan et al., 2014; Yan et al., 2017), and is also purely technical and does not impact algorithm usability in practice.

- **Assumption 7: Bounded Support.** This assumption requires the support of the feature vector to be bounded, and is needed **only** in the setting where DP is used to privatize the cluster maximizers shared by the clients. Such an assumption is not restrictive. This is because data is often collected via acquisition devices with known ranges. Therefore, feature support is either already bounded, or can be via normalization.

### F.2 INTERPRETATION OF THEORETICAL RESULTS

- **Proposition 1: Local Uncertainty Set Radius Problem.** This proposition asserts that the optimization problem solved by each client to obtain the radius of the uncertainty set centered at the maximizer of each local cluster must have a unique solution. The unique solution would be 0 at convergence. This holds under the modeling assumption thanks to the strong concavity of the complete-data log-likelihood function.

- **Theorem 1: Single-Point EM Convergence.** This theorem asserts that the finite sample EM iterates computed by each client for each local cluster must converge to a single point withing a certain proximity of the ground truth parameters. This is a subtle, but key result, as it ensures stability and lack of oscillations upon convergence.

- **Theorem 5: Local Convergence of Population GEM.** This theorem asserts that, in the population setting (i.e. infinite training samples), iterates that are computed via our proposed `FedGEM` algorithm converge exactly to the ground truth parameters. This is a very strong convergence result, which is used to establish the finite-sample convergence of the algorithm.

- **Theorem 2: Local Convergence of Finite-Sample GEM.** This theorem asserts that, with a certain probability, iterates that are computed via our proposed `FedGEM` algorithm converge within a certain radius around the ground truth parameters at any client. This is achieved with only a finite number of training samples. This result forms the basis for our convergence argument. This is because iterates of a shared cluster across multiple clients converge to a close proximity of each other. Therefore, given a final aggregation radius that meets certain conditions, they can be successfully aggregated into a single cluster.

- **Theorem 3: Number of Clusters Inference:** This theorem asserts that with a certain probability, our algorithm correctly estimates the total unique number of clusters across clients. This is reliant on the finite-sample convergence established in Theorem 2.

- **Theorem 4: Radius Problem Reformulation.** This theorem provides a tractable, bi-convex, 2-dimensional reformulation for the semi-infinite uncertainty set radius problem in the case of isotropic GMMs. This renders our algorithm tractable for this specific model, and allows us to implement it in our numerical experiments.

- **Proposition 2: Local Radius Algorithm Convergence.** This proposition shows that Algorithm 4 proposed to solve the uncertainty set radius problem reformulation from Theorem 4 enjoys a very low time complexity. This allows our `FedGEM` algorithm to scale well with problem size.

- **Theorem 6: GMM First-Order Stability.** This theorem proves that the multi-component isotropic GMM explored in this work indeed satisfies the FOS condition defined in 1. This is a very impactful result, as, to the best of our knowledge, this is the first time such result is formally proven for a GMM with more than two components. This condition is necessary for the convergence of our algorithm. Therefore, formally proving it allows us to argue that our `FedGEM` algorithm is guaranteed to converge for multi-component, isotropic GMMs.

- **Theorem 7: GMM M-Step Contraction Region.** This theorem examines the necessary problem conditions, including an upper bound on the radius of the contraction region at each client, for the multi-component isotropic GMM to converge. This discussion allows us to argue that our proposed algorithm converges for the isotropic GMM under consideration if the problem conditions are satisfied. However, we note that this is a purely technical result needed only for the theoretical convergence analysis, but not for practical implementation.

- **Theorem 8: GMM Finite-Sample and Population M-Step Distance.** This theorem derives the upper bound on the distance between the population and final sample M-steps that is required by Assumption 6. The existence of this bound guarantees the convergence of our proposed `FedGEM` algorithm for the isotropic GMM under study. Note, however, that this is also a purely technical result required only for the theoretical convergence analysis. However, it is not needed for use of our algorithm in practice.

- **Theorem 9: Client-to-Server Communication DP.** This theorem is provided as part of a preliminary DP discussion. It provides the minimum standard deviation of the Gaussian noise to be applied to the maximizers shared by the clients to guarantee DP.

## F.3 LIMITATIONS AND FUTURE WORK

This paper lays the foundation for a wide array of future work that can provide significant contributions and advance the fields of clustering, federated learning, and unsupervised representation learning via mixture models. Next, we discuss some of the limitations of our work, which should be addressed in future work.

- **Fixed Cluster Weights.** While our algorithm allows each client to set personalized local weights for their local clusters, these weights are fixed. In order to enhance modeling flexibility and personalization capabilities, future efforts should extend our algorithm to include trainable local cluster weights.

- **Stylized Clustering Model.** While the `FedGEM` algorithm we propose is generic, we mainly focus on its use with an isotropic GMM in this work. Future work may improve real-world performance by utilizing our algorithm with more complex mixture models, potentially studying anisotropic GMMs with locally learnable cluster weights. This would be theoretically challenging as it would involve verifying the needed assumptions, as well as deriving a tractable formulation for the local radius problem. Moreover, this may require an alternative convergence analysis approach, such as one that focuses on convergence to stationary points rather than (neighborhoods of) global maximizers. Furthermore, such efforts would need to study how the use of such complex models impacts the aggregation process at the central server.

- **Differential Privacy.** While we do provide a preliminary discussion on privatizing the cluster centroids shared by each client via DP in Appendix B.4, privatizing the uncertainty set radius and studying convergence in more detail remains an open problem. Since the radius is computed via an optimization problem, a key theoretical contribution would be analyzing its sensitivity and deriving the appropriate DP budget.

- **Modeling Assumptions.** In order to derive theoretical convergence guarantees for our `FedGEM` algorithm, we make various modeling assumptions. While these assumptions are very commonly made, and do not severely impact performance in practice if they are violated as shown in Section 6, a valuable contribution would still be deriving convergence guarantees with relaxed assumptions.

- **Pairwise Server Computation.** In our proposed work, the server relies on pairwise comparisons between the clusters at all clients in order to infer overlaps. While we have shown in our Scalability Study in Appendix E that our algorithm scales well with problem size, scalability can be further improved. Future work may develop a more efficient algorithm to be used by the central server to infer cluster overlaps.

- **Full Client Participation.** While the proofs presented in this work would still hold under partial client participation, they do not account for the potential drift that can be experienced by stragglers. While convergence to a neighborhood of the global maximizer is proven for centroid estimates at all clients, client drift can cause the estimates to end up in relatively distant areas of that neighborhood. This can increase the sensitivity of the estimated number of clusters to the final aggregation hyperparameter. Future work may study this setting both from the theoretical and practical perspectives, providing stricter convergence guarantees for stragglers and potential strategies to ensure an accurate estimation of $K$.

- **Final Aggregation Radius Tuning.** While we present a reliable heuristic and a guideline that can be used to set the final aggregation radius in our algorithm, it still requires hyper-parameter tuning via cross-validation to exploit our algorithm's full performance potential. Such tuning can incur very large computational costs, and can also significantly affect DP guarantees. Future work may seek to explore more robust, data-driven and theoretically verified heuristics that can achieve near-optimal performance while minimizing the computational and privacy costs associated with cross-validation-based tuning.

