# OpenReview forum: "A Federated Generalized Expectation-Maximization Algorithm for Mixture Models with an Unknown Number of Components"
_ICLR.cc/2026/Conference — ICLR 2026 Poster_

### Official Review · Reviewer_vQdW · 2025-10-30

**Soundness:** 3
**Presentation:** 4
**Contribution:** 3
**Rating:** 6
**Confidence:** 3

**Summary:**

The authors discuss clustering in the federated setting. Data is distributed across clients, who share cluster centroids but follow different cluster weights, specifically, no client has data from all clusters. . The effective novelty of the paper is the ability of the proposed FedGEM to automatically infer the number of clusters across the federation.
The authors discuss convergence properties of FedGEM in the general case and for the special case of GMMs. The paper concludes with a variety of empirical studies, specifically showing that FedGEM works even if "real" data is not well-separated, thereby invalidating a core prerequisite for the theoretical analyses (concavity).

**Strengths:**

The paper appears to be thorough and well-executed. The authors spend time analyzing a lot of aspects of their method in an extensive Appendix. The methods limitations are well-covered along with its strengths.
The problem of federated clustering is highly relevant, the method is original and automatic cluster count detection unlocks a new capability.
Note that I did not check derivations.

**Weaknesses:**

I do not follow the paper's motivation about OEM fault detection. How exactly does this relate to (federated) clustering? The authors claim the dimensionality of the data poses a problem - however all of their experiments are well within the range of internet-transferable sizes. (The largest dataset across clients comes in at ~17mb).
Although the paper describes some limitations of the proposed method, I see a few more that I think need to be addressed:
- The entire analysis and experimental result assumes full client participation. A realistic federation might only have a subset of clients participate at every round. The server-side aggregation would require rethinking
- The final aggregation radius needs to be determined through cross-validation. This is highly impractical in the federated setting as it would involve running the entire algorithm end-to-end multiple times. Furthermore, the optimal radius depends on the data geometry, scaling the feature space would require a new radius.
- It is unclear how anisotropic covariances would change the nature of aggregation radii
- Cross-validation incurs a huge price for DP guarantees

missing communication cost quantification. The authors note that AFCL requires clients sharing arrays of the same size as the local data. That price might be worth wile to pay (in terms of communication cost) if the total number of communication rounds is smaller due to a faster convergence rate. Especially as the number of clusters per round is variable, per-iteration costs of transmitting centroids could be high.

**Questions:**

If the authors could discuss the limitations I believe to be missing in the paper and also discuss communication costs, I'll consider raising my score!

---

> ### Author Response · Authors · 2025-11-20
>
> We thank the reviewer for their very insightful review of our work, which has helped us improve the quality and presentation of our paper. In the following, we split our response into three separate comments. The first discusses and clarifies our motivating problem, the second discusses the additional limitations pointed out by the reviewer and how we have discussed them in the updated manuscript, and the third provides a discussion regarding the communication costs.
>
> **Relation to OEM fault detection motivating problem.**
>
> In our motivating problem, an OEM manufactures multiple identical pieces of industrial equipment to be used by multiple different clients across the world. During operation, the clients collect condition monitoring data from the machines. The OEM is interested in utilizing this data to potentially train a classification model for fault diagnosis, which requires prior knowledge of all the unique classes in the data. However, the OEM **does not know the total number of unique faults experienced across the clients a priori**. This is because with modern complex machinery, it can be impossible to enumerate all the potential faults before deployment and use on site. Additionally, **the clients cannot share their local data with the OEM**. This could be due to stringent privacy constraints if the equipment is associated with the national security and infrastructure of different countries (e.g. gas turbines), which imposes strict data sovereignty regulations, or if the clients are competing firms. This could also be due to bandwidth limitations coupled with the large cardinality and dimensionality of the data. Admittedly, we only use medium to relatively large-scale data to validate our model due to computational limitations. However, in real-world settings the datasets can have a massive scale, which can render sharing them with the OEM infeasible. For example, a single gas turbine can generate GBs of data per day thanks to the extensive modern sensor suites. The final challenge the OEM faces is that **it cannot infer the total number of unique clusters from client-provided labels**. This is because there are no guarantees that clients use a standardized labeling convention. Indeed, two clients may give the same label to different faults, or they may give different labels to the same fault. As a result, the OEM faces a **fault discovery problem**. More precisely, the OEM must **infer the total number of fault clusters across the clients without relying on any data sharing or client labels**. This is exact problem our work addresses, as our FedGEM algorithm allows the OEM to infer this number of clusters using centroids and uncertainty sets provided by the clients rather than relying on raw data or labels. This allows the OEM to construct a registry of all the unique faults experienced across the clients, as well as which client has experienced which faults.

---

> ### Author Response · Authors · 2025-11-20
>
> **Additional limitations.**
>
> 1. **Full client participation.** We thank the reviewer for bringing up this very sharp point. We have added a discussion in our limitations section on page 49 addressing this point. The discussion highlights that while our convergence proofs do not explicitly rely on full client participation, in practice stragglers can lead to client drift. This can affect the accuracy of the estimated number of clusters in the final aggregation stage, thereby potentially increasing hyperparameter sensitivity. We have suggested that theoretically analyzing straggler behavior can be a very interesting area for future work.
>
> 2. **Tuning and scaling of final aggregation radius.** We thank the reviewer for highlighting this crucial point. We have addressed the hyperparameter scaling by adding further discussions on hyperparameter setting procedures to our paper, and we have added a discussion on the need for tuning through cross-validation to the limitations. More specifically, we note that while it was not initially clear, we did not directly tune the final aggregation radius in our experiments. Instead, we used practical heuristic that depends on the estimated minimum cluster separation and the number of samples at each client. The heuristic also involves a scalar multiplier, which is the hyperparameter we directly tune. This allows the radii to adapt to each cluster at each client, and to scale both with the scale of the feature space and the number of available data samples per client while only requiring the tuning of a single hyperparameter. We have included an additional discussion in the main body of the paper on page 8 highlighting this heuristic, and we expanded the discussion in the appendix on page 42 significantly to further elaborate on the practical tuning protocol. Finally, we have also added a discussion to the limitations on page 49 discussing that the need for hyperparameter tuning to exploit the full performance potential of our algorithm can indeed be a limitation. Therefore, future work may seek to explore more theoretically-driven heuristics that guarantee near-optimal performance while minimizing computational cost required for tuning.
>
> 3. **Anisotropic variances.** We thank the reviewer for raising this interesting point. We have added a discussion to the second limitation on page 48, highlighting that the use of isotropic GMMs is indeed a limitation of the work, and therefore future work should either verify assumptions or establish alternative convergence guarantees via different techniques for anisotropic GMMs. We note that anisotropic variances may only indirectly affect the *nature* of the aggregation radii. This is because the uncertainty sets are intended to capture potential perturbations in the estimated cluster centroids without hindering convergence, and the concept of the radii relies on the strong concavity of the local expected complete-data log-likelihood function. Therefore, if the variances are ill-conditioned such that they affect the strong concavity, they can impede the use of uncertainty sets while maintaining convergence. This is why future work may need to rely on different convergence guarantees (such as convergence to stationary points) in such edge cases.
>
> 4. **Price of cross-validation on DP.** We thank the reviewer for bringing this excellent point to our attention. We have indeed made sure to discuss it in the last limitation which we have added in page 49 in the manuscript.

---

> ### Author Response · Authors · 2025-11-20
>
> **Communication cost discussion.**
>
> We thank the reviewer for providing us with the opportunity to discuss this key point. We have added a detailed discussion on page 36 of our updated paper comparing the anticipated communication cost of our algorithm to that of AFCL. This discussion reads as follows:
>
> During each communication round of our algorithm, each client $g$ sends $K_g$ arrays of size $d$ and $K_g$ scalars to the central server, and receives $K_g$ arrays of size $d$. This results in a per-round total communication cost of approximately $2dG\tilde{K_g} + G\tilde{K_g} \leq 3dG\tilde{K_g}$, where $\tilde{K_g}$ is the mean number of clusters per client. We compare this to the communication cost of AFCL (Zhang et al., 2025). Due to its asynchronous nature, we assume that only $10$% of the clients participate in each communication round (a favorable condition for AFCL). In AFCL, each active client sends $N_g$ arrays of size $d$ to the central server, and receives $\widehat{K}$ arrays of size $d$, where $\widehat{K}$ is the estimated number of clusters. Under the assumption of roughly balanced client sample sizes, it is clear that the total per-round communication cost is approximately $0.1dG\tilde{N_g} + 0.1dG\widehat{K} > 0.1dG\tilde{N_g}$, where $\tilde{N_g}$ is the mean number of samples per client. Thus, our algorithm enjoys a lower per-round communication cost, since $\tilde{N}_g >> 30 \tilde{K}_g$ in most practical applications.
>
> Furthermore, we theoretically prove in Theorems 2 and 5 that our algorithm achieves a linear convergence rate for all clusters at all clients. In contrast, there is no theoretical convergence rate for AFCL. However, empirical findings in (Zhang et al., 2025) suggest a near-linear convergence rate at best. This suggests that our algorithm enjoys a lower total communication cost under the setting studied.

---

### Official Review · Reviewer_BFt3 · 2025-10-31

**Soundness:** 3
**Presentation:** 2
**Contribution:** 3
**Rating:** 4
**Confidence:** 3

**Summary:**

This submission presents a federated generalized expectation-maximization (FedGEM) algorithm for clustering in a setting where clients have heterogeneous data and the total number of clusters across the clients is unknown. FedGEM involves clients performing local EM steps to identify local cluster centers and then constructing uncertainty sets around these centers. These sets (centers and radiuses) are communicated to a server, which infers overlaps between local clusters to collaboratively train shared cluster parameters and estimate the possible total number of clusters. The authors provide a theoretical analysis of the convergence of the FedGEM and its ability to correctly infer the cluster count. The empirical evaluation demonstrates that FedGEM achieves state-of-the-art performance, outperforming the mentioned baselines.

**Strengths:**

- Sound Validation: The extensive experimental results demonstrate the effectiveness of the FedGEM. The results in Table 1 show that FedGEM outperforms AFCL, the only other federated baseline that operates without knowing K. Also, FedGEM is competitive with federated methods that are given the true K in advance. The sensitivity study further indicates the robustness of FedGEM, showing strong performance even when theoretical assumptions like well-separated clusters are violated.
- Theoretical Analysis: This submission is supported by a theoretical analysis. The authors provide probabilistic convergence guarantees for the general FedGEM algorithm under standard assumptions (Theorems 1, 2, and 5). Also, this submission provides a detailed analysis for the isotropic GMM setting, where they prove the First-Order Stability (FOS) condition for multi-component GMMs (Theorem 6).

**Weaknesses:**

- Generality: The implementation, theoretical verification, and experiments are based on an isotropic GMM. However, its performance on real-world data with arbitrarily shaped clusters might be limited.
- Computational Complexity: The server is required to identify cluster overlaps involves pairwise comparisons between all local components from all clients. This is a computational complexity that scales quadratically with the total number of local components across the network. Though the scalability study in the appendix shows good performance, this could become a bottleneck in FL settings with thousands of clients.
- Presentation and Grammar Issues: 1) grammar issue: line 96-97; 2) the names of the mentioned methods should be in the same font if the authors would like to emphasize

**Questions:**

- How to set `final_aggregation_radius`? Could you provide more practical heuristics for setting this value? How sensitive are the ARI and the estimated K to this hyperparameter?
- Others, please refer to weakness.

---

> ### Author Response · Authors · 2025-11-20
>
> We sincerely thank the reviewer for their helpful and constructive comments on our work, which have helped us to improve the quality, presentation, and impact of our paper. Next, we discuss the points raised by the reviewer in detail. We divide our response into two comments, where the first addresses the weaknesses while the second addresses the question raised by the reviewer.
>
> **Weaknesses**
>
> 1. **Generality.** We thank the reviewer for highlighting this point. While our theoretical analysis and implementation of the algorithm indeed focus on isotropic GMMs, we highlight that we have **verified via a Henze-Zirkler multivariate normality test** that the datasets used in our numerical experiments are indeed **non-Gaussian**. Yet, our numerical experiments reflect the very strong performance of our algorithm both in the clustering performance and number of clusters estimation. Therefore, despite the modeling limitation, our model is still capable of significantly outperforming the state-of-the-art on real-world data with arbitrarily-shaped clusters. Furthermore, we note that the isotropic Gaussian assumption is indeed common in impactful works in the literature that take a similar approach to proving convergence as ours, including (Balakrishnan et al., 2014), and (Yan et al., 2017). However, we do acknowledge that the focus on isotropic Gaussian can indeed be a limitation in some cases, and that performance may be further enhanced by considering more complex mixture models. To address this, we have expanded our discussion in the limitations on page 48 to highlight that considering anisotropic GMMs offers a great opportunity for impactful future work to further advance our proposed approach.
>
> 2. **Computational complexity.** We thank the reviewer for highlighting this excellent point, as it is indeed true that the pairwise comparisons performed by the server can be a bottleneck in our algorithm. To address this limitation, we have added a discussion on page 36 of our updated manuscript proposing a methodology to improve the efficiency of server computations. More specifically, we first note that in the pairwise scheme, the server needs to perform close to $\mathcal{O}(G^2K^2)$ comparisons between clusters, which as the reviewer highlights, can be extremely expensive when $G$ is very large. Therefore, we suggest that the server store the cluster centroids shared by all clients within a d-dimensional binary search tree (commonly known as a KD-tree). Subsequently, the server can iterate over each centroid, and obtain its $M$ nearest neighbors. This would require the server to only check for uncertainty set intersections with the $M$ neighbors rather than all the other centroids. We show in our discussion that in practice, the complexity of all the server operations in this approach would be close to $\mathcal{O}(GK \log{(GK)})$, which includes constructing the tree and performing all the searches. Therefore, this approach offers a significant computational improvement over the original $\mathcal{O}(G^2K^2)$. Moreover, since it uses a very standard and well-studied data structure, it can be directly integrated within our algorithm without the need for any changes or affecting any of the theoretical guarantees.
>
> 3. **Presentation and grammar.** We thank the reviewer for bringing this to our attention. We have taken steps to ensure that any grammar or notation issues in the manuscript have been addressed.

---

> ### Author Response · Authors · 2025-11-20
>
> **Question: Setting final aggregation radius, practical heuristics and performance sensitivity.**
>
> We thank the reviewer for highlighting this key point and giving us the opportunity to elaborate on it. While it was admittedly not very clear in the original manuscript, we indeed use a heuristic to set the final aggregation radius for each cluster at each client. The heuristic includes the estimated minimal cluster separation as well as the number of samples at each client. The heuristic also includes a scalar multiplier, which is the hyperparameter we tune directly. This heuristic offers two key advantages over directly tuning the final aggregation radii. Firstly, it allows the radii to scale with the scale of the feature space as well as the number of available training samples at each client. Secondly, it adapts to each cluster at each client while only requiring the user to tune a single hyperparameter value. We have added a brief discussion highlighting the use of the heuristic on page 8. Additionally, we have added a detailed guideline on page 42 on setting the hyperparameter value from a practical perspective, relying on both the aforementioned heuristic as well as a practical check for the estimated number of clusters. The hyperparameter tuning protocol provided relies on theoretical intuitions, while still being practical and readily usable in real-world applications. Finally, we have performed additional experiments to evaluate the sensitivity of the estimated $K$ and the clustering performance to the hyperparameter value, and we present the results on pages 43 and 44 of the updated manuscript. In this experiment, we evaluate ARI and the estimated number of clusters over a range of different hyperparameter values. The experiment reveals that the estimation of $K$ can be more sensitive to the hyperparameter choice than the ARI. However, as our other numerical results suggest, hyperparameter tuning via the practical guidelines we provide can allow our algorithm to achieve strong performance across a range of datasets.

---

### Official Review · Reviewer_iREx · 2025-11-01

**Soundness:** 3
**Presentation:** 3
**Contribution:** 3
**Rating:** 6
**Confidence:** 3

**Summary:**

The paper introduces FedGEM, a federated generalized EM algorithm for mixture models with an unknown global number of components (\\(K^\*\\)). Each client runs a local GEM on an isotropic GMM with fixed mixture weights and covariances, updating only component means and computing uncertainty radii. The server merges overlapping components based on these uncertainty sets and performs constrained parameter updates. Theoretical results establish finite-sample and population convergence, as well as a sufficient condition for correct (\\(K^\*\\)) recovery. Experiments on synthetic and benchmark datasets show competitive clustering accuracy and moderate scalability.

**Strengths:**

# originality
Federated EM with unknown global K via uncertainty-set–based merging at the server is an interesting angle relative to standard federated clustering/EM, and distinct from k-means-style approaches

# quality

The paper presents local convergence results for the GEM variant, as well as finite-sample versus population map deviation bounds under the stated assumptions.

the radius subproblem has a stated unique solution and a low-complexity solver with a convergence and complexity discussion

# clarity

Problem setup and the server pseudo-code for super-clustering/aggregation are explicit

Assumptions are listed and some are justified in an appendix

# significance

Unknown K across many clients arises in FL. The approach could be a useful building block if the server-side merging is robust and the hyperparameters are chosen well. Empirical results indicate competitive performance and a reasonable scaling trend

**Weaknesses:**

# originality

The method is positioned generally ("mixture models with unknown K"), but the analysis and implementation hinge on isotropic Gaussians with fixed weights; this narrows the contribution relative to the stated ambition. Extending to anisotropic covariances or learning pi would be more compelling.

# quality

weights fixed, covariance fixed to identity, Kg known locally. the paper also does not study the effects of mispecified pi.

The overlap-detection step entails pairwise checks over all client components, i.e., quadratic in the total number of components. the paper relies on empirical timing rather than giving a tight complexity analysis for this stage. This should also be discussed as a limitation.

it is not analyzed whether a client (or all clients) stuck at a sharp local maximizer will "snap back" after the server’s within-set update.



# clarity

Assumption 6 (supremum variable vs. the conditioning argument of M / $\hat{M}$
The role and definition of the strong concavity parameter ​$\lambda_g$ are not introduced where first used (Assumption 3)

# significance

Final aggregation radius $\epsilon^{final}$ is a user hyperparameter. While a sufficient condition wrt Rmin is given, Rmin is not known. The paper should indicate a protocol for choosing this hyperparameter

No discussion of communication/computation trade-offs vs. alternative K estimation strategies (e.g., centralized model selection surrogates, Bayesian nonparametrics) in FL.

**Questions:**

See weaknesses.

Other questions:

Anisotropic models: Do you foresee obstacles to extending the convergence and finite-sample analysis to full-covariance Gaussians (or even tied/diagonal covariances)? What breaks in the proofs?

Is fixing pi essential for your bounds, or could pi be updated (with constraints) without derailing the strong concavity/FOS arguments?

---

> ### Author Response · Authors · 2025-11-20
>
> We sincerely thank the reviewer for their very thorough and detailed review of our work, which has helped us improve the presentation and impact of the paper. We split our response into three comments: the first addresses the originality, clarity, and significance weaknesses, the second addresses the quality weaknesses, and the third addresses the questions.
>
> # Weaknesses
>
> **Originality**
>
> 1. We thank the reviewer for allowing us to elaborate on this key point. We note that our proposed algorithm itself is generic, and can be applied to mixture models that obey the modeling assumptions required for the convergence of our algorithm. We focus on proving the assumptions for the isotropic GMM case, and we empirically demonstrate that the relatively simple model structure does not prevent our approach from significantly outperforming the state-of-the-art in federated clustering with an unknown number of clusters. Indeed, we verify via a **Henze-Zirkler multivariate normality test** that the datasets used in our numerical experiments are indeed **non-Gaussian**. Nevertheless, our proposed algorithm still consistently outperforms existing methods, highlighting its significant practical impact despite its limitations.
>
>    We still acknowledge that utilizing learnable weights or anisotropic variances can advance our approach and potentially improve performance even further in practice. However, such extensions can be heavily theoretically involved as we discuss in our answers to the reviewer questions later in the response. Thus, we leave such extensions for future work.
>
> **Clarity**
>
> 1. We thank the reviewer for these very sharp observations. We have addressed both issues in the updated manuscript.
>
> **Significance**
>
> 1. We thank the reviewer for bringing up this excellent point and allowing us to further elaborate on it. While it was not initially clear, we used a practical heuristic to tune our final aggregation radii in the numerical experiments. We have added a note on the heuristic on page 8 of the updated mansucript. We have also added a detailed discussion on the heuristic and practical tuning guidelines on page 42 of the manuscript. The heuristic we use relies on the empirically estimated $R_{min}$ and the number of samples at each client, and also uses a scalar multiplier which is the hyperparameter we directly tune. This allows the final aggregation radii to naturally scale with the scale of the feature space and the number of samples available at each client. Additionally, it allows for final aggregation radii that adapt to each cluster at each client while only requiring the tuning of a single hyperparameter. Furthermore, we also provide a practical evaluation guideline for the estimated $\widehat{K}$ without knowledge of the true $K$.
>
> 2. We thank the reviewer for highlighting this point. We have added a discussion to page 36 of our manuscript exploring the communication cost incurred by our algorithm and comparing it to that incurred by AFCL, which is the only other federated clustering algorithm with an unknown $K$. This discussion reads as follows:
>
>    During each communication round of our algorithm, each client $g$ sends $K_g$ arrays of size $d$ and $K_g$ scalars to the central server, and receives $K_g$ arrays of size $d$. This results in a per-round total communication cost of approximately $2dG\tilde{K_g} + G\tilde{K_g} \leq 3dG\tilde{K_g}$, where $\tilde{K_g}$ is the mean number of clusters per client. We compare this to the communication cost of AFCL (Zhang et al., 2025). Due to its asynchronous nature, we assume that only $10\$% of the clients participate in each communication round (a favorable condition for AFCL). In AFCL, each active client sends $N_g$ arrays of size $d$ to the central server, and receives $\widehat{K}$ arrays of size $d$, where $\widehat{K}$ is the estimated number of clusters. Under the assumption of roughly balanced client sample sizes, it is clear that the total per-round communication cost is approximately $0.1dG\tilde{N_g} + 0.1dG\widehat{K} > 0.1dG\tilde{N_g}$, where $\tilde{N_g}$ is the mean number of samples per client. Thus, our algorithm enjoys a lower per-round communication cost, since $\tilde{N}_g >> 30 \tilde{K}_g$ in most practical applications.
>
>    Furthermore, we theoretically prove in Theorems 2 and 5 that our algorithm achieves a linear convergence rate for all clusters at all clients. In contrast, there is no theoretical convergence rate for AFCL. However, empirical findings in (Zhang et al., 2025) suggest a near-linear convergence rate at best. This suggests that our algorithm enjoys a lower total communication cost under the setting studied.

---

> ### Author Response · Authors · 2025-11-20
>
> # Weaknesses
> **Quality**
>
> 1. - **Fixed weights and covariances:** We address these points in detail in our answers to the questions raised by the reviewer.
>    - **Locally known $K_g$:** Our known $K_g$ assumption stems from our motivating distributed fault discovery problem. This is because in our problem, each client would be aware of the number of faults that occurred locally on their machine. However, the central server (which represents the original manufacturer of the machines, OEM) would not have knowledge of the total number of *unique* faults experienced across all the potentially privacy-restricted clients. Furthermore, the server cannot rely on client-provided labels to infer the number of unique clusters due to non-standardized labeling conventions. This assumption can be applicable in many practical settings, where each client is aware of the number of unique clusters that have been experienced in their local data, but the labels from different clients cannot be reconciled to infer the total number of unique clusters across the clients.
>    - **Effect of misspecified $\pi$:** We highlight that our numerical experiments indeed do involve misspecified cluster weights. While the Abalone, Frog A, and Frog B datasets all include significant imbalance across the clusters, our cluster weights did not reflect this aspect. Instead, weights were assigned equally for all clusters at each client. Nonetheless, our model still outperformed the relevant benchmarks both in clustering performance and number of clusters estimated. This suggests the potential robustness of our approach in practice to misspecified cluster weights.
>
> 2. We thank the reviewer for highlighting excellent point, as we do indeed acknowledge that the pairwise server computations can be inefficient in problems with a very large number of clients. To address this challenge, we have added a discussion to page 36 of our updated manuscript exploring how the server computations can be made more efficient. To that end, we argue that in the worst-case, the server would need to perform close to $\mathcal{O}(G^2K^2)$ computations under the pairwise comparison scheme. However, the server may instead choose to store all the centroids provided by all clients in a standard d-dimensional binary search tree (commonly known as a KD-tree). This would allow the server to then iterate over each centroid, and check for uncertainty set intersections with only the its $M$ nearest neighbors rather than comparing with all the available centroids. We show that in practice, the complexity of constructing the tree and using it in this fashion can be close to roughly $\mathcal{O}(GK \log{(GK)})$, making it significantly more efficient than the pairwise scheme. Additionally, since this approach relies on standard algorithms, it does not affect our proposed algorithm or convergence results in any way. Finally, we have also made sure to highlight the pairwise computations as a limitation of the work on page 49, and suggested that future work may seek to develop more efficient approaches to perform the server computations.
>
> 3. We thank the reviewer for raising this very interesting point. Firstly, let us consider the population (infinite training sample) setting. In this case, we prove in Theorem 5 that the iterates generated by our algorithm for each cluster at each client converge exactly to the ground truth parameters (i.e. a global maximizer). Therefore, no clients can get stuck at a sharp local maximizer in this setting. Similarly, for the finite-sample setting, we prove in Theorem 2 that the iterates generated by our algorithm for each cluster at each client converge to some neighborhood of the global maximizer. Therefore, different clients' estimates can converge to different points, but all such points must fall within a certain radius from the global maximizer. We note that under our modeling assumptions, if a client is "stuck" at a local maximizer that means its estimates have already converged to a certain point within the neighborhood of the global maximizer. Therefore, its uncertainty set radius would be 0 due to the strong concavity of the finite-sample expected complete-data log-likelihood function, and the centroid would not be further updated during the collaborative training stage. In the final aggregation stage, the final aggregation radii associated with all estimates of a given cluster cover the entire neighborhood of its global maximizer without including estimates of other clusters. This allows all the client estimates to be aggregated together at this stage, despite getting stuck during collaborative training.

---

> ### Author Response · Authors · 2025-11-20
>
> # Questions
>
> 1. We thank the reviewer for this very sharp question. Changing from fixed isotropic covariances to (potentially learnable) anisotropic ones would require entirely new proofs verifying the theoretical modeling assumptions for the mixture model. More specifically, it would require new proofs of the FOS condition, the contraction radius, and the existence of an upper bound on the distance between the population and finite-sample M-steps. This can be heavily theoretically involved. For context, the FOS condition was only previously proven for 2-component univariate GMMs with fixed variances and opposite means in (Balakrishnan et al., 2014). Additionally, if such proofs are not possible, the implementation of anisotropic variances within our work may require a different approach for analyzing convergence. Such approach may adopt weaker convergence guarantees, focusing only on convergence to stationary points rather than (neighborhoods of) global maximizers. For that reason, we view such developments as outside the scope of this work, and as a great area for interesting future extensions. Despite this limitation, we would like to highlight that we verified via a statistical test that the datasets used in our empirical evaluations were indeed non-Gaussian. Nevertheless, our algorithm was capable of consistently outperforming the relevant benchmarks, highlighting its practical impact despite its relatively simple structure.
>
> 2. We again thank the reviewer for this great question. Fixed weights are indeed an essential part of our FOS argument. The implementation of learnable weights would require an update of the FOS argument to also capture potential changes in weights and prove that the expected complete-data log-likelihood enjoys FOS over the weights as well. Similar to the anisotropic covariances, if such a proof is not possible, one may need to resort to weaker convergence guarantees. For this reason, we also view this as out of scope for our current work. However, we would like to emphasize that even without learnable weights and anisotropic covariances, our model is capable of *significantly outperforming* the state-of-the-art in federated clustering with an unknown number of clusters as we show in our experiments.

---

### Official Review · Reviewer_aUfD · 2025-11-07

**Soundness:** 3
**Presentation:** 3
**Contribution:** 3
**Rating:** 4
**Confidence:** 3

**Summary:**

This paper proposes FedGEM: a federated generalized expectation-maximization algorithm for the training of mixture models with an unknown number of components. The algorithm relies on each of the clients performing EM steps locally, and constructing an uncertainty set around the maximizer associated with each local component. The central server utilizes the uncertainty sets to learn potential cluster overlaps between clients, and infer the global number of clusters via closed-form computations.

This paper performs a thorough theoretical study of our algorithm, presenting probabilistic convergence guarantees under common assumptions. Subsequently, this paper studys the specific setting of isotropic GMMs, providing tractable, low-complexity computations to be performed by each client during each iteration of the algorithm, as well as rigorously verifying assumptions required for algorithm convergence. This paper also performs various numerical experiments.

**Strengths:**

1. The structure of the paper is relatively clear.
2. The theoretical derivation and proof in the paper are quite thorough.

**Weaknesses:**

There are only two compared algorithms that do not assume prior knowledge of K. Among them, one is the algorithm proposed in 1974, and the other is AFCL, which uses entirely different datasets from those in this paper. These two aspects result in flaws in the experimental design, making it difficult to truly reflect the algorithm's performance.

**Questions:**

As above.

---

> ### Author Response · Authors · 2025-11-20
>
> We thank the reviewer for their review and comments on our work. We address the weaknesses raised by the reviewer next.
>
> **Only two benchmarks with an unknown number of clusters.** We note that federated clustering approaches with an unknown $K$ are very limited in the literature. Indeed, AFCL is the only federated clustering approach that addresses this setting to the best of our knowledge. The remainder of the existing algorithms that assume unknown $K$ are all *centralized*. Nevertheless, we still wanted to compare our federated algorithm to a centralized benchmark that assumes an unknown $K$ to provide some context to our results and the performance of our algorithm. In doing so, we focused on the DP-GMM algorithm as it remains the standard GMM-based method for clustering with an unknown $K$ in a centralized setting, making it a very relevant benchmark in our work. Notably, our algorithm consistently outperforms DP-GMM, even though DP-GMM operates in a fully centralized setting. This underscores that significant practical effectiveness of our proposed model, despite the constraints of federation.
>
> **Choice of datasets.** We would like to emphasize that the Abalone dataset was used both in our work and in the work introducing AFCL. Nevertheless, we demonstrate that our algorithm still outperforms AFCL on this dataset both in clustering performance and number of clusters estimation. With regard to the use of other datasets, we have made every effort to ensure experimental fairness and transparency. To that end, we set any hyperparameters as recommended in the original AFCL paper, we assume full client participation for both algorithms, and we use identical data splits for all repetitions across both algorithms. Our motivation in using other datasets is to include a mix of small to relatively large-scale problems, thereby allowing us to comprehensively examine the performance of our algorithm across different numbers of clusters, samples, and clients. Finally, we highlight that AFCL is not specifically tailored to the datasets used in its evaluation. Thus, it is entirely fair to compare its performance to that of our algorithm on different datasets if all aspects of the experimental design are fair and transparent, as is the case in our study.

---

### Author Response · Authors · 2025-11-20

We sincerely thank all the reviewers for their very helpful and insightful reviews on our work. The thorough insights provided by the reviewers have helped us significantly improve the quality, presentation, and impact of our paper. Next, we enumerate the changes made to the paper. Please note that all changes are made in a **blue** font in the updated manuscript to facilitate ease of review. Please see the updates next:

1. **Efficient Server Computations.** We have added a discussion to page 36 of the manuscript discussing the potential use of a KD tree to make the server computations more efficient than the pairwise comparison scheme. We have theoretically examined the potential computational gains for utilizing this approach, and we have highlighted that implementing this approach does not affect any other aspects of our algorithm, including convergence guarantees.

2. **Communication Costs.** We have added a discussion to page 36 of our manuscript theoretically discussing the communication cost incurred by our algorithm and comparing it to that incurred by AFCL. To that end, we argue that in many practical settings our algorithm incurs a smaller communication cost. This, coupled with its stronger performance, makes it preferable from a practical perspective

3. **Practical Guidance for Hyperparameter Tuning.** We have highlighted a practical heuristic that we used for setting our final aggregation radius on page 8 of the manuscript, while also providing a thorough practical guideline on hyperparameter tuning on page 42. To that end, our heuristic offers two key advantages. Firstly, it allows the final aggregation radii to naturally scale with the scale of the feature space as well as the number of samples at each client. Secondly, it allows for final aggregation radii that adapt to each cluster at each client while only requiring the tuning of a single hyperparameter.

4. **Hyperparameter Sensitivity.** We have added an experiment on pages 43 and 44 of our updated manuscript examining the sensitivity of our algorithm's ARI and estimated number of clusters to the final aggregation radius hyperparameter.

5. **Limitations.** We have expanded our discussion of the limitations of our work on pages 48 and 49 to further discuss topics such as use of isotropic GMMs and the need for cross-validation for hyperparameter tuning. We note that we view these limitations as rich areas for future work to build on the foundations established by our work.

6. **Typographical and Notation Issues.** We have carefully examined the manuscript and addressed any spelling or grammar issues, as well as any issues with notation.

We thank the reviewers again for their very helpful and insightful comments. We look forward to engaging in fruitful discussions and addressing any further questions, comments, or concerns.

---

### Meta-Review · Area_Chair_N2ZS · 2026-01-02

**Summary:**

Strengths. The methodology of uncertainty-set-based merging to effectively handle federated clustering is novel and interesting. The paper also has strong theoretical results backing up its algorithm. Furthermore, it demonstrates competitive empirical performance in a thorough experimental setup.

The reviews surfaced a number of limitations that have mostly been addressed by the rebuttal. The authors also made substantial changes to their submission to incorporate the reviewers' feedback.

However, there is still one major experiment missing. I strongly encourage the authors to add an a comparision to known $K$ methods coupled with standard heuristics for picking $K$.

**Reviewer Concerns:**

- Reviewer aUfD
  - Weak empirical results since very few baselines were used (only compared to AFCL and an 1974 algorithm) - partially adressed. The rebuttal states this is because AFCL is the only method designed for unknown $K$. However, just like authors' algorithm requires heuristics and tuning of parameters, standard heuristics like the Elbow method or CV could have been used to pick $K$ and fairly compare the different algorithms. This is especially important since FedKmeans seems to consistently outperform the current algorithm.

- Reviewer iREx
  - Theory restricted to unrealistic isotropic Gaussians with fixed weights assumption - not adressed (limitations acknowledged but not fixed). The response acknowledges the limitation and states proof would need to be completely reworked to generalize. They also state that the empirical tests on non-Gaussian data show the method is robust despite these assumptions, but (as reviewer aUfD notes) the experiments are too limited to demonstrate this fully.
  - Server computation cost is quadratic - adressed. Authors propose using a KD-tree structure to reduce this computation cost.
  - Can clients stuck at local maximizers "snap back" during aggregation - adressed. The response clarifies that aggregation radii are designed to allow even "stuck" estimates to be correctly merged.
  - Tuning final_aggregation_radius - adressed. Authors clarified the heuristic used.
  - Communication cost comparision - partially adressed. Authors compared to AFCL, but would should also compare with known $K$ baselines with heuristic choice of $K$.


- Reviewer BFt3
  - Unrealistic isotropic Gaussians assumption -  not adressed (see  iREx).
  - Quadratic computation cost for server - adressed (see  iREx)
  - Tuning final_aggregation_radius - adressed (see  iREx)

- Reviewer vQdW
  - Relation of OEM fault detection to clustering - adressed. The task is to find total number of unique machine faults across clients.
  - Unrealistic assumption of full client participation - not adressed (limitations acknowledged but not fixed). Response acknowledges this limitation and added a discussion of this.
  - Cost of CV under privacy constraints - not adressed (limitations acknowledged but not fixed). Response admitted this trade-off but stated their new heuristic reduces tuning effort.
  - Communication cost - partially adressed. (see  iREx)

**Reviewer Scores:**

- Reviewer aUfD would have retained **4** since more baselines or experiments were not added.
- Reviewer iREx would likely have retained their **6** since their major concern about isotropic Gaussians assumption was not addressed.
- Reviewer BFt3 had a subset of concerns of iREx and so after response would have increased to **6**.
- Reviewer vQdW had minor concerns adressed while other limitations acknowledged but not fixed. They explictly said they would raise their score after adding limitations and comparing communication costs. So they would have increased to **8**.

Average of 6.

---

### Decision · Program_Chairs · 2026-01-26

Accept (Poster)